# Mean Estimation in High-Dimensional Binary Markov Gaussian Mixture Models

**Yihan Zhang**
Institute of Science and Technology, Austria
zephyr.z798@gmail.com

**Nir Weinberger**
Technion - Israel Institute of Technology
nirwein@technion.ac.il

## Abstract

We consider a high-dimensional mean estimation problem over a binary hidden Markov model, which illuminates the interplay between memory in data, sample size, dimension, and signal strength in statistical inference. In this model, an estimator observes $n$ samples of a $d$-dimensional parameter vector $\theta_* \in \mathbb{R}^d$, multiplied by a random sign $S_i$ ($1 \leq i \leq n$), and corrupted by isotropic standard Gaussian noise. The sequence of signs $\{S_i\}_{i \in [n]} \in \{-1, 1\}^n$ is drawn from a stationary homogeneous Markov chain with flip probability $\delta \in [0, 1/2]$. As $\delta$ varies, this model smoothly interpolates two well-studied models: the Gaussian Location Model for which $\delta = 0$ and the Gaussian Mixture Model for which $\delta = 1/2$. Assuming that the estimator knows $\delta$, we establish a nearly minimax optimal (up to logarithmic factors) estimation error rate, as a function of $\|\theta_*\|, \delta, d, n$. We then provide an upper bound to the case of estimating $\delta$, assuming a (possibly inaccurate) knowledge of $\theta_*$. The bound is proved to be tight when $\theta_*$ is an accurately known constant. These results are then combined to an algorithm which estimates $\theta_*$ with $\delta$ unknown a priori, and theoretical guarantees on its error are stated.

## 1 Introduction

Memory between data samples is ubiquitous in practical applications, as data collected from networks or sampled time series inevitably inherit spatial or temporal statistical dependencies. Numerous examples arise in imaging, meteorology, health care, finance, social science, and so on [Glaeser et al., 1996, Bertrand et al., 2000, Sacerdote, 2001, Duflo and Saez, 2003, Christakis and Fowler, 2013]. In principle, prior knowledge of the existence of such memory can be used to improve the performance of statistical estimators compared to the performance obtained in memoryless models. Development and analysis of statistical inference algorithms for models with memory has been extensively explored from the algorithmic perspective, and computationally efficient algorithms such as Baum-Welch and message-passing were developed [Ephraim and Merhav, 2002, van Handel, 2008, MacKay, 2003, Wainwright and Jordan, 2008]. It was also extensively studied from the theoretical perspective in the classical, fixed-dimensional and asymptotic regime, e.g., Györfi et al. [2002, Chapter 27 and references therein]. However, much less is understood about the high dimensional, non-asymptotic regime, which is of paramount importance in modern applications, and the focus of current extensive research [Vershynin, 2018, Wainwright, 2019]. As we exemplify in this paper, the error in such estimation problems depends in an intricate way on the interplay between the number of samples, the dimension of the vector parameters to be estimated, the noise level (signal-to-noise ratio), and the level of memory between the samples.

An ever popular and fundamental model is the Gaussian mixture model, in which memory exists between samples whenever the latent variables determining the component of each sample are dependent. Numerous recent papers, e.g., Balakrishnan et al. [2017], Xu et al. [2016], Klusowski and Brinda [2016], Jin et al. [2016], Dwivedi et al. [2020b], Dwivedi et al. [2018], Dwivedi et al.

36th Conference on Neural Information Processing Systems (NeurIPS 2022).

[2020a], Zhao et al. [2018], Yan et al. [2017], Weinberger and Bresler [2022] have focused on high-dimensional *memoryless* models, and analyzed computationally efficient estimation algorithms, most notably the expectation maximization (EM) algorithm. Specifically, the seminal Balakrishnan et al. [2017] has provided theoretical guarantees for EM in memoryless models, though without proving minimax optimality, which was subsequently established in Wu and Zhou [2019]. As a notable exception, Yang et al. [2015] has generalized the analysis of Balakrishnan et al. [2017] to a *hidden Markov model* (HMM) that has memory, yet again, without determining how the minimax error rate depends on the number of samples, dimension, noise level, and the amount of memory. In this paper, we address the question of precise characterization of the minimax error rate in terms of these parameters, in the context of a high-dimensional Gaussian HMM. We next turn to formally define this model and the estimation problem, describe known results, and then present our contributions. Our obtained results illuminate the opportunities and challenges associated with optimal inference in high dimensional models with memory.

## 1.1 Problem formulation

Let $S_0^n := (S_0, S_1, \ldots, S_n)$ be the following homogeneous binary symmetric Markov chain, $S_i \in \{-1, 1\}$, $\mathbb{P}[S_0 = 1] = 1/2$ and

$$S_i = \begin{cases} S_{i-1}, & \text{w.p. } 1 - \delta \\ -S_{i-1}, & \text{w.p. } \delta \end{cases} \tag{1}$$

for $i \in [n] := \{1, \ldots, n\}$, and where $\delta \in [0, 1]$ is the flip probability of the binary Markov chain. We also denote $\rho := 1 - 2\delta \in [-1, 1]$ which is the correlation between adjacent samples $\rho = \mathbb{E}[S_i S_{i+1}]$. At each time point $i \in [n]$, a sample of a $d$-dimensional Gaussian mixture model is observed

$$X_i = S_i \theta_* + Z_i, \tag{2}$$

where $Z_i \sim N(0, I_d)$ is an i.i.d. sequence, independent of $S_0^n$, and where $\theta_* \in \mathbb{R}^d$, $d \geq 1$. At its two extremes, this model degenerates to one of two fundamental models, which are well studied. When $\delta = 0$, the memory length is infinite, and the sign $S_0 = S_1 \cdots = \cdots = S_n$ is fixed. Thus, up to this sign ambiguity, the model (2) is the standard *Gaussian location model* (GLM), which is essentially a memoryless model (and exactly so if $S_0$ is known). When $\delta = \frac{1}{2}$, the signs $S_0^n$ are i.i.d. and have no memory at all. The model (2) is then a *Gaussian mixture model* (GMM) with two symmetric components, which is also a memoryless model. In all other cases, $0 < \delta < \frac{1}{2}$ (or $\frac{1}{2} < \delta < 1$), the model is a simple version of a HMM.

The inference problem we consider in this paper is the estimation of $\theta_* \in \mathbb{R}^d$, under the loss function

$$\mathsf{loss}(\hat{\theta}, \theta_*) := \min\{\|\hat{\theta} - \theta_*\|, \|\hat{\theta} + \theta_*\|\}, \tag{3}$$

that is, the Euclidean distance error under a possible sign ambiguity.[1] An intermediate goal (or an additional problem) is to estimate $\delta$, under the regular absolute error loss function $|\hat{\delta} - \delta|$. The fundamental limits of this estimation problem will be gauged by the *local* minimax rate, which is the maximal decrease rate of the loss possible for any estimator, given $n$ samples, at dimension $d \geq 2$, for signal strength $\|\theta_*\| = t$, and under flip probability $\delta$. Specifically, for $d \geq 2$ it is defined as

$$\mathsf{M}(n, d, \delta, t) := \inf_{\hat{\theta}(X_1^n)} \sup_{\|\theta_*\| = t} \mathbb{E}\left[\mathsf{loss}(\theta_*, \hat{\theta}(X_1^n))\right]. \tag{4}$$

For general $d \geq 1$, the *global* minimax rate is defined with the condition $\|\theta_*\| = t$ replaced by $\|\theta_*\| \leq t$ (this condition trivializes the estimator for $d = 1$).

## 1.2 Known minimax estimation errors rates for GLM and GMM

Before delving into models with memory ($0 < \delta < 1/2$), we review known results on the minimax rates in memoryless high dimensional Gaussian models – the GLM ($\delta = 0$) and the GMM ($\delta = 1/2$). We refer the reader to note that the minimax error rate in these models may undergo two possible phase transitions – one as $t$ increases, and the other one as $d$ increases. We also remark that the regime of interest is that of low-separation ($t \lesssim 1$), in which accurately detecting the components is impossible, yet parameter estimation with vanishing loss is possible.

---

[1]Similar bounds can be derived for the squared loss by trivial extensions.

At the first extreme, the local minimax rate for the GLM ($\delta = 0$) is the usual parametric error rate

$$\mathsf{M}_{\mathrm{GLM}}(n,d,t) := \mathsf{M}(n,d,0,t) \asymp \begin{cases} t, & t \leq \sqrt{\frac{d}{n}} \\ \sqrt{\frac{d}{n}}, & t \geq \sqrt{\frac{d}{n}} \end{cases}. \tag{5}$$

This is achieved by the trivial estimator $\hat{\theta} = 0$ if $t \leq \sqrt{\frac{d}{n}}$ and the simple empirical average estimator $\hat{\theta} = \frac{1}{n}\sum_{i=1}^{n} X_i$ if $t \geq \sqrt{\frac{d}{n}}$ (see [Wu, 2017, Sec. III-9]). The rate $\Theta(\sqrt{\frac{d}{n}})$ is then the *global* minimax rate, i.e., the largest error over $t = \|\theta_*\| > 0$. This model does not have a phase transition with dimension. At the other extreme, the GMM model ($\delta = \frac{1}{2}$) undergoes a phase transition at the dimension $d = n$. At low dimension, $d \leq n$, the minimax rate was neatly shown in [Wu and Zhou, 2019, Appendix B] to be

$$\mathsf{M}_{\mathrm{GMM}}(n,d,t) \equiv \mathsf{M}\left(n,d,\frac{1}{2},t\right) \asymp \begin{cases} t, & t \leq \left(\frac{d}{n}\right)^{1/4} \\ \frac{1}{t}\sqrt{\frac{d}{n}}, & \left(\frac{d}{n}\right)^{1/4} \leq t \leq 1 \\ \sqrt{\frac{d}{n}}, & t > 1 \end{cases}, \tag{6}$$

whereas at high dimension $d \geq n$, it is as for the Gaussian location model in (5), i.e., $\mathsf{M}_{\mathrm{GMM}}(n,d,t) = \mathsf{M}_{\mathrm{GLM}}(n,d,t)$. Hence, at this high dimensional regime, the loss does not vanishes by increasing the signal strength $t$, or by increasing the number of samples $n$. The loss in (6) is achieved by the trivial estimator $\hat{\theta} = 0$ if $t \leq \left(\frac{d}{n}\right)^{1/4}$ and by an estimator given by a properly scaled and shifted principal component of the empirical covariance matrix of $X_1^n := (X_1, X_2, \ldots, X_n)$, if $t \geq \left(\frac{d}{n}\right)^{1/4}$. For the GMM at low dimension, $d \leq n$, the global minimax rate is $(\frac{d}{n})^{1/4}$, which is worse than the minimax rate of the Gaussian location model $\sqrt{\frac{d}{n}}$.

Therefore, the GMM has worse estimation performance compared to the GLM from three aspects: First, at low dimension, $d \leq n$, it has a larger global minimax rate $(\frac{d}{n})^{1/4}$ compared to the parametric error rate of the GLM, $\sqrt{\frac{d}{n}}$; Second, at low dimension, $d \leq n$, parametric error rate is achieved only for constant separation $t \geq 1$; Third, the transition to the high dimension regime occurs at $d = n$.

As is intuitively appealing from a "data-processing" reasoning, a Markov model with flip probability $\delta'$ should allow for lower estimation error of $\theta_*$ compared to a Markov model with $\delta > \delta'$. Indeed, and as a specific simple example, any Markov model with $\delta < \frac{1}{2}$ can be easily transformed to a GMM model by randomizing the signs of each of the samples by an independent Rademacher variable. Thus we may deduce, e.g., that since at high dimension ($d \geq n$) the GLM and the GMM have the same minimax rates, the minimax rates for $d \geq n$ are in fact as in (5) for *any* $\delta \in [0, 1]$. We thus henceforth exclusively focus on the regime $d \leq n$. As we show, it is generally true that the improvement in estimation error when $\delta$ is reduced is less profound as the dimension increases.

## 1.3 Contributions

We first consider the case in which $\delta$ is known to the estimator of $\theta_*$. For this case, we analyze the loss of an estimator that is based on a computation of the principal component of a properly chosen empirical covariance matrix. We show (Theorem 1) that, at low dimension, $d \leq \delta n$, it achieves a local minimax rate of

$$\mathsf{M}(n,d,\delta,t) \lesssim \begin{cases} t, & t \leq \left(\frac{\delta d}{n}\right)^{1/4} \\ \frac{1}{t}\sqrt{\frac{\delta d}{n}}, & \left(\frac{\delta d}{n}\right)^{1/4} \leq t \leq \sqrt{\delta} \\ \sqrt{\frac{d}{n}}, & t \geq \sqrt{\delta} \end{cases}, \tag{7}$$

and at high dimension, $d \geq \delta n$, it is as for the Gaussian location model in (5). The rate of this estimator is then further shown to be asymptotically optimal (up to a logarithmic factor) via a minimax lower bound (Theorem 2). Evidently, the loss in (7) smoothly interpolates the rates of the GLM in (5) and the GMM in (6). Moreover, it is evident that the loss is improved with the decrease of $\delta$ from all three aspects previously mentioned. First, at low dimension, $d \leq \delta n$, the global minimax rate is

$\left(\frac{\delta d}{n}\right)^{1/4}$ (obtained by equating the first and second cases in (7)). Hence, reducing the flip probability from $\delta = \Theta(1)$ to $\delta = \frac{d}{n}$ smoothly reduces the global minimax rate from the parametric error rate $\Theta(\sqrt{\frac{d}{n}})$ of the GLM to the $\Theta((\frac{d}{n})^{1/4})$ of the GMM. Second, at low dimension, $d \leq \delta n$, the minimal signal strength required to obtain parametric error rate is $t = \Theta(\sqrt{\delta})$ (obtained from the third case in (7)). This, again, improves by decreasing $\delta$, and matches the extremes $t = \Theta(\sqrt{\frac{d}{n}})$ of GLM and $t = \Theta(1)$ of GMM. Third, the transition to the high dimension regime occurs at $d = \delta n$, which is again better with lower $\delta$, and matches the transition point of the GMM given by $d = n$. This lower transition point allows us to achieve the error rate of the GLM for any signal strength, even in a regime in which the loss $O(\sqrt{\frac{d}{n}})$ vanishes with $n \to \infty$ (unlike for GMM, $\delta = \frac{1}{2}$); specifically, this occurs whenever $d \geq \delta n$ yet $d = o(n)$. Beyond the formal proof, Appendix A provides a heuristic justification for why the minimax error is naturally expected to scale as in (7).

Second, as a step towards the removal of the assumption that $\delta$ is known to the estimator, we consider the complementary problem of estimating $\delta$ whenever an estimate $\theta_\sharp$ of $\theta_*$ is available (which can be either exact $\theta_\sharp = \theta_*$, or inaccurate $\theta_\sharp \neq \theta_*$). We propose a simple estimator for $\delta$, and analyze its error in case of a mismatch (Theorem 4). We then specify this result to the matched case $\theta_\sharp = \theta_*$ and show that in the non-trivial regime ($\|\theta_*\| \lesssim 1$) its error rate is $\tilde{O}(\frac{1}{\|\theta_*\|^2}\sqrt{\frac{1}{n}})$. We then proceed to show an impossibility lower bound of $\Omega(\sqrt{\frac{1}{n}})$ (Proposition 6) for this error rate. The precise dependence of the estimation error rate of $\delta$ on $\|\theta_*\|$ is therefore not precisely determined, and we discuss the challenges in settling this matter.

Third, we consider the case in which the estimator of $\theta_*$ has no prior knowledge of $\delta$. We propose a three-step algorithm for this case (Algorithm 1). First, a (possibly) gross estimate $\hat{\theta}^{(A)}$ of $\theta_*$ is computed based on third of the samples, assuming the worst case of $\delta = \frac{1}{2}$. Then, an estimate $\hat{\delta}^{(B)}$ of $\delta$ is computed using another third of the samples, assuming the estimate $\hat{\theta}^{(A)}$. Finally, a refined estimate $\hat{\theta}^{(C)}$ of $\theta_*$ is obtained by (essentially) assuming that $\delta$ is $\hat{\delta}^{(B)}$. At each of the steps above, the algorithm may stop and decide to return its current estimate when it determines that no further improvement is possible by moving on to the next steps. We analyze the loss of this algorithm (Theorem 7), and show that this algorithm is capable of partially achieving the gains associated with the case of known $\delta$.

Our technique of partitioning the samples to blocks according to the dependence structure of the Markov chain renders the possibility of understanding the minimax rate of more general models, such as mixtures with multiple components [Doss et al., 2020] and memory, Ising models [Daskalakis et al., 2019], Boltzmann machines [Bresler et al., 2019], Markov random fields, etc. Our findings in the mean estimation setting also stand in contrast to the *data wastage* phenomenon observed in the linear regression setting in the prior work Bresler et al. [2020].

## 1.4 Additional related work

Both the GLM [Johnstone, 2002, Tsybakov, 2008] and GMM [Lindsay, 1995, McLachlan et al., 2019] are classic models which were well-explored from numerous perspectives. In the last few years, there is a surge of interest in the non-asymptotic performance analysis of computationally efficient estimation algorithms for this estimation task. For example, Moitra and Valiant [2010], Kalai et al. [2010], Anandkumar et al. [2014], Hardt and Price [2015], Wu and Yang [2020] have analyzed method-of-moments-based algorithms, and various other papers considered the EM algorithm [Balakrishnan et al., 2017, Xu et al., 2016, Jin et al., 2016, Klusowski and Brinda, 2016, Weinberger and Bresler, 2022, Dwivedi et al., 2020b,a, 2018, Zhao et al., 2018, Yan et al., 2017]. Specifically, the local minimax rate for GMM in (6) was determined in [Wu and Zhou, 2019] as a benchmark for the operation of the EM algorithm.

The model (2) is a simple instance of a HMM [Ephraim and Merhav, 2002, van Handel, 2008] in high dimensions. Parameter estimation in such models is practically performed via the Baum-Welch algorithm [Baum et al., 1970], which is a computationally efficient version of EM for HMMs. To the best of our knowledge, there were hardly any attempts to characterize the minimax rates in such models, with the exception of [Aiylam, 2018] and [Yang et al., 2015], previously mentioned. In

[Aiylam, 2018], a local version of the Baum-Welch algorithm was proposed, and vanishing error of the convergence of the estimate to the true parameter was established for both a population version as well as a finite-sample version. In [Yang et al., 2015], general bounds on the performance of Baum-Welch algorithm were specified to the Gaussian model with Markov signs (2) considered here. It was shown that the Baum-Welch algorithm achieves a parametric error rate, and converges in a finite number of iterations [Yang et al., 2015, Corollary 2]. However, the qualifying condition for the estimation error bound of [Yang et al., 2015] is that $t = \|\theta_*\| \gtrsim \log \frac{1}{1-(1-2\delta)^2}$, that is, a non-trivial separation when $\delta$ is constant, which further blows up as $\delta \downarrow 0$. By contrast, in this paper, our goal is to characterize the estimation error in the regime of $\delta$ and $t = \|\theta_*\|$ in which the minimax rate is affected by these parameters, and this requires analyzing vanishing $\delta$ and $t$.

More broadly, there is a growing interest in advancing the quantitative understanding of the performance of statistical learning and inference with dependent data. Bresler et al. [2020] studied linear regression with Markovian covariates and characterized the minimax error rate in terms of the mixing time of the Markov chain. A stochastic gradient descent-style algorithm adapted to the Markov setting was shown to be minimax optimal. Statistical estimation problems including linear and logistic regression with more general network dependencies among response variables were studied by Daskalakis et al. [2019] and Kandiros et al. [2021]. Learnability and generalization bounds were derived by Dagan et al. [2019] for dependent data satisfying the so-called Dobrushin's condition.

## 1.5 Notation conventions

For a vector $v \in \mathbb{R}^d$, $\|v\|$ is the Euclidean norm. For a positivedefinite matrix $A$, $\lambda_{\max}(A)$ and $v_{\max}(A)$ are the maximal eigenvalue and the associated eigenvector (of unit norm) of $A$. Unless otherwise stated, the constants involved in Bachmann-Landau notation are numerical, and do not depend on the parameters $(n, d, \delta, t)$. It holds that $a \gtrsim b$ (resp. $a \lesssim b$) if there exists a constant $c > 0$ (resp. $C > 0$) such that $a \geq cb$ (resp. $a \leq Cb$). If $a \lesssim b$ and $a \gtrsim b$. then $a \asymp b$. Integer constraints (ceiling and floor) on large quantities that do not affect the results are omitted for brevity. For a real numbers $a, b$ the shorthand notation $a \vee b := \max\{a, b\}, a \wedge b := \min\{a, b\}$ and $(a)_+ := a \vee 0$ is used. A sequence of objects $X_1, \cdots, X_n$ is denoted by $X_1^n$. Expectation, variance and probability are denoted by $\mathbb{E}, \mathbb{V}$ and $\mathbb{P}$, respectively. Equality in distribution of random variables $X$ and $Y$ is denoted by $X \overset{d}{=} Y$. All logarithms $\log$ are to the base $e$.

## 2 Mean estimation for a known flip probability

In this section, we consider the problem of estimating $\theta_*$ whenever $\delta$ is exactly known to the estimator. In that case, it may be assumed w.l.o.g. that $\delta \in [0, \frac{1}{2}]$, as otherwise one may negate each of the even samples to obtain an equivalent model with $\delta$ replaced with $1 - \delta$. Hence also $\rho \in [0, 1]$. We next describe an estimator for this task, state a bound on its performance, and then show that it matches (up to a logarithmic factor) an impossibility lower bound.

The estimator operationally interpolates and therefore simultaneously generalizes the empirical average estimator (30) and the (properly scaled) principal component estimator (32) analyzed in [Wu and Zhou, 2019, Appendix B]. It degenerates to the latter estimators if $\delta \downarrow 0$ or $\delta \uparrow \frac{1}{2}$. Specifically, the estimator partitions the sample into blocks of equal length $k$ each (which will later be set to $k = \frac{1}{8\delta}$, according to the mixing time of the Markov chain $S_0^n$). Let $\ell$ denote the number of blocks respectively, so that $k\ell = n$. Let $\mathcal{I}_i = \{(i-1)k + 1, (i-1)k + 2, \cdots, ik\}$ denote the indices of the $i$th block. Further, let $\{R_i\}_{i \in [\ell]}$ be an i.i.d. Rademacher sequence ($R_i \sim \text{Uniform}\{-1, 1\}$), and let

$$\overline{X}_i := R_i \cdot \frac{1}{k} \sum_{j \in \mathcal{I}_i} X_j = \overline{S}_i \theta_* + \overline{Z}_i \tag{8}$$

denote the average of the samples in the $i$th block (randomized with a sign $R_i$), where

$$\overline{S}_i := R_i \cdot \frac{1}{k} \sum_{j \in \mathcal{I}_i} S_j \tag{9}$$

is the *gain* (average of the signs) of the $i$th block, and

$$\overline{Z}_i := R_i \cdot \frac{1}{k} \sum_{j \in \mathcal{I}_i} Z_j \sim N\left(0, \frac{1}{k} \cdot I_d\right) \tag{10}$$

is the average noise of the $i$th block. Due to the sign randomization, it holds that $\{\overline{S}_i\}_{i \in [\ell]}$ is an i.i.d. sequence. Since $\{\overline{Z}_i\}_{i \in [\ell]}$ is also an i.i.d. sequence, then so is $\{\overline{X}_i\}_{i \in [\ell]}$. For notational simplicity we will omit the block index $i$ of a generic block. For block length $k$, we denote by

$$\xi_k := \mathbb{E}[\overline{S}^2] = \mathbb{E}\left[\left(\frac{1}{k}\sum_{j=1}^{k} S_j\right)^2\right] \tag{11}$$

the second moment of the gain $\overline{S}$. Note that $\xi_k \in [\frac{1}{k}, 1]$ for any $\delta \in [0, \frac{1}{2}]$, and, in particular, it is always positive. For a sequence of samples $X_1^n = (X_1, \ldots, X_n)$, we define by $\hat{\Sigma}_{n,k}(X_1^n)$ the empirical covariance matrix of the averaged samples over blocks $\{\overline{X}_i\}_{i \in [\ell]}$, that is

$$\hat{\Sigma}_{n,k}(X_1^n) := \frac{1}{\ell}\sum_{i=1}^{\ell} \overline{X}_i \overline{X}_i^\top, \tag{12}$$

whose population average is $\Sigma_{n,k}(\theta_*)$, where

$$\Sigma_{n,k}(\theta_*) := \mathbb{E}[\overline{X}\,\overline{X}^\top] = \xi_k \theta_* \theta_*^\top + \frac{1}{k}I_d. \tag{13}$$

We note that $\theta$ is the principal component of $\Sigma_{n,k}(\theta)$, that is, $\lambda_{\max}(\Sigma_{n,k}(\theta)) = \xi_k\|\theta\|^2 + \frac{1}{k}$ and the corresponding eigenvector is $v_{\max}(\Sigma_{n,k}(\theta)) = \theta$. We thus consider the following estimator for $\theta_*$, from a sequence $X_1^n$, and with a block length of $k$

$$\hat{\theta}_{\text{cov}}(X_1^n; k) := \sqrt{\frac{1}{\xi_k}\left(\lambda_{\max}(\hat{\Sigma}_{n,k}(X_1^n)) - \frac{1}{k}\right)_+} \cdot v_{\max}\left(\hat{\Sigma}_{n,k}(X_1^n)\right). \tag{14}$$

The estimator is thus constructed from two types of averages: First, a coherent average of the samples at each block, to obtain $\ell$ block-samples $\overline{X}_i$ with gain $\overline{S}_i$ and noise variance reduced by a factor of $k$. Second, an incoherent average of the "square" of the $\ell$ block-samples $\overline{X}_i \overline{X}_i^\top$, which resolves the remaining sign ambiguity between blocks. This balance two extreme cases: If $\delta = 0$, then this reduces the problem to the GLM (with a sign ambiguity) and $k = n$ is an optimal choice. If $\delta = \frac{1}{2}$, then this reduces the problem to the GMM, in which coherent averaging is non-beneficial and $k = 1$ is rate optimal. Generally, the optimal choice of the block length $k$ is proportional to the mixing time of the Markov chain $\Theta(\frac{1}{\delta})$. This choice assures that the random gain $\overline{S}$ is $\pm 1$ with a (constant) high probability. In fact, an elementary, yet crucial, part of the analysis establishes that the random gain $\overline{S}$ has constant variance for this choice of block length (see Lemma 9 in Appendix B.1). On the other hand, if $k = \Omega(\frac{1}{\delta})$, then the random gain $\overline{S}$ will not be $\pm 1$ (or not even bounded away from zero) with high probability, and such choice is never efficient. Specifically, we consider the estimator in (14) with $k = \frac{1}{8\delta}$. The above estimation procedure is depicted in Figure 2 in Appendix B.1.

Let us denote

$$\beta(n, d, \delta) := \sqrt{\frac{d}{n}} \vee \left(\frac{\delta d}{n}\right)^{1/4}, \tag{15}$$

which will actually be the global minimax rate.

**Theorem 1.** *Assume that $\delta \geq \frac{1}{n}$ and $d \leq n$, and set $\hat{\theta} \equiv \hat{\theta}_{\text{cov}}(X_1^n; k)$ with $k = \frac{1}{8\delta}$. Then, there exist numerical constants $c_0, c_1, c_2 > 0$ such that for every $\theta_* \in \mathbb{R}^d$*

$$\mathbb{E}\left[\text{loss}(\hat{\theta}, \theta_*)\right] \leq c_0 \cdot \begin{cases} \beta(n, d, \delta), & \|\theta_*\| \leq \beta(n, d, \delta) \\ \sqrt{\frac{d}{n}} + \frac{1}{\|\theta_*\|}\sqrt{\frac{\delta d}{n}} + \frac{1}{\|\theta_*\|} \cdot \frac{d}{n}, & \beta(n, d, \delta) \leq \|\theta_*\| \end{cases} \tag{16}$$

*and*

$$\text{loss}(\hat{\theta}, \theta_*) \leq c_1 \cdot \log(n) \cdot \mathbb{E}\left[\text{loss}(\hat{\theta}, \theta_*)\right] \tag{17}$$

*with probability larger than $1 - \frac{c_2}{n}$.*

Theorem 1 is proved in Appendix B.1. Evidently, Theorem 1 implies that the upper bound on the minimax rate stated in (7) above holds in low dimension, $d \leq \delta n$, whereas $\mathsf{M}(n,d,\delta,t) \asymp \mathsf{M}_{\mathrm{GLM}}(n,d,t)$ holds in high dimension $d \geq \delta n$.[2] We remark that the condition $\delta \geq \frac{1}{n}$ is mild as otherwise the model (2) is essentially equivalent to GLM. See Remark 8 in Appendix B.1. Numerical validation of the performance of the estimator $\hat{\theta}_{\mathrm{cov}}(X_1^n; k)$ is shown in Appendix F.

We next consider an impossibility result. As we have seen, at high dimension, $d \geq \delta n$, the minimax error rates achieved are the same as for the Gaussian location model, and thus clearly cannot be improved. We thus next focus on the low dimensional regime $d \leq \delta n$.

**Theorem 2.** *Assume that $2 \leq d \leq \delta n$ and $n \geq \frac{128}{d}$. Then the local minimax rate is bounded as*

$$\mathsf{M}(n,d,\delta,t) \gtrsim \frac{1}{\sqrt{\log(n)}} \cdot \begin{cases} t, & t \leq \left(\frac{\delta d}{n}\right)^{1/4} \\ \frac{1}{t}\sqrt{\frac{\delta d}{n}}, & \left(\frac{\delta d}{n}\right)^{1/4} \leq t \leq \sqrt{\delta} \\ \sqrt{\frac{d}{n}}, & t \geq \sqrt{\delta} \end{cases} . \tag{18}$$

Hence, the minimax rates achieved by the estimator in Theorem 1 are nearly asymptotically optimal, up to a $\sqrt{\log(n)}$ factor. The full proof of Theorem 2 together with a summary of the main ideas used in the proof is presented in Appendix B.2.

*Remark* 3 (Relaxation of the noise distribution assumption). For the sake of clarity of exposition, we have assumed in Theorems 1 and 2 that the noise samples $\{Z_i\}_{i=1}^n$ are i.i.d. isotropic Gaussians. There are two straightforward relaxations of this assumption. First, our minimax upper bound (Theorem 1) can be proved to any subGaussian noise distribution, simply because all the concentration bounds for Gaussian random variables used in the proof admit subGaussian analogues. The impossibility result (converse, Theorem 2) trivially holds for subGaussian noise since Gaussians are special case of subGaussians. Second, the isotropic assumption can be relaxed to anisotropic noise with known covariance $\Sigma$ by simple standardization: If $Z_i \sim N(0_d, \Sigma)$ are i.i.d. for some *known* $\Sigma \succ 0$, then the estimator will multiply the samples by $\Sigma^{-1/2}$ and reduce the problem back to the isotropic setting. After applying the estimator we propose for the isotropic case, the estimator will obtain its final estimate by multiplying its isotropic estimate by $\Sigma^{1/2}$. The loss of the estimator will then be gauged by the Mahalanobis distance, parameterized by $\Sigma$. We refer the reader to Appendix G for a discussion on more challenging directions in which the isotropic Gaussian assumption can be relaxed.

## 3 Flip probability estimation for a given estimator of $\theta_*$

In this section, we consider the problem of estimating $\delta$ whenever $\theta_*$ is approximately known to be $\theta_\sharp$. We propose a simple estimator, and then discuss the importance of the accuracy of $\theta_*$. We then derive an impossibility result for the matched case, $\theta_\sharp = \theta_*$.

First note that an estimator for $\delta$ can be easily obtained from an estimator for $\rho$, with essentially the same error rate, via $\hat{\delta} = \frac{1}{2}(1 - \hat{\rho})$. Thus we focus on estimating $\rho$. Assume for simplicity that $n$ is even. Observing that $\mathbb{E}[X_{2i}^\top X_{2i+1}] = \rho \|\theta_*\|^2$, we propose the following natural estimator for $\rho$, which replaces the population average with empirical average:

$$\hat{\rho}_{\mathrm{corr}}(X_1^n; \theta_\sharp) = \frac{1}{\|\theta_\sharp\|^2} \cdot \frac{2}{n} \sum_{i=1}^{n/2} X_{2i}^\top X_{2i-1}. \tag{19}$$

That is, the estimator is based on evaluating the correlation of each of two adjacent samples $X_{2i}$ and $X_{2i-1}$. We first state a general bound on the estimation error of this estimator. We then consider the case in which $\theta_*$ is known, and show how the estimation error is improved in this case.

---

[2]When $\|\theta_*\| \leq \beta(n,d,\delta)$, the estimator $\hat{\theta}_{\mathrm{cov}}(X_1^n; k)$ in Theorem 1 only achieves a rate $\beta(n,d,\delta)$ which is larger than the promised rate $\|\theta_*\|$ in (7). However, since the estimator is assumed to know $t$ (but not the direction of $\theta_*$; a common formulation in high-dimensional statistics), then it can output the zero vector. It then incurs loss $\|\theta\|_*$ for any $\theta_* \in \mathbb{R}^d$, matching the promised rate when $\|\theta_*\| \leq \beta(n,d,\delta)$. To summarize, for any value of $t$, the minimax rate is achieved by the minimum rate of $\hat{\theta}_{\mathrm{cov}}(X_1^n; k)$ and $\hat{\theta}_0(X_1^n) \equiv 0$.

**Theorem 4.** *Assume that $d \leq n$. Let $\theta_* \in \mathbb{R}^d$ and let $\theta_\sharp$ be an estimate of $\theta_*$. Set $\hat{\rho} \equiv \hat{\rho}_{\text{corr}}(X_1^n; \theta_\sharp)$ and $\hat{\delta} = \frac{1}{2}(1 - \hat{\rho})$. Then, it holds with probability $1 - \frac{8}{n}$ that*

$$\left| \hat{\delta} - \delta \right| = \frac{1}{2} |\hat{\rho} - \rho| \leq \frac{\left| \|\theta_*\|^2 - \|\theta_\sharp\|^2 \right|}{\|\theta_\sharp\|^2} + 16 \log(n) \left[ \sqrt{\frac{\delta}{n}} + \frac{1}{\|\theta_\sharp\|} \sqrt{\frac{1}{n}} + \frac{1}{\|\theta_\sharp\|^2} \sqrt{\frac{d}{n}} \right]. \quad (20)$$

The proof of Theorem 4 appears in Appendix C.1. Note that Theorem 4 states a high-probability bound, suitable to its usage later on in Section 4. A bound on the expectation of the error can be obtained by the standard method of integrating tails.

**The effect of knowledge of $\theta_*$**   If $\theta_*$ is known up to a sign, i.e., $\theta_\sharp = \pm\theta_*$, then for the purpose of $\rho$ (or equivalently $\delta$) estimation, the model (2) can be reduced to a one-dimensional model by rotational invariance of isotropic Gaussian (See additional details in Appendix C.2). It then immediately follows from Theorem 4 that:

**Corollary 5.** *Assume that $d \leq n$, $\|\theta_*\| \leq 1$ and $\theta_\sharp = \pm\theta_*$. Let $U_1^n$ be defined in as $U_i := \|\theta_*\| \cdot S_i + W_i$ where $W_i \sim N(0,1)$ i.i.d., $\hat{\rho} \equiv \hat{\rho}_{\text{corr}}(U_1^n; \theta_\sharp)$ and $\hat{\delta} = \frac{1}{2}(1 - \hat{\rho})$. Then it holds with probability $1 - \frac{8}{n}$ that*

$$\left| \hat{\delta} - \delta \right| = \frac{1}{2} |\hat{\rho} - \rho| \leq \frac{18 \log(n)}{\|\theta_*\|^2} \sqrt{\frac{1}{n}}. \quad (21)$$

Numerical validation of the performance of the estimator $\hat{\delta}_{\text{corr}}(X_1^n; \theta_\sharp) = \frac{1}{2}(1 - \hat{\rho}_{\text{corr}}(X_1^n; \theta_\sharp))$ in the mismatched (Theorem 4) and matched (Corollary 5) cases is provided in Appendix F.

We next consider an impossibility lower bound.

**Proposition 6.** *Suppose that $\theta_\sharp = \theta_*$ and $\|\theta_*\| \leq \frac{1}{\sqrt{2}}$. Then*

$$\inf_{\hat{\delta}(U_1^n)} \sup_{\delta \in [0,1]} \mathbb{E}[|\delta - \hat{\delta}(U_1^n)|] \geq \frac{1}{32\sqrt{n}}, \quad (22)$$

*where the infimum is over any estimator $\hat{\delta}(U_1^n)$ based on the model $U_i = \|\theta_*\| S_i + W_i$ where each $W_i$ is i.i.d. $N(0,1)$.*

The proof of Proposition 6 is presented in Appendix C.2. According to Corollary 5 and Proposition 6, in estimating $\delta$ with a known $\theta_*$, though the dependence $\Theta(\frac{1}{\sqrt{n}})$ of the minimax error rate on the sample size is shown to be nearly optimal, it is unclear what the optimal dependence on the signal strength should be. This is left as an interesting open question and we discuss the challenges associated with this problem in Appendix C.2.

## 4   Mean estimation under an unknown flip probability

As we have seen, if an estimator for $\theta_*$ knows the value of $\delta$, and if both $\delta \leq \frac{1}{2}$ and $d \leq \delta n$ hold, then the estimator can achieve improved error rates over the GMM case ($\delta = \frac{1}{2}$). In this section, we assume that both $\theta_*$ and $\delta$ are unknown, and so the estimator is required to estimate $\delta$ in order to use this knowledge for an estimator of $\theta_*$. We propose an estimation procedure of three steps based on sample splitting of $3n$ samples. We mention at the outset that the regime in which improvement is possible will be for low signal strength $\|\theta_*\| \lesssim 1$ (low separation between the components), and up to a dimension which depends on $\delta$. Of course the estimation procedure does not know $(\theta_*, \delta)$ in advance, and so it is required to identify if $(\theta_*, \delta)$ are in this regime during its operation.

We now begin with an overview of the steps of the estimation algorithm. At Step A, the algorithm estimates $\theta_*$ based on $X_1^n$ assuming a Gaussian mixture model ($\delta = \frac{1}{2}$) to obtain an estimate $\hat{\theta}^{(A)}$. Then, based on $\|\hat{\theta}^{(A)}\|$, the algorithm decides whether improvement is potentially possible had $\delta$ was known. There are two cases. The first case is that $\|\hat{\theta}^{(A)}\|$ is too low, and then its estimate is not sufficiently accurate to be used in the next steps. Essentially, this happens when the norm is below the global minimax rate $(\frac{d}{n})^{1/4}$, and the estimation error of the norm on the same scale as the norm of $\|\theta_*\|$. A trivial estimator of $\hat{\theta} = 0$ is then optimal in terms of error rates. It can be already noted

at this step that while the global minimax rate for the known $\delta$ case is $(\frac{\delta d}{n})^{1/4}$, here the algorithm already stops and estimates $\hat{\theta} = 0$ even if just $\|\theta_*\| \lesssim (\frac{d}{n})^{1/4}$, leading to larger global minimax rate. The second case is that $\|\theta_*\|$ is larger than a constant. In this case, the estimation based on a GMM already achieves the optimal parametric $O(\sqrt{\frac{d}{n}})$ error rate of the Gaussian location model, and so no further estimation steps are necessary. Otherwise, an improvement in the estimation is possible. The algorithm proceeds to Step B, and uses $X_{n+1}^{2n}$ to obtain an estimate $\hat{\delta}^{(B)}$ of $\delta$ based on the mismatched $\theta_\sharp \equiv \hat{\theta}^{(A)}$. Then, based on the estimate $\hat{\delta}^{(B)}$ the algorithm decides whether the accuracy of $\hat{\delta}^{(B)}$ is sufficient to be used in an refined estimation of $\theta_*$. If the accuracy of $\hat{\delta}^{(B)}$ is not good enough, then the algorithm outputs the estimate from Step A, that is $\hat{\theta}^{(A)}$. Otherwise, it proceeds to Step C, in which $\theta_*$ is re-estimated using $X_{2n+1}^{3n}$, based on a mismatched choice of $k$, that is $k \asymp \frac{1}{\hat{\delta}^{(B)}}$ instead of $k = \frac{1}{8\delta}$. Intuitively, the estimated value $\hat{\delta}^{(B)}$ should be larger than $\delta$ so the resulting block size $k \asymp \frac{1}{\hat{\delta}^{(B)}}$ will be such that the gain in the block is still close to 1 with high probability. On the other hand, it is desired that $\hat{\delta}^{(B)}$ will be on the same scale as $\delta$ so that the estimation rate (304) (see also (7)) – which now essentially holds with $\hat{\delta}^{(B)}$ instead of $\delta$ – would be as small as possible. Thus, if the algorithm has assured in Step B that $\hat{\delta}^{(B)} \asymp \delta$, then at Step C it will achieve the error rate indicated in (304).

---

**Algorithm 1** Mean estimation for unknown $\delta$

---

1: **input:** Parameters $\lambda_\theta, \lambda_\delta > 0$ (from (304) (305) (307)), $3n$ data samples $X_1^{3n}$ from the model (2)

2: **step A:** Estimate $\theta_*$ assuming a Gaussian mixture model:

$$\hat{\theta}^{(A)} \equiv \hat{\theta}_{\text{cov}}(X_1^n; k = 1) \tag{23}$$

3: **if** $\|\hat{\theta}^{(A)}\| \leq 2\lambda_\theta \cdot \log(n) \cdot (\frac{d}{n})^{1/4}$ **then**

4:    **return** $\hat{\theta} = 0$                    $\triangleright$ No further improvement can be guaranteed

5: **else if** $\|\hat{\theta}^{(A)}\| \geq \frac{1}{2}$ **then**

6:    **return** $\hat{\theta} = \hat{\theta}^{(A)}$                    $\triangleright$ No further improvement is possible

7: **end if**

8: **step B:** Estimate $\delta$ assuming a mismatched mean value $\hat{\theta}^{(A)}$:

$$\hat{\delta}^{(B)} = \hat{\delta}_{\text{corr}}(X_{n+1}^{2n}; \hat{\theta}^{(A)}) \tag{24}$$

9: **if** $\hat{\delta}^{(B)} \leq 64\lambda_\delta\lambda_\theta \frac{\log(n)}{\|\hat{\theta}^{(A)}\|^2}\sqrt{\frac{d}{n}}$ **then**

10:    **return** $\hat{\theta} = \hat{\theta}^{(A)}$                    $\triangleright$ No further improvement can be guaranteed

11: **end if**

12: **step C**: Estimate $\theta_*$ assuming a mismatched flip probability $\hat{\delta}^{(B)}$:

$$\hat{\theta}^{(C)} = \hat{\theta}_{\text{cov}}\left(X_{2n+1}^{3n}; k = \frac{1}{16\hat{\delta}^{(B)}}\right) \tag{25}$$

13: **return** $\hat{\theta} = \hat{\theta}^{(C)}$.

---

The formal description of the estimation algorithm is provided in Algorithm 1. We remark that refining the estimation of $\delta$ can be easily incorporated as a fourth step of this algorithm, but we do not present this in order to keep the statement of the result simple. The error of the estimator output by Algorithm 1 is as follows:

**Theorem 7.** *There exist constants $c_1, c_2 \geq 0$ and $\lambda_\theta, \lambda_\delta \geq 1$ such that if $d \leq \frac{n}{4\lambda_\theta^2 \log^2(n) \wedge 16}$ then the output $\hat{\theta}$ of Algorithm 1 satisfies for any $\theta_* \in \mathbb{R}^d$, with probability $1 - O(\frac{1}{n})$:*

*If $d \leq \frac{1}{64\lambda_\delta^2 \lambda_\theta^2 \log^2(n)} \delta^4 n$ then*

$$\text{loss}(\hat{\theta}, \theta_*) \leq c_1 \log n \cdot \begin{cases} \|\theta_*\|, & \|\theta_*\| \leq \lambda_\theta \log(n) \left(\frac{d}{n}\right)^{1/4} \\ \frac{1}{\|\theta_*\|} \sqrt{\frac{d}{n}}, & \lambda_\theta \log(n) \left(\frac{d}{n}\right)^{1/4} \leq \|\theta_*\| \leq \sqrt{8\lambda_\delta \lambda_\theta \log(n)} \left(\frac{d}{\delta^2 n}\right)^{1/4} \\ \frac{1}{\|\theta_*\|} \sqrt{\frac{\delta d}{n}}, & \sqrt{8\lambda_\delta \lambda_\theta \log(n)} \left(\frac{d}{\delta^2 n}\right)^{1/4} \leq \|\theta_*\| \leq \sqrt{\delta} \\ \sqrt{\frac{d}{n}}, & \|\theta_*\| \geq \sqrt{\delta} \end{cases} ; \quad (26)$$

*If $d \geq \frac{1}{64\lambda_\delta^2 \lambda_\theta^2 \log^2(n)} \delta^4 n$ then*

$$\text{loss}(\hat{\theta}, \theta_*) \leq c_2 \log n \cdot \begin{cases} \|\theta_*\|, & \|\theta_*\| \leq \lambda_\theta \log(n) \left(\frac{d}{n}\right)^{1/4} \\ \frac{1}{\|\theta_*\|} \sqrt{\frac{d}{n}}, & \lambda_\theta \log(n) \left(\frac{d}{n}\right)^{1/4} \leq \|\theta_*\| \leq \sqrt{8\lambda_\delta \lambda_\theta \log(n)} \left(\frac{d}{\delta^2 n}\right)^{1/4} \\ \sqrt{\frac{d}{n}}, & \|\theta_*\| \geq \sqrt{8\lambda_\delta \lambda_\theta \log(n)} \left(\frac{d}{\delta^2 n}\right)^{1/4} \end{cases} . \quad (27)$$

Theorem 7 implies that up to logarithmic factors, the error rates of the known $\delta$ case are recovered for low enough dimension $d \lesssim \delta^4 n$ and $\|\theta_*\| \gtrsim \left(\frac{d}{\delta^2 n}\right)^{1/4}$. The analysis of Algorithm 1 appears in Appendix D. Numerical validation of the performance of Algorithm 1 can be found in Appendix F.

**The impact of unknown $\delta$ on the estimation error of $\theta_*$**   Comparing Theorems 2 (known $\delta$) and 7 (unknown $\delta$) reveals the deterioration in the estimation of $\theta_*$ due to the lack of knowledge of $\delta$ from the three aspects mentioned in Section 1.3 (ignoring logarithmic factors): First, the global minimax rate is $O(\frac{d}{n})^{1/4}$ as for the GMM, instead of the rate $O(\frac{\delta d}{n})^{1/4}$ for the Markov model case with known $\delta$. Second, at the regime $\left(\frac{d}{n}\right)^{1/4} \lesssim \|\theta_*\| \lesssim \left(\frac{d}{\delta^2 n}\right)^{1/4}$ the error rate is $O(\frac{1}{\|\theta_*\|}\sqrt{\frac{d}{n}})$ instead of the lower $O(\frac{1}{\|\theta_*\|}\sqrt{\frac{\delta d}{n}})$. Third, the algorithm is only effective when the dimension is as low as $d \lesssim \delta^4 n$. For higher dimensions, the rates of the GMM are achieved, which can be achieved even without the knowledge of $\delta$.

## 5   Conclusion and future work

In this paper, we have considered an elementary, yet fundamental, high-dimensional model with memory. We have obtained a sharp bound on the minimax rate of estimation in case the underlying statistical dependency (flip probability) is known, and proposed a three-step estimation algorithm when it is unknown. This has revealed the gains possible in estimation rates due to the memory between the samples, and smoothly interpolated between the extreme cases of GLM and GMM. An interesting open problem is to either characterize the optimality of the algorithm or improving in the unknown $\delta$ case, which requires understanding optimal estimation of the flip probability.

Naturally, as the model considered in this paper is basic, there is an ample of possibilities to generalize this model. These include, a larger number of components in the mixture, statistical dependency with a more complicated graphical structure between the data samples, existence of nuisance parameters such as the noise variance, sharp finite-sample/finite-iteration analysis of specific practical algorithms such as Baum-Welch, location-scale model with anisotropic noise, heavy-tailed noise, and so on.

## Acknowledgments and Disclosure of Funding

Part of this work was done when YZ was a postdoc at Technion where he received funding from the European Union's Horizon 2020 research and innovation programme under grant agreement No 682203-ERC-[Inf-Speed-Tradeoff]. The work of of NW was supported in part by the Israel Science Foundation (ISF) under Grant 1782/22. NW is grateful to Guy Bresler for introducing him to this problem, for the initial ideas that led to this research, and for many helpful discussions on the topic.

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
