In Appendix A we provide heuristic justification for the scaling of the optimal minimax error rate in (7). In Appendix B we provide the proofs for Theorems 1 and 2. In Appendix C we provide the proofs for Theorem 4 and Proposition 6. In Appendix D we provide the proofs for Theorem 7. In Appendix E we include some useful results for the sake of completeness.

**Additional notation** For a matrix $A \in \mathbb{R}^{d_1 \times d_2}$, $\|A\|_{\mathrm{op}}$ is the operator norm (with respect to Euclidean norms), and $\|A\|_F$ is the Frobenius norm of $A$. $\mathbb{S}^{d-1} := \{\theta \in \mathbb{R}^d : \|\theta\| = 1\}$ is the unit sphere and $\mathbb{B}^d := \{\theta \in \mathbb{R}^d : \|\theta\| \leq 1\}$ is the unit ball in the $d$-dimensional Euclidean space. For a pair of probability distributions $P$ and $Q$ on a common alphabet $\mathcal{X}$ with densities $p$ and $q$ w.r.t. to a base measure $\nu$, we denote the total variation distance by $\mathrm{d}_{\mathrm{TV}}(P, Q) := \frac{1}{2} \int |p - q| \mathrm{d}\nu$, the Kullback-Leibler (KL) divergence by $\mathrm{D}_{\mathrm{KL}}(P \,\|\, Q) := \int p \log \frac{p}{q} \mathrm{d}\nu$, and the chi-square divergence by $\chi^2(P \,\|\, Q) := \int \frac{(p-q)^2}{q} \mathrm{d}\nu = \int \frac{p^2}{q} \mathrm{d}\nu - 1$.

## A Heuristic justification of minimax rate (7)

The main intuition behind the HMM considered in this paper comes from the correlation decay phenomenon in graphical model. Indeed, one can view the Markov chain $X_1 \to X_2 \to \cdots \to X_n$ sampled from the model (2) as a simple graphical model whose conditional independence structure is expressed by a line graph on $n$ nodes. Each node $X_i$ is conditionally independent of other samples given its neighbors $X_{i-1}$ and $X_{i+1}$. Furthermore, a pair of nodes $X_i$ and $X_j$ of large graph distance (i.e., $|i - j|$) is approximately independent. Informally, we expect that there is one sign flip (i.e., $S_i = -S_{i+1}$) per $\approx \frac{1}{\delta}$ samples. Therefore, signs $S_i$ and $S_j$ are likely to have the same value if $|i - j| \lesssim \frac{1}{\delta}$ and are approximately independent if $|i - j| \gtrsim \frac{1}{\delta}$. This is the leading guideline for the design of our estimator (14) for upper bound and the reduction to the genie-aided model (147) for lower bound.

Recall that for the Gaussian location model

$$X = \theta_* + Z \tag{28}$$

with $Z \sim N(0, I_d)$, the minimax rate is given by

$$\mathsf{M}_{\mathrm{GLM}}(n, d, t) \asymp t \wedge \sqrt{\frac{d}{n}}. \tag{29}$$

Note that this is a compact way of writing the rate (5). Given $n$ i.i.d. samples $X_1^n$ from model (28), the estimator

$$\hat{\theta} = \frac{1}{n} \sum_{i=1}^n X_i \tag{30}$$

achieves the rate (29). For the Gaussian mixture model $X = S\theta_* + Z$ with $S \sim \mathrm{Unif}\{-1, 1\}$ and $Z \sim N(0, I_d)$, the minimax rate (6) [Wu and Zhou, 2019, Appendix B] can be compactly written as

$$\mathsf{M}_{\mathrm{GMM}}(n, d, t) \asymp \left[ \frac{1}{t} \left( \sqrt{\frac{d}{n}} + \frac{d}{n} \right) + \sqrt{\frac{d}{n}} \right] \wedge t. \tag{31}$$

The above rate is attained by the estimator

$$\hat{\theta} = \sqrt{(\lambda_{\max}(\hat{\Sigma}) - 1)_+} \cdot v_{\max}(\hat{\Sigma}) \tag{32}$$

where $\hat{\Sigma} = \frac{1}{n} \sum_{i=1}^n X_i X_i^\top$. More generally, we can also consider the GMM

$$\tilde{X} = S\theta_* + \tilde{Z} \tag{33}$$

with $S \sim \mathrm{Unif}\{-1, 1\}$ and $\tilde{Z} \sim N(0, \sigma^2 \cdot I_d)$, for some $\sigma > 0$, which is equivalent in distribution to $\tilde{X} = \sigma(S\theta_*/\sigma + Z)$ where $Z \sim N(0, I_d)$. Generalizing (31), it is straightforward to check that the optimal minimax rate for model (33) is given by

$$\tilde{\mathsf{M}}_{\mathrm{GMM}}(n, d, t, \sigma) \asymp \sigma \left[ \left( \frac{1}{t/\sigma} \left( \sqrt{\frac{d}{n}} + \frac{d}{n} \right) + \sqrt{\frac{d}{n}} \right) \wedge \frac{t}{\sigma} \right] \tag{34}$$

$$= \left[ \frac{\sigma^2}{t} \left( \sqrt{\frac{d}{n}} + \frac{d}{n} \right) + \sigma \sqrt{\frac{d}{n}} \right] \wedge t. \tag{35}$$

Now, for the HMM at hand, the rationale is that the original model (2) is equivalent to $\tilde{X} = \overline{S} \cdot \overline{X}$, where $\overline{S} \sim \text{Unif}\{-1, 1\}$ is the coherent sign of a block of $\frac{1}{\delta}$ raw samples from model (2), $\overline{X} = \frac{1}{1/\delta} \sum_{i=1}^{1/\delta} X_i$ represents the block-sample obtained by applying the estimator (30) to a block of $\frac{1}{\delta}$ raw samples, hence, $\overline{X} \overset{d}{=} \theta_* + N(0, \delta \cdot I_d)$. Furthermore, the signs across different blocks are essentially independent. Therefore

$$\tilde{X} \overset{d}{=} \overline{S}\theta_* + \overline{Z} \tag{36}$$

where $\overline{Z} \sim N(0, \delta \cdot I_d)$, and we have $\frac{n}{1/\delta} = n\delta$ i.i.d. block-samples from model (36). According to (35), the optimal minimax rate of the HMM (2) should be

$$\mathsf{M}_{\text{HMM}}(n, d, \delta, t) \asymp \tilde{\mathsf{M}}_{\text{GMM}}(n\delta, d, t, \sqrt{\delta}) \tag{37}$$

$$\asymp \left[ \frac{\delta}{t} \left( \sqrt{\frac{d}{n\delta}} + \frac{d}{n\delta} \right) + \sqrt{\delta} \sqrt{\frac{d}{n\delta}} \right] \wedge t \tag{38}$$

$$= \left[ \frac{1}{t} \left( \sqrt{\frac{d\delta}{n}} + \frac{d}{n} \right) + \sqrt{\frac{d}{n}} \right] \wedge t. \tag{39}$$

Evaluating (39) immediately yields (7).

The optimal minimax error rates for GLM (cf. (5)), GMM (cf. (6), originally proved in [Wu and Zhou, 2019, Appendix B]) and HMM (cf. (7), implied by our results Theorems 1 and 2) are plotted in Figure 1.

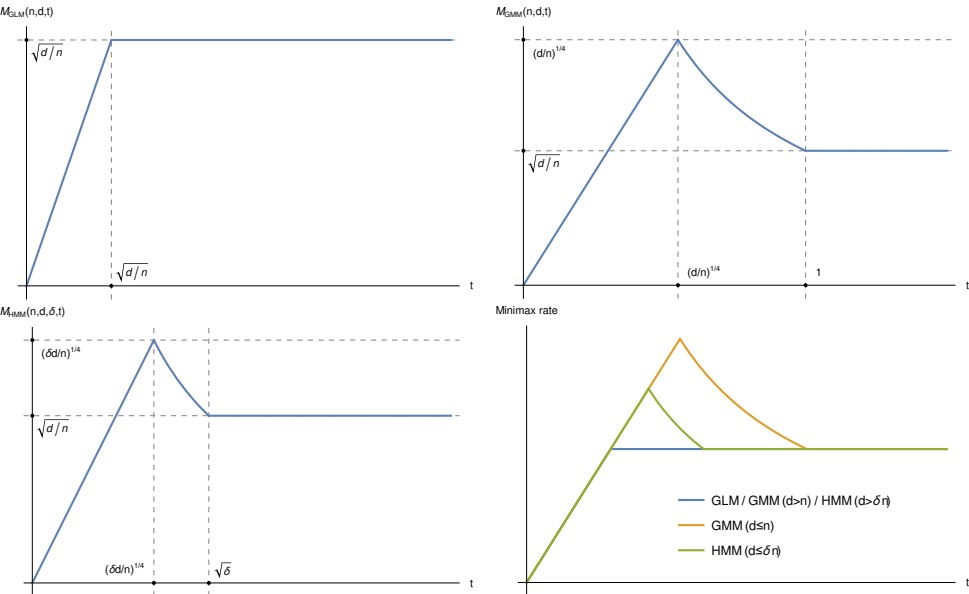

Figure 1: Plots of minimax rates of the Gaussian Location Model, the Gaussian Mixture Model and the Hidden Markov Model. The top left, top right and bottom left figures show the scaling of the minimax rates of GLM (cf. (5)), GMM with $d \leq n$ (cf. (6)) and HMM with $d \leq \delta n$ (cf. (7)), respectively. All the above error rates are plotted in the bottom right figure in which one can clearly see how the rate varies as $\delta$ varies.

# B  Proofs for Section 2: Mean estimation for known $\delta$

## B.1  Proof of Theorem 1: Analysis of the estimator

Our proposed procedure in Theorem 1 for $\theta_*$ estimation with a known flip probability $\delta$ is depicted in Figure 2.

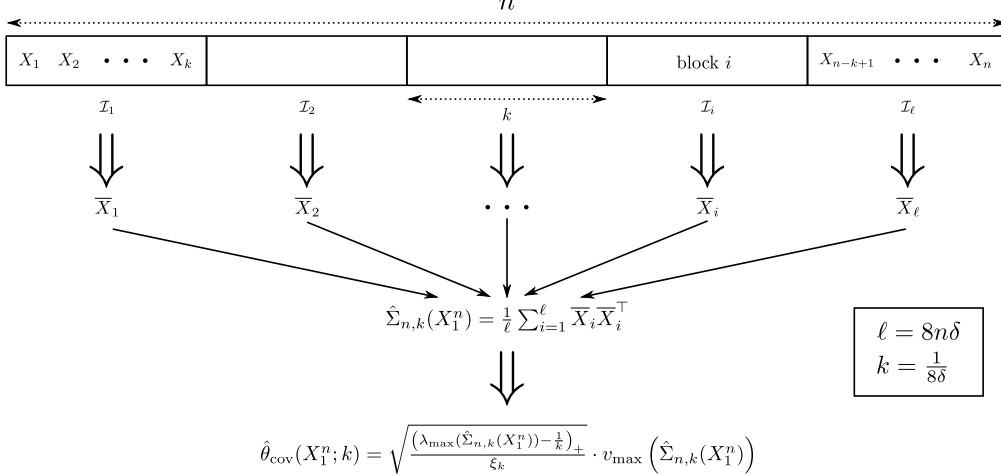

Figure 2: A visual illustration of the construction of the estimator $\hat{\theta}_{\mathrm{cov}}(X_1^n; k)$ in (14) given $n$ samples $X_1^n$ from the model (2) with a known flip probability $\delta$. The block size $k$ is chosen to be $\frac{1}{8\delta}$.

*Remark* 8. The condition $\delta \geq \frac{1}{n}$ in Theorem 1 is inconsequential. Indeed, if $\delta \leq \frac{1}{n}$, then also $d \geq \delta n$ and we are at the high dimension regime. In that case, the estimator can artificially increase the flip probability (by sign randomization) to $\frac{1}{n}$, which results the local minimax rate of the GLM, that is, $\mathsf{M}(n, d, \delta, t) \asymp \mathsf{M}_{\mathrm{GLM}}(n, d, t)$, which is optimal. We therefore assume $\delta \leq \frac{1}{n}$ throughout the proof.

To begin with the analysis of the estimator in Figure 2, the following lemma is a simple, yet key tool for the proof. It establishes the variance of the random gain $\overline{S}$. The proof relies on a sort of *self-bounding* property (cf. [Boucheron et al., 2013, Chapter 3.3]), which stems from the fact that $|\overline{S}| \leq 1$.

**Lemma 9.** *It holds that* $0 \leq \overline{S}^2 \leq 1$ *(with probability 1) and that*

$$\mathbb{V}(\overline{S}^2) \leq \mathbb{E}[1 - \overline{S}^2] \leq 4\delta k. \tag{40}$$

*Proof.* Trivially $0 \leq \overline{S}^2 \leq 1$ and so the variance is bounded as

$$\mathbb{V}(\overline{S}^2) = \mathbb{V}(1 - \overline{S}^2) \leq \mathbb{E}[(1 - \overline{S}^2)^2] \leq \mathbb{E}[1 - \overline{S}^2]. \tag{41}$$

The expected value is then upper bounded as

$$\mathbb{E}[1 - \overline{S}^2] = \mathbb{E}\left[ 1 - \left( \frac{1}{k} \sum_{j=1}^{k} S_j \right)^2 \right] \tag{42}$$

$$= \mathbb{E}\left[ 1 - \left( \frac{1}{k} \sum_{j=1}^{k} S_j \right)^2 \,\Bigg|\, S_0 = 1 \right] \tag{43}$$

$$= \mathbb{E}\left[ \left( 1 - \left( \frac{1}{k} \sum_{j=1}^{k} S_j \right) \right) \left( 1 + \left( \frac{1}{k} \sum_{j=1}^{k} S_j \right) \right) \,\Bigg|\, S_0 = 1 \right] \tag{44}$$

$$\leq 2 \cdot \mathbb{E}\left[1 - \left(\frac{1}{k}\sum_{j=1}^{k} S_j\right) \,\middle|\, S_0 = 1\right] \tag{45}$$

$$= 2 - \frac{2}{k}\sum_{j=1}^{k} \mathbb{E}\left[S_j \,\middle|\, S_0 = 1\right] \tag{46}$$

$$\overset{(a)}{=} 2 - \frac{2}{k}\sum_{j=1}^{k} \rho^j \tag{47}$$

$$= 2 - \frac{2}{k}\sum_{j=1}^{k} (1 - 2\delta)^j \tag{48}$$

$$\overset{(b)}{\leq} 2 - \frac{2}{k}\sum_{j=1}^{k} (1 - 2j\delta) \tag{49}$$

$$= \frac{4\delta}{k}\sum_{j=1}^{k} j \tag{50}$$

$$= \frac{2\delta(k^2 + k)}{k} \tag{51}$$

$$\leq 4\delta k, \tag{52}$$

where $(a)$ follows from

$$\mathbb{E}[S_j \mid S_0 = 1]$$
$$= \mathbb{P}[S_{j-1} = 1 \mid S_0 = 1]\mathbb{E}[S_j \mid S_0 = 1, S_{j-1} = 1]$$
$$\quad + \mathbb{P}[S_{j-1} = -1 \mid S_0 = 1]\mathbb{E}[S_j \mid S_0 = 1, S_{j-1} = -1] \tag{53}$$
$$= \mathbb{P}[S_{j-1} = 1 \mid S_0 = 1]\mathbb{E}[S_j \mid S_{j-1} = 1] + \mathbb{P}[S_{j-1} = -1 \mid S_0 = 1]\mathbb{E}[S_j \mid S_{j-1} = -1] \tag{54}$$
$$= \mathbb{P}[S_{j-1} = 1 \mid S_0 = 1]\left(\frac{1+\rho}{2} - \frac{1-\rho}{2}\right) + \mathbb{P}[S_{j-1} = -1 \mid S_0 = 1]\left(-\frac{1+\rho}{2} + \frac{1-\rho}{2}\right) \tag{55}$$
$$= \mathbb{P}[S_{j-1} = 1 \mid S_0 = 1]\cdot\rho - \mathbb{P}[S_{j-1} = -1 \mid S_0 = 1]\cdot\rho \tag{56}$$
$$= \rho \cdot \mathbb{E}[S_{j-1} \mid S_0 = 1] \tag{57}$$
$$= \cdots = \rho^j, \tag{58}$$

$(b)$ follows from Bernoulli's inequality $(1 - x)^r \geq 1 - rx$ for $x \in [0, 1]$ and $r \geq 1$. $\qquad\square$

The next lemma summarizes concentration results and bounds on the expected value of various empirical quantities needed for the rest of the analysis. For a sequence of samples $(S_1^n, Z_1^n)$, let us denote the following events:

$$\mathcal{E}'_{n,\delta,k} := \left\{ S_1^n : \left|\frac{1}{\ell}\sum_{i=1}^{\ell} \overline{S}_i^2 - \xi_k\right| \leq 5\sqrt{\frac{\delta k^2 \log(n)}{n}} \right\}, \tag{59}$$

$$\mathcal{E}''_{n,d,k} := \left\{ (S_1^n, Z_1^n) : \left\|\frac{1}{\ell}\sum_{i=1}^{\ell} \overline{S}_i \overline{Z}_i\right\| \leq 7\sqrt{\frac{d}{n}} \right\}, \tag{60}$$

$$\mathcal{E}'''_{n,d,k} := \left\{ Z_1^n : \left\|\frac{1}{\ell}\sum_{i=1}^{\ell} \overline{Z}_i \overline{Z}_i^\top - \frac{1}{k}\cdot I_d\right\|_{\mathrm{op}} \leq 4\sqrt{\frac{d}{kn}} + \frac{4d}{n} \right\}, \tag{61}$$

and let

$$\mathcal{E}_{n,d,\delta,k} := \mathcal{E}'_{n,\delta,k} \cap \mathcal{E}''_{n,d,k} \cap \mathcal{E}'''_{n,d,k}. \tag{62}$$

Note that these events depend on the choice of the block length $k$.

**Lemma 10.** *Assume that* $\delta \geq \frac{1}{n}$ *and* $d \leq n$. *It holds that* $\mathbb{P}[\mathcal{E}_{n,d,\delta,k}] \geq 1 - \frac{5}{n}$ *and*

$$\mathbb{E}\left[\left|\frac{1}{\ell}\sum_{i=1}^{\ell}\overline{S}_i^2 - \xi_k\right|\right] \leq 2\sqrt{\frac{\delta k^2}{n}}, \tag{63}$$

$$\mathbb{E}\left[\left\|\frac{1}{\ell}\sum_{i=1}^{\ell}\overline{S}_i\overline{Z}_i\right\|\right] \leq \sqrt{\frac{d}{n}}, \tag{64}$$

*and*

$$\mathbb{E}\left[\left\|\frac{1}{\ell}\sum_{i=1}^{\ell}\overline{Z}_i\overline{Z}_i^\top - \frac{1}{k}\cdot I_d\right\|_{op}\right] \leq 13\sqrt{\frac{d}{nk}} + 10\frac{d}{n}. \tag{65}$$

*Proof.* We analyze the expected value and concentration of each of the three terms.

First, the random variable $\overline{S}_i^2 - \xi_k$ has zero mean, it is bounded in $[-1, 1]$, and its variance is bounded by $4\delta k$ according to Lemma 9. Since $\{\overline{S}_i^2 - \xi_k\}_{i=1}^{\ell}$ are i.i.d., Bernstein's inequality for bounded distributions implies that (e.g., (323) from Wainwright [2019, Proposition 2.14], by setting in the notation therein $X_i = \overline{S}_i^2 - \xi_k$ which is a zero-mean random variable bounded in $[-1, 1]$)

$$\mathbb{P}\left[\left|\frac{1}{\ell}\sum_{i=1}^{\ell}\overline{S}_i^2 - \xi_k\right| \geq t\right] \leq 2\exp\left(-\frac{\ell^2 t^2}{4\ell\delta k + \ell t/3}\right). \tag{66}$$

Requiring that the r.h.s. is at most $\frac{2}{n}$, and using $n = \ell k$, this implies that

$$\left|\frac{1}{\ell}\sum_{i=1}^{\ell}\overline{S}_i^2 - \xi_k\right| \leq \sqrt{\frac{8\delta k^2 \log(n)}{n}} + \frac{\frac{3}{2}k\log(n)}{n} \tag{67}$$

with probability $1 - \frac{2}{n}$. Under the assumption $\delta \geq \frac{1}{n}$, we may further upper bound

$$\sqrt{\frac{8\delta k^2 \log(n)}{n}} + \frac{\frac{3}{2}k\log(n)}{n} \leq \log(n)\cdot\left[\sqrt{\frac{8\delta k^2}{n}} + \frac{\frac{3}{2}k}{n}\right] \tag{68}$$

$$\leq \log(n)\cdot\left[\sqrt{\frac{8\delta k^2}{n}} + \frac{3}{2}\sqrt{\frac{\delta k^2}{n}}\right] \tag{69}$$

$$\leq 5\log(n)\cdot\sqrt{\frac{\delta k^2}{n}}, \tag{70}$$

and this implies $\mathbb{P}[\mathcal{E}'_{n,\delta,k}] \geq 1 - \frac{2}{n}$. In addition, since $\overline{S}_i^2 - \xi_k$ are zero mean i.i.d., Lemma 9 implies that

$$\mathbb{E}\left[\left|\frac{1}{\ell}\sum_{i=1}^{\ell}\overline{S}_i^2 - \xi_k\right|\right] \leq \sqrt{\mathbb{V}\left[\frac{1}{\ell}\sum_{i=1}^{\ell}\overline{S}_i^2 - \xi_k\right]} \leq \sqrt{\frac{4\delta k}{\ell}} = 2\sqrt{\frac{\delta k^2}{n}} \tag{71}$$

which proves (63).

Second, conditioned on any fixed $\{\overline{S}_i\}_{i=1}^{\ell}$, it holds that $\overline{S}_i\overline{Z}_i \sim N(0, \frac{\overline{S}_i^2}{k}I_d)$. Hence, $\frac{1}{\ell}\sum_{i=1}^{\ell}\overline{S}_i\overline{Z}_i \sim N(0, \sigma^2 \cdot I_d)$ with

$$\sigma^2 := \frac{1}{k\ell^2}\sum_{i=1}^{\ell}\overline{S}_i^2 \leq \frac{1}{k\ell} = \frac{1}{n}. \tag{72}$$

So, by standard concentration of norm of Gaussian (or, more generally subGaussian) random vectors (325) from [Rigollet and Hütter, 2019, Theorem 1.19] (or a degenerate case of [Wainwright, 2019, Example 6.2]), it holds with probability $1 - \epsilon$ that

$$\left\|\frac{1}{\ell}\sum_{i=1}^{\ell}\overline{S}_i\overline{Z}_i\right\| \leq 4\sigma\sqrt{d} + 2\sigma\sqrt{2\log\left(\frac{1}{\epsilon}\right)} \leq 4\sqrt{\frac{d}{n}} + \sqrt{\frac{8}{n}\log\left(\frac{1}{\epsilon}\right)}. \tag{73}$$

Since the r.h.s. does not depend on $\{\overline{S}_i\}_{i=1}^{\ell}$, the same bound holds unconditionally. Setting $\epsilon = \frac{1}{n}$ and further upper bounding

$$4\sqrt{\frac{d}{n}} + \sqrt{\frac{8}{n}\log(n)} \leq \sqrt{\frac{32d + 16\log(n)}{n}} \leq 7\sqrt{\frac{d}{n}}, \tag{74}$$

using $\sqrt{a} + \sqrt{b} \leq \sqrt{2(a+b)}$ and $d \geq 2\log(n)$, results $\mathbb{P}[\mathcal{E}''_{n,d,k}] \geq 1 - \frac{1}{n}$. In addition, since $\{\overline{S}_i\overline{Z}_i\}_{i=1}^{\ell}$ are i.i.d., it holds that

$$\mathbb{E}\left[\left\|\frac{1}{\ell}\sum_{i=1}^{\ell}\overline{S}_i\overline{Z}_i\right\|\right] \leq \sqrt{\mathbb{E}\left[\left\|\frac{1}{\ell}\sum_{i=1}^{\ell}\overline{S}_i\overline{Z}_i\right\|^2\right]} = \sqrt{d\sigma^2} \leq \sqrt{\frac{d}{n}} \tag{75}$$

which proves (64).

Third, by Gaussian covariance estimation from [Wainwright, 2019, Example 6.2], it holds for any $\eta > 0$ that

$$V := k\left\|\frac{1}{\ell}\sum_{i=1}^{\ell}\overline{Z}_i\overline{Z}_i^\top - \frac{1}{k}I_d\right\|_{op} \leq \left[2\sqrt{\frac{d}{\ell}} + 2\eta + \left(\sqrt{\frac{d}{\ell}} + \eta\right)^2\right] \tag{76}$$

with probability larger than $1 - 2e^{-\ell\eta^2/2}$.

We begin with a high probability event. Setting $\epsilon = 2e^{-\ell\eta^2/2}$, it holds with probability larger than $1 - \epsilon$ that

$$\left\|\frac{1}{\ell}\sum_{i=1}^{\ell}\overline{Z}_i\overline{Z}_i^\top - \frac{1}{k}\cdot I_d\right\|_{op} \leq 2\sqrt{\frac{d}{k^2\ell}} + \sqrt{\frac{8\log\left(\frac{2}{\epsilon}\right)}{k^2\ell}} + \left(\sqrt{\frac{d}{k\ell}} + \sqrt{\frac{2\log\left(\frac{2}{\epsilon}\right)}{k\ell}}\right)^2 \tag{77}$$

$$= 2\sqrt{\frac{d}{kn}} + \sqrt{\frac{8\log\left(\frac{2}{\epsilon}\right)}{kn}} + \left(\sqrt{\frac{d}{n}} + \sqrt{\frac{2\log\left(\frac{2}{\epsilon}\right)}{n}}\right)^2 \tag{78}$$

$$\overset{(a)}{\leq} 2\sqrt{\frac{d}{kn}} + \sqrt{\frac{8\log\left(\frac{2}{\epsilon}\right)}{kn}} + \frac{2d + 4\log\left(\frac{2}{\epsilon}\right)}{n}, \tag{79}$$

where $(a)$ follows from $(a+b)^2 \leq 2a^2 + 2b^2$. Setting $\epsilon = \frac{2}{n}$ and further upper bounding

$$2\sqrt{\frac{d}{kn}} + \sqrt{\frac{8\log(n)}{kn}} + \frac{2d + 4\log(n)}{n} \leq \sqrt{\frac{8d + 16\log(n)}{kn}} + \frac{2d + 4\log(n)}{n} \tag{80}$$

$$\leq 4\sqrt{\frac{d}{kn}} + \frac{4d}{n}, \tag{81}$$

using $\sqrt{a} + \sqrt{b} \leq \sqrt{2(a+b)}$ and $d \geq 2\log(n)$, results $\mathbb{P}[\mathcal{E}'''_{n,d,k}] \geq 1 - \frac{2}{n}$. For the bound on the expectation, we further upper bound

$$V \leq 2\sqrt{\frac{d}{\ell}} + 2\eta + \frac{2d}{\ell} + 2\eta^2. \tag{82}$$

Let $\alpha = 2\sqrt{\frac{d}{\ell}} + 2\frac{d}{\ell}$. Then, $U := V - \alpha$ satisfies that $\mathbb{P}\left[U \geq 2\eta + 2\eta^2\right] \leq 2e^{-\ell\eta^2/2}$. Then,

$$\mathbb{E}[U] = \mathbb{E}[U \cdot \mathbb{1}\{U < 0\}] + \mathbb{E}[U \cdot \mathbb{1}\{U \geq 0\}] \tag{83}$$

$$\leq \mathbb{E}[U \cdot \mathbb{1}\{U \geq 0\}] \tag{84}$$

$$= \int_0^\infty \mathbb{P}[U \cdot \mathbb{1}\{U \geq 0\} > t]\,\mathrm{d}t \tag{85}$$

$$= \int_0^\infty \mathbb{P}[U > t]\,\mathrm{d}t \tag{86}$$

$$\overset{(a)}{=} \int_0^\infty (2+4\eta)\, \mathbb{P}\left[U > 2\eta + 2\eta^2\right] \mathrm{d}\eta \tag{87}$$

$$\leq \int_0^\infty (2+4\eta)\, 2e^{-\ell\eta^2/2}\mathrm{d}\eta \tag{88}$$

$$= 4\int_0^\infty e^{-\ell\eta^2/2}\mathrm{d}\eta + 8\int_0^\infty \eta e^{-\ell\eta^2/2}\mathrm{d}\eta \tag{89}$$

$$\overset{(b)}{=} 4\sqrt{\frac{2\pi}{\ell}} + \frac{8}{\ell}, \tag{90}$$

where $(a)$ follows from the change of variables $t = 2\eta + 2\eta^2$, and $(b)$ follows from Gaussian integration (the first term). Hence,

$$\mathbb{E}\left[\frac{V}{k}\right] \leq \frac{1}{k}\mathbb{E}[U+\alpha] \leq 13\sqrt{\frac{d}{nk}} + 10\frac{d}{n} \tag{91}$$

which proves (65). The claim $\mathbb{P}[\mathcal{E}_{n,d,\delta,k}] \geq 1 - \frac{5}{n}$ then follows from the union bound over the three events considered above. $\qquad\square$

The next lemma utilizes Lemma 10 to establish concentration on the maximal eigenvalue and the associated eigenvector.

**Lemma 11.** *Let*

$$\psi(n,d,\delta,k) := 2\sqrt{\frac{\delta k^2}{n}} \cdot \|\theta_*\|^2 + 2\sqrt{\frac{d}{n}} \cdot \|\theta_*\| + 13\sqrt{\frac{d}{nk}} + 10\frac{d}{n}. \tag{92}$$

*Then, it holds that*

$$\mathbb{E}\left[\left|\lambda_{\max}\left(\hat{\Sigma}_{n,k}(X_1^n)\right) - \left(\xi_k\|\theta_*\|^2 + \frac{1}{k}\right)\right|\right] \leq \psi(n,d,\delta,k) \tag{93}$$

*and*

$$\mathbb{E}\left[\mathsf{loss}\left(v_{\max}\left(\hat{\Sigma}_{n,k}(X_1^n)\right), \frac{\theta_*}{\|\theta_*\|}\right)\right] \leq \frac{8 \cdot \psi(n,d,\delta,k)}{\|\theta_*\|^2}. \tag{94}$$

*If $\mathcal{E}_{n,d,\delta,k}$ holds then similar bounds hold (with probability 1) with $\psi(n,d,\delta,k)$ multiplied by $7\log(n)$.*

*Proof.* Note that $\lambda_{\max}(\Sigma_{n,k}(\theta_*)) = \xi_k \cdot \|\theta_*\|^2 + \frac{1}{k}$. Then,

$$\left|\lambda_{\max}\left(\hat{\Sigma}_{n,k}(X_1^n)\right) - \left(\xi_k\|\theta_*\|^2 + \frac{1}{k}\right)\right|$$

$$= \left|\lambda_{\max}\left(\hat{\Sigma}_{n,k}(X_1^n)\right) - \lambda_{\max}\left(\Sigma_{n,k}(\theta_*)\right)\right| \tag{95}$$

$$\overset{(a)}{\leq} \left\|\hat{\Sigma}_{n,k}(X_1^n) - \Sigma_{n,k}(\theta_*)\right\|_{\mathrm{op}} \tag{96}$$

$$= \left\|\hat{\Sigma}_{n,k}(X_1^n) - \xi_k\theta_*\theta_*^\top - \frac{1}{k}I_d\right\|_{\mathrm{op}} \tag{97}$$

$$\leq \left|\frac{1}{\ell}\sum_{i=1}^\ell \overline{S}_i^2 - \xi_k\right| \cdot \|\theta_*\|^2 + \left\|\frac{2}{\ell}\sum_{i=1}^\ell \overline{S}_i\overline{Z}_i\right\| \cdot \|\theta_*\| + \left\|\frac{1}{\ell}\sum_{i=1}^\ell \overline{Z}_i\overline{Z}_i^\top - \frac{1}{k} \cdot I_d\right\|_{\mathrm{op}}, \tag{98}$$

where $(a)$ follows from Weyl's inequality. Taking expectation and using Lemma 10 proves the claimed bound.

Next, note that $v_{\max}(\Sigma_{n,k}(\theta_*)) = \frac{\theta_*}{\|\theta_*\|}$, its eigenvalue is $\xi_k\|\theta_*\|^2 + \frac{1}{k}$, and that all the $(d-1)$ other eigenvalues of $\Sigma_{n,k}(\theta_*)$ are $\frac{1}{k}$. Thus, the eigen-gap of the maximal eigenvalue is $\xi_k\|\theta_*\|^2$. Then, by Davis-Kahan's perturbation bound (328) from [Vershynin, 2018, Theorem 4.5.5]

$$\mathsf{loss}\left(v_{\max}\left(\hat{\Sigma}_{n,k}(X_1^n)\right), \frac{\theta_*}{\|\theta_*\|}\right) = \mathsf{loss}\left(v_{\max}\left(\hat{\Sigma}_{n,k}(X_1^n)\right), v_{\max}\left(\Sigma_{n,k}(\theta_*)\right)\right)$$

$$\leq 4 \frac{\left\| \hat{\Sigma}_{n,k}(X_1^n) - \Sigma_{n,k}(\theta_*) \right\|_{\text{op}}}{\xi_k \|\theta_*\|^2}$$

$$\overset{(a)}{\leq} 8 \frac{\left\| \hat{\Sigma}_{n,k}(X_1^n) - \Sigma_{n,k}(\theta_*) \right\|_{\text{op}}}{\|\theta_*\|^2},$$

where $(a)$ follows since $\xi_k = \mathbb{E}[\overline{S}^2] \geq 1 - 4k\delta = \frac{1}{2}$ according to Lemma 9, and the choice $k = \frac{1}{8\delta}$. The operator norm is upper bounded as for the maximal eigenvalue. The proof then follows by taking expectation, and utilizing Lemma 10.

Finally, assuming the event $\mathcal{E}_{n,d,\delta,k}$ holds, Lemma 10 implies that the same bound holds with $7 \log(n) \cdot \psi(n, d, \delta, k)$ instead of $\psi(n, d, \delta, k)$. $\qquad\square$

The next lemma upper bounds $\hat{\theta}$ in expectation and in high probability whenever $\|\theta_*\|$ is below the minimax rate $\beta(n, d, \delta) = \Theta(\sqrt{\frac{d}{n}} \vee (\frac{\delta d}{n})^{1/4})$, and show that both are (roughly) at most on the scale of $\beta(n, d, \delta)$.

**Lemma 12.** *If $\|\theta_*\| \leq \beta(n, d, \delta)$ and $k = \frac{1}{8\delta}$ then*

$$\mathbb{E}\left[\left\| \hat{\theta}_{cov}(X_1^n; k) \right\|\right] \leq 11 \cdot \beta(n, d, \delta), \tag{99}$$

*and if the event $\mathcal{E}_{n,d,\delta,k}$ holds then*

$$\left\| \hat{\theta}_{cov}(X_1^n; k) \right\| \leq 77 \log(n) \cdot \beta(n, d, \delta). \tag{100}$$

*Proof.* It holds that

$$\|\hat{\theta}_{\text{cov}}(X_1^n; k)\|^2 = \frac{\lambda_{\max}\left(\hat{\Sigma}_{n,k}(X_1^n)\right) - \frac{1}{k}}{\xi_k} \vee 0 \tag{101}$$

$$\overset{(a)}{\leq} 2 \left[\lambda_{\max}\left(\hat{\Sigma}_{n,k}(X_1^n)\right) - \frac{1}{k}\right] \vee 0 \tag{102}$$

$$\leq 2 \left[\lambda_{\max}\left(\hat{\Sigma}_{n,k}(X_1^n)\right) - \xi_k \|\theta_*\|^2 - \frac{1}{k}\right] \vee 0 + 2\xi_k \|\theta_*\|^2 \tag{103}$$

$$\overset{(b)}{\leq} 2 \left[\lambda_{\max}\left(\hat{\Sigma}_{n,k}(X_1^n)\right) - \xi_k \|\theta_*\|^2 - \frac{1}{k}\right] \vee 0 + 2\|\theta_*\|^2, \tag{104}$$

where $(a)$ follows since $\xi_k \geq \frac{1}{2}$ and $(b)$ follows since $\xi_k \leq 1$. Taking expectation of both sides, and utilizing Lemma 11 results

$$\mathbb{E}\left[\|\hat{\theta}_{\text{cov}}(X_1^n; k)\|^2\right]$$
$$\leq 2 \cdot \psi(n, d, \delta, k) + 2\|\theta_*\|^2 \tag{105}$$

$$\leq 4\sqrt{\frac{\delta k^2}{n}} \cdot \|\theta_*\|^2 + 4\sqrt{\frac{d}{n}} \cdot \|\theta_*\| + 26\sqrt{\frac{d}{nk}} + 20\frac{d}{n} + 2\|\theta_*\|^2 \tag{106}$$

$$\overset{(a)}{=} 4\sqrt{\frac{1}{64\delta n}} \cdot \|\theta_*\|^2 + 4\sqrt{\frac{d}{n}} \cdot \|\theta_*\| + 26\sqrt{\frac{8\delta d}{n}} + 20\frac{d}{n} + 2\|\theta_*\|^2 \tag{107}$$

$$\overset{(b)}{\leq} 4\sqrt{\frac{1}{64\delta n}} \cdot \left(\frac{2d}{n} + 2\sqrt{\frac{\delta d}{n}}\right) + 4\sqrt{\frac{d}{n}} \cdot \left(\sqrt{\frac{d}{n}} + \left(\frac{\delta d}{n}\right)^{1/4}\right)$$

$$\qquad + 26\sqrt{\frac{8\delta d}{n}} + 20\frac{d}{n} + 2\left(\frac{2d}{n} + 2\sqrt{\frac{\delta d}{n}}\right) \tag{108}$$

$$\leq 4\sqrt{\frac{1}{64\delta n}} \cdot \left(\frac{2d}{n} + 2\sqrt{\frac{\delta d}{n}}\right) + 4\sqrt{\frac{d}{n}} \cdot \left(\frac{\delta d}{n}\right)^{1/4} + 78\sqrt{\frac{\delta d}{n}} + 28\frac{d}{n} \tag{109}$$

$$= \frac{d}{\sqrt{\delta}n^{3/2}} + \sqrt{\frac{d}{n^2}} + 4\sqrt{\frac{d}{n}} \cdot \left(\frac{\delta d}{n}\right)^{1/4} + 78\sqrt{\frac{\delta d}{n}} + 28\frac{d}{n} \tag{110}$$

$$\overset{(c)}{\leq} 4\sqrt{\frac{d}{n}} \cdot \left(\frac{\delta d}{n}\right)^{1/4} + 78\sqrt{\frac{\delta d}{n}} + 30\frac{d}{n} \tag{111}$$

$$\overset{(d)}{\leq} 80\sqrt{\frac{\delta d}{n}} + 32\frac{d}{n}, \tag{112}$$

where $(a)$ follows by setting $k = \frac{1}{8\delta}$, and $(b)$ follows from the assumption $\|\theta_*\| \leq \beta(n,d,\delta) \leq \sqrt{\frac{d}{n}} + \left(\frac{\delta d}{n}\right)^{1/4}$ (and $(a+b)^2 \leq 2a^2 + 2b^2$), $(c)$ follows from $\delta \geq \frac{1}{n}$, so that $\frac{d}{\sqrt{\delta}n^{3/2}} \leq \frac{d}{n}$, and $(d)$ follows from $ab \leq \frac{1}{2}a^2 + \frac{1}{2}b^2$ applied to the first term.

The proof is then completed by Jensen's inequality

$$\mathbb{E}\left[\|\hat{\theta}_{\text{cov}}(X_1^n; k)\|\right] \leq \sqrt{\mathbb{E}\left[\|\hat{\theta}_{\text{cov}}(X_1^n; k)\|^2\right]} \tag{113}$$

$$\leq \sqrt{80\sqrt{\frac{\delta d}{n}} + 32\frac{d}{n}} \tag{114}$$

$$\leq \sqrt{112 \cdot \left(\sqrt{\frac{\delta d}{n}} \vee \frac{d}{n}\right)} \tag{115}$$

$$\leq 11 \cdot \beta(n,d,\delta). \tag{116}$$

If $\lambda_{\max}(\hat{\Sigma}_{n,k}(X_1^n)) - \frac{1}{k} \leq 0$ then $\hat{\theta}_{\text{cov}}(X_1^n; k) = 0$ and the claim is trivial. Otherwise, from the same reasoning as above, assuming that the event $\mathcal{E}_{n,d,\delta,k}$ holds, it also holds from Lemma 11 that $\|\hat{\theta}_{\text{cov}}(X_1^n; k)\| \leq 77\log(n) \cdot \beta(n,d,\delta)$.  $\square$

We may now prove Theorem 1.

*Proof of Theorem 1.* If $\|\theta_*\| \leq \beta(n,d,\delta)$ then Lemma 12 implies that

$$\mathbb{E}\left[\left\|\hat{\theta}_{\text{cov}}(X_1^n; k)\right\|\right] \leq 11 \cdot \beta(n,d,\delta). \tag{117}$$

Otherwise, assume that $\|\theta_*\| \geq \beta(n,d,\delta)$. For any estimator $\tilde{\theta}$ we may write $\tilde{\theta} = \|\tilde{\theta}\| \cdot v$ where $v \in \mathbb{S}^{d-1}$, it holds that

$$\text{loss}(\tilde{\theta}, \theta_*) = \left\|\|\tilde{\theta}\| \cdot v - \theta_*\right\| \tag{118}$$

$$= \left\|\|\tilde{\theta}\| \cdot v - \|\theta_*\| \cdot v + \|\theta_*\| \cdot v - \theta_*\right\| \tag{119}$$

$$\leq \left|\|\tilde{\theta}\| - \|\theta_*\|\right| + \|\theta_*\| \cdot \text{loss}\left(v, \frac{\theta_*}{\|\theta_*\|}\right), \tag{120}$$

where the last inequality follows from the triangle inequality. Specifying this result to $\tilde{\theta} = \hat{\theta} \equiv \hat{\theta}_{\text{cov}}(X_1^n; k)$, results

$$\text{loss}(\hat{\theta}, \theta_*) \leq \left|\|\hat{\theta}\| - \|\theta_*\|\right| + \|\theta_*\| \cdot \text{loss}\left(v_{\max}\left(\hat{\Sigma}_{n,k}(X_1^n)\right), \frac{\theta_*}{\|\theta_*\|}\right). \tag{121}$$

Lemma 11 implies that the first term in (121) is bounded as

$$\left|\|\hat{\theta}\| - \|\theta_*\|\right| = \frac{\left|\|\hat{\theta}\|^2 - \|\theta_*\|^2\right|}{\|\hat{\theta}\| + \|\theta_*\|} \tag{122}$$

$$\leq \frac{\left|\|\hat{\theta}\|^2 - \|\theta_*\|^2\right|}{\|\theta_*\|} \tag{123}$$

$$= \left| \frac{\left( \lambda_{\max}(\hat{\Sigma}_{n,k}(X_1^n)) - \frac{1}{k} \right)_+ - \xi_k \|\theta_*\|^2}{\xi_k \|\theta_*\|} \right| \tag{124}$$

$$\leq \frac{\left| \lambda_{\max}(\hat{\Sigma}_{n,k}(X_1^n)) - \frac{1}{k} - \xi_k \|\theta_*\|^2 \right|}{\xi_k \|\theta_*\|} \tag{125}$$

$$\leq \frac{2 \left| \lambda_{\max}(\hat{\Sigma}_{n,k}(X_1^n)) - \frac{1}{k} - \xi_k \|\theta_*\|^2 \right|}{\|\theta_*\|}, \tag{126}$$

where the last inequality follows since $\xi_k \geq 1 - 4k\delta$ (from Lemma 9), and $k \leq \frac{1}{8\delta}$. Taking expectation of both sides, and utilizing Lemma 11 results

$$\mathbb{E}\left[ \left| \|\hat{\theta}\| - \|\theta_*\| \right| \right] \leq \frac{2 \cdot \psi(n,d,\delta,k)}{\|\theta_*\|}. \tag{127}$$

Lemma 11 further implies that the second term in (121) is bounded as

$$\mathbb{E}\left[ \|\theta_*\| \cdot \mathsf{loss}\left( v_{\max}\left( \hat{\Sigma}_{n,k}(X_1^n) \right), \frac{\theta_*}{\|\theta_*\|} \right) \right] \leq \frac{8 \cdot \psi(n,d,\delta,k)}{\|\theta_*\|}. \tag{128}$$

Hence, (121)

$$\mathsf{loss}(\hat{\theta}, \theta_*) \leq \frac{10 \cdot \psi(n,d,\delta,k)}{\|\theta_*\|} \tag{129}$$

$$\leq 20\sqrt{\frac{\delta k^2}{n}} \cdot \|\theta_*\| + 20\sqrt{\frac{d}{n}} + \frac{130}{\|\theta_*\|}\sqrt{\frac{d}{nk}} + 100\frac{1}{\|\theta_*\|}\frac{d}{n} \tag{130}$$

$$\overset{(a)}{=} 20\sqrt{\frac{1}{64\delta n}} \cdot \|\theta_*\| + 20\sqrt{\frac{d}{n}} + \frac{130}{\|\theta_*\|}\sqrt{\frac{8\delta d}{n}} + \frac{100}{\|\theta_*\|}\frac{d}{n} \tag{131}$$

$$\overset{(b)}{\leq} 20\sqrt{\frac{d}{n}} + \frac{130}{\|\theta_*\|}\sqrt{\frac{8\delta d}{n}} + \frac{110}{\|\theta_*\|}\frac{d}{n}, \tag{132}$$

where $(a)$ follows from setting $k = \frac{1}{8\delta}$, and $(b)$ follows since for $\|\theta_*\| \geq \beta(n,d,\delta)$,

$$20\sqrt{\frac{1}{64\delta n}} \cdot \|\theta_*\| = 20\frac{\|\theta_*\|^2}{\|\theta_*\|}\sqrt{\frac{1}{64\delta n}} \tag{133}$$

$$\leq 20 \cdot \frac{\left( \sqrt{\frac{d}{n}} + \left( \frac{\delta d}{n} \right)^{1/4} \right)^2}{\|\theta_*\|} \cdot \sqrt{\frac{1}{64\delta n}} \tag{134}$$

$$\leq 5 \cdot \frac{\frac{d}{n} + \sqrt{\frac{\delta d}{n}}}{\|\theta_*\|} \cdot \sqrt{\frac{1}{\delta n}} \tag{135}$$

$$= 5 \cdot \frac{1}{\|\theta_*\|} \left( \frac{d}{n^{3/2}\sqrt{\delta}} + \frac{\sqrt{d}}{n} \right) \tag{136}$$

$$\leq \frac{5}{\|\theta_*\|}\frac{d}{n^{3/2}\sqrt{\delta}} + \frac{5}{\|\theta_*\|}\frac{d}{n} \tag{137}$$

$$\leq \frac{10}{\|\theta_*\|}\frac{d}{n}, \tag{138}$$

where the last inequality follows since $\delta \geq \frac{1}{n}$. $\qquad \square$

## B.2 Proof of Theorem 2: Impossibility lower bound

**The proof's main ideas** The estimation error for the Markov model cannot be better than a genie-aided model for which the sign is known during a block whose size is larger than the mixing time

$\Theta(\frac{1}{\delta})$ of the original Markov chain. The estimator can then align the signs of the samples in each block, and thus reduce the noise variance by a factor of $\delta$. This effectively reduces the problem into a Markov model whose flip probability is asymptotically close to $1/2$, that is, the resulting model is close to a GMM. This aforementioned closeness is quantified by a uniform bound on the ratio between the probability distributions. Based on the closeness of the models, the known lower bound for the GMM model [Wu and Zhou, 2019, Appendxi B] implies a bound on the estimation error in the Markov model.

The proof follows from an application of Fano's method [Wainwright, 2019, Section 15.3] [Yang and Barron, 1999], where, as usual, the main technical challenge follows from the bounding of the mutual information term. We first begin with a brief description of the lower bound on the estimation error, and then explain how it is solved in the Gaussian mixture case in [Wu and Zhou, 2019, Appendix B]. We then describe our analysis method for the Markov case.

**Construction of a packing set**  The version of Fano's method that we use is based on constructing a packing set $\Theta_M := \{\theta_m\}_{m \in [M]} \subset \mathbb{R}^d$ and an additional center vector $\theta_0 \in \mathbb{R}^d$. Here, the set $\Theta_M$ packs points in a spherical cap of a fixed angle with a center at $\theta_0$. To construct this set, we let $\Phi_M := \{\phi_m\}_{m \in [M]} \subset \mathbb{B}^{d-2}$ be a $\frac{1}{16}$-packing set of $\mathbb{B}^{d-2}$ of size larger than $M \geq 16^{d-2}$ in the Euclidean distance, whose existence is assured by a standard argument (e.g., (329) from [Wainwright, 2019, Lemma 5.7 and Example 5.8]). We then append to each $\phi_m$ another coordinate to obtain $\overline{\phi}_m := (\phi_m, \sqrt{1 - \|\phi_m\|}) \in \mathbb{S}^{d-2}$. Then, $\overline{\Phi}_M := \{\overline{\phi}_m\}_{m \in [M]} \subset \mathbb{S}^{d-2}$ is a $\frac{1}{16}$-packing set of $\mathbb{S}^{d-2}$ of size $M$. Now, the packing set $\overline{\Phi}_M$ packs points in the Euclidean distance, however, the loss function $\mathsf{loss}(\cdot, \cdot)$ in (3) is sign-insensitive, and so the packing set requires further dilution. Specifically, there must exist an orthant in $\mathbb{R}^{d-1}$ which contains at least a $2^{-(d-1)}$ fraction of the points in $\overline{\Phi}_M$. So, there must also exist a rotation matrix $O \in \mathbb{R}^{(d-1)\times(d-1)}$ of $\overline{\Phi}_M$ so that $\tilde{\Phi}_M := O\overline{\Phi}_M \cap \mathbb{R}_+^{d-1} = \{O\overline{\phi}_m\}_{m \in [M]} \cap \mathbb{R}_+^{d-1}$ has at least $\frac{|\overline{\Phi}_M|}{2^{d-1}} \geq \frac{1}{16} \cdot 8^{d-1}$ points (with $\mathbb{R}_+^{d-1}$ being the positive orthant). Furthermore, for any distinct $\tilde{\phi}_m, \tilde{\phi}_{m'} \in \tilde{\Phi}_M$ it holds that $\mathsf{loss}(\tilde{\phi}_m, \tilde{\phi}_{m'}) = \|\tilde{\phi}_m - \tilde{\phi}_{m'}\|$. We now choose $\epsilon \in (0, 1)$, set $\tilde{\theta}_0 = [1, 0, \ldots, 0] \in \mathbb{R}^d$ to be the center vector (this choice is arbitrary and made for convenience), and $\tilde{\theta}_m = [\sqrt{1 - \epsilon^2}, \epsilon\tilde{\phi}_m] \in \mathbb{R}^d$, for $m \in [M]$. Evidently, the angle between any $\tilde{\theta}_m$ and $\tilde{\theta}_0$ is fixed for all $m \in [M]$. Finally, given the prescribed norm $t$ in the statement of the theorem, we set the packing set that will be next used in Fano's-inequality based argument as $\Theta_M := \{t \cdot \tilde{\theta}_m\}_{m \in [M]}$ and set $\theta_0 = t\tilde{\theta}_0$. To summarize, $\Theta_M$ satisfies the following properties: (i) $|\Theta_M| \geq \frac{1}{16} \cdot 8^{d-1}$, (ii) for any $\theta_m \in \Theta_M$ it holds that $\|\theta_m\| = t$, (iii) for any distinct $\theta_m, \theta_{m'} \in \Theta_M$ it holds that $\mathsf{loss}(\theta_m, \theta_{m'}) = \|\theta_m - \theta_{m'}\| \geq \frac{1}{16}\epsilon t$, (iv) for any $\theta_m \in \Theta_m$ it holds that

$$\mathsf{loss}(\theta_m, \theta_0) = \|\theta_m - \theta_0\| = t \cdot \sqrt{(\sqrt{1 - \epsilon^2} - 1)^2 + \epsilon^2} \leq t \cdot \left[\sqrt{1 - \epsilon^2} - 1 + \epsilon\right] \leq 2t\epsilon, \quad (139)$$

where the first inequality is from $\sqrt{a^2 + b^2} \leq a + b$ for $a, b \in \mathbb{R}_+$ and the second inequality from $\sqrt{1 - \epsilon^2} \leq 1 + \epsilon$.

**Fano's inequality based lower bound**  Recall that $P_\theta^{(n)}$ is the probability distribution of $X_1^n$ under the Markov model $X_i = S_i\theta + Z_i$, $i \in [n]$ with $S_0^n$ is as in (1). Let $J \sim \mathsf{Unif}[M]$ and assume that given a prescribed norm $t > 0$, it holds that $X_1^n \mid J = j \sim P_{\theta_j}^{(n)}$. Then, based on the four properties of the packing set $\Theta_M$, Fano's method states that [Wainwright, 2019, Proposition 15.2]

$$\mathsf{M}(n, d, \delta, t) \geq \frac{\epsilon t}{32} \cdot \left[1 - \frac{I(J; X_1^n) + \log 2}{\log M}\right] \tag{140}$$

$$\geq \frac{\epsilon t}{32} \cdot \left[1 - \frac{I(J; X_1^n) + \log 2}{(d-1)\log 8}\right] \tag{141}$$

$$\overset{(a)}{\geq} \frac{\epsilon t}{32} \cdot \left[\frac{1}{2} - 4 \cdot \frac{I(J; X_1^n)}{d}\right] \tag{142}$$

$$\overset{(b)}{\geq} \frac{\epsilon t}{32} \cdot \left[\frac{1}{2} - 4 \cdot \frac{\max_{m \in [M]} \mathrm{D}_{\mathrm{KL}}(P_{\theta_m}^{(n)} \| P_{\theta_0}^{(n)})}{d}\right] \tag{143}$$

where $I(J; X_1^n)$ is the mutual information between $J$ and $X_1^n$, and in $(a)$ we have used the assumption $d \geq 3$ which implies $(d-1) \log 8 - \log 16 \geq \frac{d}{4}$, and in $(b)$ we have used the standard "information-radius" bound on the mutual information (e.g., (15.52) in [Wainwright, 2019, Proof of Lemma 15.21 and Exercise 15.11]). The crux of the proof is to establish a bound of the form $D_{KL}(P_{\theta_m}^{(n)} \| P_{\theta_0}^{(n)}) \leq \frac{d}{16}$ for a given $\epsilon_* > 0$ to obtain a lower bound of $\Omega(\epsilon_* t)$. Note that due to symmetry of the packing set, the last KL divergence is the same for all $m \in [M]$.

Before continuing the proof for the Markov case with $\delta < \frac{1}{2}$, we next describe the analysis of the Gaussian mixture model in [Wu and Zhou, 2019]. For the Gaussian mixture model (that is, a degenerate Markov model with $\delta = \frac{1}{2}$), $P_\theta^{(n)}$ is an i.i.d. distribution, and so the tensorization property of the KL divergence immediately implies that $D_{KL}(P_{\theta_m}^{(n)} \| P_{\theta_0}^{(n)}) = n \cdot D_{KL}(P_{\theta_m} \| P_{\theta_0})$ (where $P_\theta \equiv P_\theta^{(1)}$). Then, it was established in [Wu and Zhou, 2019, Lemma 27] that $D_{KL}(P_{\theta_m} \| P_{\theta_0}) \lesssim t^2 \cdot \|\theta_0 - \theta_m\|^2$ as follows. First, it was noted that under $\theta_0 = [t, 0, \ldots, 0] \in \mathbb{R}^d$, $P_\theta = P_t \otimes N(0, I_{d-1})$, that is a product distribution, with the first coordinate being one-dimensional Gaussian mixture with means $\pm t$, and all the other $d-1$ coordinates being standard Gaussian. Based on this and the chain rule of the KL divergence, the KL divergence $D_{KL}(P_{\theta_m} \| P_{\theta_0})$ was evaluated separately by the KL divergence between the distribution of the first coordinate, and the KL divergence between the distributions of the other $d-1$ coordinates, conditioned on the first coordinates. Each of these two KL divergences was bounded by the corresponding chi-square divergences, and was further shown to be $O(t^2 \cdot \|\theta_0 - \theta_m\|^2)$. Finally, the minimax lower bound stated in (6) was obtained by setting $\epsilon = c \cdot \min\{1, \frac{1}{t^2}\sqrt{\frac{d}{n}}\}$ for some small enough $c > 0$ in the construction of the packing set.

For the Markov model (with $\delta < \frac{1}{2}$), it seems rather cumbersome to bound $D_{KL}(P_{\theta_m}^{(n)} \| P_{\theta_0}^{(n)})$ since the model $P_\theta^{(n)}$ has memory. Thus we next propose an indirect approach, which requires three preliminary steps. At the first step, we reduce, via a genie-based argument, the original Gaussian-Markov model with $n$ samples to a Gaussian-Markov model with $\ell$ samples, where each of these $\ell$ sample is a coherent average of a block of $k$ consecutive samples, $\ell = \frac{n}{k}$ (in a similar, yet not identical, form to the estimator from Section 2). The sequences of signs underlying each of these $\ell$ samples also forms a Markov model, where the block length $k$ is judiciously chosen (roughly) as $\Theta(\frac{1}{\delta})$ so that the dependency between the signs is much weaker, in the sense that the flip probability $\bar{\delta}$ of its signs *asymptotically* tends to $\frac{1}{2}$ as $\delta \to 0$. Since for Gaussian mixture models the flip probability is *exactly* $\frac{1}{2}$, this Markov model is *approximately* a Gaussian mixture model.

At the second step, we develop a change-of-measure argument. As said, the reduced model is only approximately Gaussian mixture ($\bar{\delta} \approx \frac{1}{2}$); Had it was exactly a Gaussian mixture model ($\bar{\delta} = \frac{1}{2}$) then the bound on the KL divergence from [Wu and Zhou, 2019, Lemma 27] could have been directly used. Nonetheless, since $\bar{\delta} \to \frac{1}{2}$ with our choice of $k$, the probability distribution of the samples tends to that of the memoryless Gaussian mixture. Since divergences are essentially a continuous functions of the measured distributions, we expect that the KL divergence for the Markov model with $\bar{\delta}$ is close to the KL divergence for the Gaussian mixture model. Lemma 13 forms the basis of this change-of-measure argument in terms of the probability distribution of the signs, and Lemma 14 provides the corresponding change-of-measure bound for the KL divergence.

At the third step, we bound the chi-square divergence for a $d$ dimensional Gaussian mixture model. As described above, in [Wu and Zhou, 2019, Lemma 27] the KL divergence was upper bounded for the Gaussian mixture model by first separating to the KL divergence in the first coordinate and the KL divergence in all other $d-1$ coordinates, exploiting the chain rule of the KL divergence, and then bounding each of the two KL divergences by a chi-square divergence. Here, we provide a refined proof which directly bounds the chi-square divergence, without the need to separate the first coordinate from the others. This is the result of Lemma 15.

The proof of Theorem 2 is then completed using the results of the three preliminary steps. We next turn to the detailed proof.

**First step (genie-aided reduction):** In this step we reduce the original model $P_\theta^{(n)}$ to a model $\tilde{P}_\theta^{(\ell)}$ which is aided by knowledge from a genie, and so estimation errors in the new model are only lower. Inspired by the operation of our proposed estimator in Section 2, we consider $\ell = \frac{n}{k}$ blocks of size

$k$ each, where, for simplicity of exposition, we assume that both $\ell, k$ are integers. A genie informs the estimator with the sign changes $S_j S_{j+1}$ at the time indices $j \in [n-1] \setminus \{ik\}_{i=1}^{\ell-1}$, that is, at all times *except for* $j = k, 2k, 3k, \cdots, n$. Hence, the estimator can align the signs within each block $\mathcal{I}_i := \{(i-1)k+1, (i-1)k+2, \cdots, ik\}$ of size $k$, but not between blocks. Consequently, a statistically *equivalent* model to the original model with genie information is the model

$$Y_i = R_i \theta_* + Z_i \tag{144}$$

where $i \in [n]$, $Z_i \sim N(0, I_d)$ are i.i.d. exactly as in the original Markov model, and the signs $R_i \in \{-1, 1\}$ are such that

$$R_{i+1} = R_i \tag{145}$$

with probability 1 if $i \in [n-1] \setminus \{jk\}_{j=1}^{\ell-1}$ and

$$R_{jk+1} = \begin{cases} R_{(j-1)k+1}, & \text{w.p. } \frac{1+\rho^k}{2} \\ -R_{(j-1)k+1}, & \text{w.p. } \frac{1-\rho^k}{2} \end{cases} \tag{146}$$

if $i \in \{jk\}_{j=1}^{\ell-1}$. Now, since the signs are fixed during the blocks $\{\mathcal{I}_i\}_{i=1}^{\ell}$ of length $k$, the mean of the samples in each block is a *sufficient statistic* for estimation of $\theta_*$. Thus, in the same spirit of the estimator in Section 2, the genie-aided model is equivalent to

$$\overline{Y}_i := \frac{1}{k} \sum_{j \in \mathcal{I}_i} Y_i = \overline{R}_i \theta_* + \overline{Z}_i \tag{147}$$

where $i \in [\ell]$, $\mathbb{P}[\overline{R}_0 = 1] = \frac{1}{2}$, and

$$\overline{R}_{i+1} = \begin{cases} \overline{R}_i & \text{w.p. } \frac{1+\rho^k}{2} \\ -\overline{R}_i & \text{w.p. } \frac{1-\rho^k}{2} \end{cases} \tag{148}$$

is the gain (average of the signs) of the $i$th block, and

$$\overline{Z}_i := \frac{1}{k} \sum_{j \in \mathcal{I}_i} Z_j \sim N\left(0, \frac{1}{k} \cdot I_d\right) \tag{149}$$

is a averaged Gaussian noise. Thus, there are $\ell$ samples in the equivalent model, and the flip probability is $\overline{\delta} := \frac{1-\rho^k}{2}$, which is closer to $\frac{1}{2}$ compared to the flip probability in the original model, $\delta = \frac{1-\rho}{2}$. We note that there are two differences compared to the model used by the estimator in Section 2. First, due to the information supplied from the genie, here the average gain is $\overline{R}_i \in \{-1, 1\}$ with probability 1, to wit $|\overline{R}_i|$ never drops below 1; Second, there is no randomization between the blocks. In the definition of the estimator in Section 2 we indeed had the freedom to randomize the first sign in each block in order to make the blocks statistically independent, whereas here, in an impossibility bound, it is possible that statistical dependence between the blocks may result improved estimation rates (though as we shall see, this is essentially not the case). In what follows, we denote by $\overline{P}_\theta^{(\ell)}$ the probability distribution of $\overline{Y}_1^\ell$ under the model (147) with flip probability $\overline{\delta}$ (i.e., as with signs as in (148)). At this point, the trade-off involved in an optimal choice of $k$ is already apparent – as the block length $k$ increases, the dependence between the blocks, as reflected by the flip probability $\overline{\delta} = \frac{1-\rho^k}{2}$, decreases. On the other hand, the largest lower bound is attained when the genie reveals minimal information, that is, when $k$ is minimal. Our choice of $k$ is thus essentially the minimal value required so that $\frac{1-\rho^k}{2} \to \frac{1}{2}$, and specifically chosen as $k = \frac{\log(n)}{\delta}$.

**Second step (change of measure):** We denote by $p_\delta(s_1^\ell) = \mathbb{P}[S_1^\ell = s_1^\ell]$ the probability distribution of $\ell$ sign samples drawn according to the law of a homogeneous binary symmetric Markov chain with $\mathbb{P}[S_0 = 1] = 1/2$ and flip probability $\delta$. Note that $p_{1/2}(s_1^\ell)$ is then an i.i.d. model, with uniform random signs. The next lemma uniformly bounds the ratio between $p_\delta(s_1^\ell)$ and $p_{1/2}(s_1^\ell)$ when $\delta$ is close to $1/2$, as was obtained in the first step.

**Lemma 13.** *Let $S_0^\ell \in \{-1, 1\}^{\ell+1}$ be a homogeneous binary symmetric Markov chain with $\mathbb{P}[S_0 = 1] = 1/2$ and flip probability $\delta \in [0, \frac{1}{2}]$, and let $p_\delta(s_1^\ell) = \mathbb{P}[S_1^\ell = s_1^\ell]$. Furthermore, for $\overline{\delta} = \frac{1-\rho^k}{2}$ where $\rho = 1 - 2\delta$ and $k = \frac{\log(n)}{\delta}$ it holds that*

$$1 - \frac{1}{n} \le \frac{p_{\overline{\delta}}(s_1^\ell)}{p_{1/2}(s_1^\ell)} \le 1 + \frac{2}{n}. \tag{150}$$

*Proof.* Note that $p_{1/2}(s_1^\ell) = \frac{1}{2^\ell}$ for all $s_1^\ell \in \{-1, 1\}^\ell$. Then, since $\delta \in [0, \frac{1}{2}]$, letting $\rho = 1 - 2\delta$,

$$p_\delta(s_1^\ell) \leq (1 - \delta)^\ell = \left(\frac{1 + \rho}{2}\right)^\ell = p_{1/2}(s_1^\ell)(1 + \rho)^\ell = p_{1/2}(s_1^\ell)(2 - 2\delta)^\ell \tag{151}$$

and

$$p_\delta(s_1^\ell) \geq \delta^\ell = \left(\frac{1 - \rho}{2}\right)^\ell = p_{1/2}(s_1^\ell)(1 - \rho)^\ell = p_{1/2}(s_1^\ell)(2\delta)^\ell. \tag{152}$$

It this holds that

$$(2\delta)^\ell \leq \frac{p_\delta(s_1^\ell)}{p_{1/2}(s_1^\ell)} \leq (2 - 2\delta)^\ell. \tag{153}$$

Substituting $\delta = \overline{\delta} = \frac{1 - \rho^k}{2}$, the upper bound on $\frac{p_{\overline\delta}(s_1^\ell)}{p_{1/2}(s_1^\ell)}$ is further upper bounded as

$$\left(1 + \rho^k\right)^\ell = \left(1 + (1 - 2\delta)^{\frac{\log(n)}{\delta}}\right)^\ell \tag{154}$$

$$\overset{(a)}{\leq} \left(1 + \frac{1}{n^2}\right)^\ell \tag{155}$$

$$\leq \left(1 + \frac{1}{n^2}\right)^n \tag{156}$$

$$\overset{(a)}{\leq} e^{1/n} \tag{157}$$

$$\overset{(b)}{\leq} 1 + \frac{2}{n}, \tag{158}$$

where both transitions denoted $(a)$ follow from $1 + x \leq e^x$ for $x \in \mathbb{R}$, and $(b)$ follows from $e^x \leq 1 + 2x$ for $x \in [0, 1]$. Similarly, the lower bound on $\frac{p_{\overline\delta}(s_1^\ell)}{p_{1/2}(s_1^\ell)}$ is further lower bounded as

$$\left(1 - \rho^k\right)^\ell = \left(1 - (1 - 2\delta)^{\frac{\log(n)}{\delta}}\right)^\ell \tag{159}$$

$$\overset{(a)}{\geq} \left(1 - \frac{1}{n^2}\right)^\ell \tag{160}$$

$$\geq \left(1 - \frac{1}{n^2}\right)^n \tag{161}$$

$$\overset{(b)}{\geq} 1 - \frac{1}{n}, \tag{162}$$

where $(a)$ follows again from $1 + x \leq e^x$ for $x \in \mathbb{R}$, and $(b)$ follows from Bernoulli's inequality $(1 - x)^r \geq 1 - rx$ for $x \in [0, 1]$ and $r \geq 1$. $\qquad\square$

Let $\tilde{P}_\theta^{(\ell)}$ denote the Gaussian mixture model (i.e., with flip probability $1/2$) corresponding to $\ell$ samples with means at $\pm\theta \in \mathbb{R}^d$, and note that it has the same $\ell$ marginal distributions as the genie-aided reduced model $\overline{P}_\theta^{(\ell)}$ (with flip probability $\overline{\delta}$). As discussed, the probability distributions $\tilde{P}_\theta^{(\ell)}$ and $\overline{P}_\theta^{(\ell)}$ are close. The following lemma provides a change-of-measure bound on the KL divergence for this case.

**Lemma 14.** *Let $U_1^\ell$ (resp. $V_1^\ell$) be distributed according to a probability distribution $P \equiv P_{U_1 \cdots U_\ell}$ (resp. $Q \equiv Q_{U_1 \cdots U_\ell}$) on $(\mathbb{R}^d)^\ell$, and let $\tilde{P} \equiv \tilde{P}_{U_1 \cdots U_\ell} := \prod_{i=1}^n P_{U_i}$ and $\tilde{Q} \equiv \tilde{Q}_{V_1 \cdots V_\ell} := \prod_{i=1}^n Q_{V_i}$ be the product distributions of their marginal distributions. Let*

$$\beta_P := \sup_{u_1^\ell \in (\mathbb{R}^d)^\ell} \left(\frac{P_{U_1 \cdots U_\ell}(u_1^\ell)}{\tilde{P}_{U_1 \cdots U_\ell}(u_1^\ell)}\right) \tag{163}$$

*and*

$$\beta_Q := \sup_{u_1^\ell \in (\mathbb{R}^d)^\ell} \left(\frac{\tilde{Q}_{V_1 \cdots V_\ell}(u_1^\ell)}{Q_{V_1 \cdots V_\ell}(u_1^\ell)}\right). \tag{164}$$

*Then,*

$$D_{KL}\left(P \parallel Q\right) \leq D_{KL}\left(\tilde{P} \parallel \tilde{Q}\right) + \log(\beta_P \cdot \beta_Q). \tag{165}$$

*Proof.* Since $\tilde{Q}$ is a product distribution, and since $P$ and $\tilde{P}$ have the same marginal distributions on $\mathbb{R}^d$ (at each time point) then

$$\int P_{U_1 \dots U_\ell}(u_1^\ell) \log\left(\frac{1}{\tilde{Q}_{V_1 \dots V_\ell}(u_1^\ell)}\right) \mathrm{d}u_1^\ell$$

$$= \sum_{i=1}^{\ell} \int P_{U_1 \dots U_\ell}(u_1^\ell) \log\left(\frac{1}{\tilde{Q}_{V_i}(u_i)}\right) \mathrm{d}u_1^\ell \tag{166}$$

$$= \sum_{i=1}^{\ell} \int P_{U_i}(u_i) \log\left(\frac{1}{\tilde{Q}_{V_i}(u_i)}\right) \mathrm{d}u_i \tag{167}$$

$$= \sum_{i=1}^{\ell} \int \tilde{P}_{U_i}(u_i) \log\left(\frac{1}{\tilde{Q}_{V_i}(u_i)}\right) \mathrm{d}u_i \tag{168}$$

$$= \sum_{i=1}^{\ell} \int \tilde{P}_{U_1 \dots U_\ell}(u_1^\ell) \log\left(\frac{1}{\tilde{Q}_{V_i}(u_i)}\right) \mathrm{d}u_1^\ell \tag{169}$$

$$= \int \tilde{P}_{U_1 \dots U_\ell}(u_1^\ell) \log\left(\frac{1}{\tilde{Q}_{V_1 \dots V_\ell}(u_1^\ell)}\right) \mathrm{d}u_1^\ell. \tag{170}$$

Using similar reasoning

$$\int \tilde{P}_{U_1 \dots U_\ell}(u_1^\ell) \log\left(\tilde{P}_{U_1 \dots U_\ell}(u_1^\ell)\right) \mathrm{d}u_1^\ell$$

$$= \sum_{i=1}^{\ell} \int \tilde{P}_{U_1 \dots U_\ell}(u_1^\ell) \log\left(\tilde{P}_{U_i}(u_i)\right) \mathrm{d}u_1^\ell \tag{171}$$

$$= \sum_{i=1}^{\ell} \int \tilde{P}_{U_i}(u_i) \log\left(\tilde{P}_{U_i}(u_i)\right) \mathrm{d}u_i \tag{172}$$

$$= \sum_{i=1}^{\ell} \int P_{U_i}(u_i) \log\left(\tilde{P}_{U_i}(u_i)\right) \mathrm{d}u_i \tag{173}$$

$$= \sum_{i=1}^{\ell} \int P_{U_1 \dots U_\ell}(u_1^\ell) \log\left(\tilde{P}_{U_i}(u_i)\right) \mathrm{d}u_1^\ell \tag{174}$$

$$= \int P_{U_1 \dots U_\ell}(u_1^\ell) \log\left(\tilde{P}_{U_1 \dots U_\ell}(u_1^\ell)\right) \mathrm{d}u_1^\ell \tag{175}$$

$$= \int P_{U_1 \dots U_\ell}(u_1^\ell) \log\left(P_{U_1 \dots U_\ell}(u_1^\ell)\right) \mathrm{d}u_1^\ell + \int P_{U_1 \dots U_\ell}(u_1^\ell) \log\left(\frac{\tilde{P}_{U_1 \dots U_\ell}(u_1^\ell)}{P_{U_1 \dots U_\ell}(u_1^\ell)}\right) \mathrm{d}u_1^\ell. \tag{176}$$

Combining (170) and (176) results the following bound

$$D_{KL}\left(P \parallel Q\right)$$

$$= \int P_{U_1 \dots U_\ell}(u_1^\ell) \log\left(\frac{P_{U_1 \dots U_\ell}(u_1^\ell)}{Q_{V_1 \dots V_\ell}(u_1^\ell)}\right) \mathrm{d}u_1^\ell \tag{177}$$

$$= \int P_{U_1 \dots U_\ell}(u_1^\ell) \log\left(\frac{P_{U_1 \dots U_\ell}(u_1^\ell)}{\tilde{Q}_{V_1 \dots V_\ell}(u_1^\ell)}\right) \mathrm{d}u_1^\ell + \int P_{U_1 \dots U_\ell}(u_1^\ell) \log\left(\frac{\tilde{Q}_{V_1 \dots V_\ell}(u_1^\ell)}{Q_{V_1 \dots V_\ell}(u_1^\ell)}\right) \mathrm{d}u_1^\ell \tag{178}$$

$$= \int \tilde{P}_{U_1 \dots U_\ell}(u_1^\ell) \log\left(\tilde{P}_{U_1 \dots U_\ell}(u_1^\ell)\right) \mathrm{d}u_1^\ell - \int P_{U_1 \dots U_\ell}(u_1^\ell) \log\left(\frac{\tilde{P}_{U_1 \dots U_\ell}(u_1^\ell)}{P_{U_1 \dots U_\ell}(u_1^\ell)}\right) \mathrm{d}u_1^\ell$$

$$+ \int P_{U_1 \cdots U_\ell}(u_1^\ell) \log \left( \frac{1}{\tilde{Q}_{V_1 \cdots V_\ell}(u_1^\ell)} \right) \mathrm{d} u_1^\ell + \int P_{U_1 \cdots U_\ell}(u_1^\ell) \log \left( \frac{\tilde{Q}_{V_1 \cdots V_\ell}(u_1^\ell)}{Q_{V_1 \cdots V_\ell}(u_1^\ell)} \right) \mathrm{d} u_1^\ell \quad (179)$$

$$\leq \mathrm{D}_{\mathrm{KL}} \left( \tilde{P} \parallel \tilde{Q} \right) + \sup_{u_1^\ell \in (\mathbb{R}^d)^\ell} \log \left( \frac{P_{U_1 \cdots U_\ell}(u_1^\ell)}{\tilde{P}_{U_1 \cdots U_\ell}(u_1^\ell)} \right) + \sup_{u_1^\ell \in (\mathbb{R}^d)^\ell} \log \left( \frac{\tilde{Q}_{V_1 \cdots V_\ell}(u_1^\ell)}{Q_{V_1 \cdots V_\ell}(u_1^\ell)} \right), \quad (180)$$

as claimed by the lemma. $\qquad \square$

**Third step (bound on the chi-square divergence):** Let $\varphi(y; \theta, \sigma^2)$ be the Gaussian PDF with mean $\theta \in \mathbb{R}^d$ and covariance matrix $\sigma^2 \cdot I_d$. The next lemma bounds the chi-square divergence between a pair of such distributions with different means. Originally, a bound of this order was established in [Wu and Zhou, 2019, proof of Lemma 27] on the KL divergence, by splitting the first coordinate (which is assumed, w.l.o.g., to contain the signal) and the other $d - 1$ coordinates, using the chain rule, and then bounding each of the two KL terms with the corresponding chi-square divergence. Here we provide a direct upper bound on the chi-square divergence, which does not rely on splitting between the coordinates of the mean vector, and which might be of independent use in future works.

**Lemma 15.** *Let* $\tilde{P}_\theta = \frac{1}{2} N(\theta, \sigma^2 \cdot I_d) + \frac{1}{2} N(-\theta, \sigma^2 \cdot I_d)$ *be a balanced Gaussian mixture with means at* $\pm \theta \in \mathbb{R}^d$. *Then, if* $\|\theta_0\| = \|\theta_1\| = t \leq \sigma$, *then*

$$\chi^2(\tilde{P}_{\theta_1} \parallel \tilde{P}_{\theta_0}) \leq \frac{8t^2}{\sigma^4} \cdot \|\theta_0 - \theta_1\|^2. \quad (181)$$

*Proof.* For any $y \in \mathbb{R}^d$,

$$\tilde{P}_\theta(y) = \frac{1}{2} \frac{1}{(2\pi\sigma^2)^{d/2}} e^{-\frac{\|y - \theta\|^2}{2\sigma^2}} + \frac{1}{2} \frac{1}{(2\pi\sigma^2)^{d/2}} e^{-\frac{\|y + \theta\|^2}{2\sigma^2}} \quad (182)$$

$$= \varphi(y; 0, \sigma^2) \cdot e^{-\frac{\|\theta\|^2}{2\sigma^2}} \cdot \cosh \left( \frac{\theta^\top y}{\sigma^2} \right). \quad (183)$$

Then,

$$\chi^2 \left( \tilde{P}_{\theta_1} \parallel \tilde{P}_{\theta_0} \right)$$

$$= \int \frac{\left[ \varphi(y; 0, \sigma^2) e^{-\frac{\|\theta_1\|^2}{2\sigma^2}} \cdot \cosh \left( \frac{\theta_1^\top y}{\sigma^2} \right) - \varphi(y; 0, \sigma^2) e^{-\frac{\|\theta_0\|^2}{2\sigma^2}} \cdot \cosh \left( \frac{\theta_0^\top y}{\sigma^2} \right) \right]^2}{\varphi(y; 0, \sigma^2) \cdot e^{-\frac{\|\theta_0\|^2}{2\sigma^2}} \cosh \left( \frac{\theta_0^\top y}{\sigma^2} \right)} \cdot \mathrm{d}y \quad (184)$$

$$= e^{\frac{\|\theta_0\|^2}{2\sigma^2}} \int \varphi(y; 0, \sigma^2) \frac{\left[ e^{-\frac{\|\theta_1\|^2}{2\sigma^2}} \cdot \cosh \left( \frac{\theta_1^\top y}{\sigma^2} \right) - e^{-\frac{\|\theta_0\|^2}{2\sigma^2}} \cdot \cosh \left( \frac{\theta_0^\top y}{\sigma^2} \right) \right]^2}{\cosh \left( \frac{\theta_0^\top y}{\sigma^2} \right)} \cdot \mathrm{d}y \quad (185)$$

$$\overset{(a)}{\leq} e^{\frac{\|\theta_0\|^2}{2\sigma^2}} \cdot \int \varphi(y; 0, \sigma^2) \left[ e^{-\frac{\|\theta_1\|^2}{2\sigma^2}} \cdot \cosh \left( \frac{\theta_1^\top y}{\sigma^2} \right) - e^{-\frac{\|\theta_0\|^2}{2\sigma^2}} \cdot \cosh \left( \frac{\theta_0^\top y}{\sigma^2} \right) \right]^2 \cdot \mathrm{d}y \quad (186)$$

$$= e^{\frac{\|\theta_0\|^2}{2\sigma^2}} \cdot \int \varphi(y; 0, \sigma^2) e^{-\frac{\|\theta_1\|^2}{\sigma^2}} \cdot \cosh^2 \left( \frac{\theta_1^\top y}{\sigma^2} \right) \cdot \mathrm{d}y$$

$$- 2 e^{\frac{\|\theta_0\|^2}{2\sigma^2}} \cdot \int \varphi(y; 0, \sigma^2) e^{-\frac{\|\theta_1\|^2 + \|\theta_0\|^2}{2\sigma^2}} \cosh \left( \frac{\theta_1^\top y}{\sigma^2} \right) \cosh \left( \frac{\theta_0^\top y}{\sigma^2} \right) \cdot \mathrm{d}y$$

$$+ e^{\frac{\|\theta_0\|^2}{2\sigma^2}} \cdot \int \varphi(y; 0, \sigma^2) e^{-\frac{\|\theta_0\|^2}{\sigma^2}} \cdot \cosh^2 \left( \frac{\theta_0^\top y}{\sigma^2} \right) \cdot \mathrm{d}y, \quad (187)$$

where $(a)$ follows since $\cosh(x) \geq 1$ for all $x \in \mathbb{R}$. We next evaluate the integral for each of the terms. First, for any $y \in \mathbb{R}^d$,

$$\int \varphi(y; 0, \sigma^2) \cosh^2 \left( \frac{\theta_1^\top y}{\sigma^2} \right) \mathrm{d}y$$

$$\stackrel{(a)}{=} \int \varphi(y; 0, 1) \cdot \cosh^2 \left( \frac{\theta_1^\top y}{\sigma} \right) \mathrm{d}y \tag{188}$$

$$\stackrel{(b)}{=} \int \varphi(t; 0, 1) \cdot \cosh^2 \left( \frac{\|\theta_1\|}{\sigma} t \right) \mathrm{d}t \tag{189}$$

$$= \int \varphi(t; 0, 1) \left[ \frac{\exp\left( \frac{\|\theta_1\|}{\sigma} t \right) + \exp\left( -\frac{\|\theta_1\|}{\sigma} t \right)}{2} \right]^2 \mathrm{d}t \tag{190}$$

$$= \int \varphi(t; 0, 1) \frac{\exp\left( 2\frac{\|\theta_1\|}{\sigma} t \right) + 2 + \exp\left( -2\frac{\|\theta_1\|}{\sigma} t \right)}{4} \mathrm{d}t \tag{191}$$

$$\stackrel{(c)}{=} \frac{1}{2} + \frac{1}{2} \cdot e^{2\frac{\|\theta_1\|^2}{\sigma^2}} \tag{192}$$

$$= e^{\frac{\|\theta_1\|^2}{\sigma^2}} \cosh\left( \frac{\|\theta_1\|^2}{\sigma^2} \right), \tag{193}$$

where $(a)$ follows from the change of variables $y \to \frac{y}{\sigma}$, $(b)$ follows from the rotational invariance of the Gaussian PDF $\varphi(y; 0, \sigma^2)$, we may assume that $\theta_1 = (\|\theta_1\|, 0, 0, \dots, 0)$, setting $t \in \mathbb{R}$ to be the first coordinate of $y$, and integrating over all other $d - 1$ coordinates, and $(c)$ follows from the Gaussian moment-generating function formula. The third term in the integral is similarly evaluated. For the second term,

$$\int \varphi(y; 0, \sigma^2) \left[ 2 \cosh\left( \frac{\theta_1^\top y}{\sigma^2} \right) \cosh\left( \frac{\theta_0^\top y}{\sigma^2} \right) \right] \cdot \mathrm{d}y$$

$$\stackrel{(a)}{=} \int \varphi(y; 0, 1) \left[ 2 \cosh\left( \frac{\theta_1^\top y}{\sigma} \right) \cosh\left( \frac{\theta_0^\top y}{\sigma} \right) \right] \cdot \mathrm{d}y \tag{194}$$

$$\stackrel{(b)}{=} \int \varphi(y; 0, 1) \cosh\left( \frac{(\theta_1 + \theta_0)^\top y}{\sigma} \right) \cdot \mathrm{d}y + \int \varphi(y; 0, 1) \cosh\left( \frac{(\theta_1 - \theta_0)^\top y}{\sigma} \right) \cdot \mathrm{d}y \tag{195}$$

$$\stackrel{(c)}{=} \int \varphi(t; 0, 1) \cosh\left( \frac{\|\theta_1 + \theta_0\| t}{\sigma} \right) \cdot \mathrm{d}t + \int \varphi(t; 0, 1) \cosh\left( \frac{\|\theta_1 - \theta_0\| t}{\sigma} \right) \cdot \mathrm{d}t \tag{196}$$

$$\stackrel{(d)}{=} e^{\frac{\|\theta_1 + \theta_0\|^2}{2\sigma^2}} + e^{\frac{\|\theta_1 - \theta_0\|^2}{2\sigma^2}} \tag{197}$$

$$= e^{\frac{\|\theta_0\|^2 + \|\theta_1\|^2}{2\sigma^2}} \cdot \left[ e^{\frac{\theta_1^\top \theta_0}{\sigma^2}} + e^{\frac{-\theta_1^\top \theta_0}{\sigma^2}} \right] \tag{198}$$

$$= e^{\frac{\|\theta_0\|^2 + \|\theta_1\|^2}{2\sigma^2}} \cdot 2 \cosh\left( \frac{\theta_1^\top \theta_0}{\sigma^2} \right). \tag{199}$$

where $(a)$ follows from the change of variables $y \to \frac{y}{\sigma}$, $(b)$ follows from the identity $2 \cosh(x) \cosh(y) = \cosh(x + y) + \cosh(x - y)$, and $(c)$ follows from rotational invariance, and $(d)$ follows from

$$\int \varphi(t; 0, 1) \cosh(at) \mathrm{d}t = \int \varphi(t; 0, 1) \frac{e^{at} + e^{-at}}{2} \mathrm{d}t = e^{\frac{a^2}{2}}. \tag{200}$$

Continuing (187), we thus have

$$\chi^2\left( \tilde{P}_{\theta_1} \,\|\, \tilde{P}_{\theta_0} \right) \le e^{\frac{\|\theta_0\|^2}{2\sigma^2}} \cdot \left[ \cosh\left( \frac{\|\theta_1\|^2}{\sigma^2} \right) - 2 \cosh\left( \frac{\theta_1^\top \theta_0}{\sigma^2} \right) + \cosh\left( \frac{\|\theta_0\|^2}{\sigma^2} \right) \right]. \tag{201}$$

Now, $\frac{\mathrm{d}}{\mathrm{d}x} \cosh(x) = \sinh(x)$ and $\frac{\mathrm{d}^2}{\mathrm{d}x^2} \cosh(x) = \cosh(x)$. In addition, if $0 \le x \le 1$ then it can be easily verified that $\sinh(x) \le 2x$. Thus, if $0 \le x \le y \le 1$ then

$$\cosh(y) - \cosh(x) = \int_x^y \frac{\mathrm{d}}{\mathrm{d}r} \cosh(r) \mathrm{d}r = \int_x^y \sinh(r) \mathrm{d}r \le \int_x^y 2r \mathrm{d}r = y^2 - x^2. \tag{202}$$

Moreover, $e^x \le 1 + 2x$ for $x \in [0, 1]$. Thus, if we assume $\|\theta_0\| = \|\theta_1\| \le \sigma$ we may further upper bound (201) as

$$\chi^2\left( \tilde{P}_{\theta_1} \,\|\, \tilde{P}_{\theta_0} \right)$$

$$\le e^{\frac{\|\theta_0\|^2}{2\sigma^2}} \cdot \left[\cosh\left(\frac{\|\theta_0\|^2}{\sigma^2}\right) - \cosh\left(\frac{|\theta_1^\top \theta_0|}{\sigma^2}\right) + \cosh\left(\frac{\|\theta_0\|^2}{\sigma^2}\right) - \cosh\left(\frac{|\theta_1^\top \theta_0|}{\sigma^2}\right)\right] \tag{203}$$

$$\overset{(a)}{\le} 2 \cdot \left[\cosh\left(\frac{\|\theta_0\|^2}{\sigma^2}\right) - \cosh\left(\frac{|\theta_1^\top \theta_0|}{\sigma^2}\right) + \cosh\left(\frac{\|\theta_0\|^2}{\sigma^2}\right) - \cosh\left(\frac{|\theta_1^\top \theta_0|}{\sigma^2}\right)\right] \tag{204}$$

$$\overset{(b)}{\le} 2 \cdot \frac{\|\theta_1\|^4 - (\theta_1^\top \theta_0)^2 + \|\theta_0\|^4 - (\theta_1^\top \theta_0)^2}{\sigma^4}, \tag{205}$$

where $(a)$ follows since $e^{\frac{\|\theta_0\|^2}{2\sigma^2}} \le 2$ under the assumption $\|\theta_0\|^2 \le \sigma^2$, $(b)$ follows from (202). Now, the numerator of (205) is further upper bounded as

$$\|\theta_1\|^4 - 2(\theta_1^\top \theta_0)^2 + \|\theta_0\|^4 \tag{206}$$

$$= (\theta_1^\top \theta_1)^2 - (\theta_1^\top \theta_0)(\theta_0^\top \theta_1) - (\theta_0^\top \theta_1)(\theta_1^\top \theta_0) + (\theta_0^\top \theta_0)^2 \tag{207}$$

$$= \mathrm{Tr}\left[\theta_0\theta_0^\top \theta_0\theta_0^\top - \theta_0\theta_0^\top \theta_1\theta_1^\top - \theta_1\theta_1^\top \theta_0\theta_0^\top + \theta_1\theta_1^\top \theta_1\theta_1^\top\right] \tag{208}$$

$$= \mathrm{Tr}[(\theta_0\theta_0^\top - \theta_1\theta_1^\top)^2] \tag{209}$$

$$= \|\theta_0\theta_0^\top - \theta_1\theta_1^\top\|_F^2 \tag{210}$$

$$= \|\theta_0\theta_0^\top - \theta_0\theta_1^\top + \theta_0\theta_1^\top - \theta_1\theta_1^\top\|_F^2 \tag{211}$$

$$\le 2\|\theta_0\theta_0^\top - \theta_0\theta_1^\top\|_F^2 + 2\|\theta_0\theta_1^\top - \theta_1\theta_1^\top\|_F^2 \tag{212}$$

$$= 2\|\theta_0\|^2 \cdot \|\theta_0 - \theta_1\|^2 + 2\|\theta_1\|^2 \cdot \|\theta_0 - \theta_1\|^2, \tag{213}$$

where the inequality follows from $\|A + B\|_F^2 \le 2\|A\|_F^2 + 2\|B\|_F^2$. Inserting this bound into (205), and using $\|\theta_0\| = \|\theta_1\| = t$ results the bound (181). $\qquad\square$

With the results of the three steps at hand, we may complete the proof of Theorem 2.

*Proof of Theorem 2.* Recall that we assume the low-dimension regime $3 \le d \le \delta n$. The condition $d \ge 3$ can be relaxed to $d \ge 2$, as promised in Theorem 2, using a different construction of the packing set. We leave this refinement to the end of this section. If $t \ge \sqrt{\delta}$ then the lower bound for the Gaussian location model (5) implies a bound of $\Theta(\sqrt{\frac{d}{n}})$. This can be verified by separately checking the only two cases $t \ge \sqrt{\frac{d}{n}} \ge \sqrt{\delta}$ and $t \ge \sqrt{\delta} \ge \sqrt{\frac{d}{n}}$ possible in the low dimension regime. We thus henceforth may concentrate on the regime $t \le \sqrt{\delta}$. To continue, we henceforth assume the slightly stronger requirement $t \le \sqrt{\frac{\delta}{\log n}} = \sqrt{\frac{1}{k}}$, where $\frac{1}{k}$ is the variance in the genie-aided reduced model.

Let $\overline{\delta} = \frac{1 - \rho^k}{2}$ and recall that $\overline{P}_{\theta_m}^{(\ell)}$ is the Gaussian model with Markovian signs with flip probability $\overline{\delta}$. Further let $\varphi(y_1^\ell; \mu)$ be the Gaussian PDF for $\ell$ samples from a $d_0$-dimensional model with mean $\mu \in (\mathbb{R}^{d_0})^\ell$ and covariance matrix $\Sigma = \frac{1}{k}I_{d_0} \otimes I_\ell \in \mathbb{R}_+^{d_0\ell \times d_0\ell}$. With this notation, it holds that

$$\overline{P}_{\theta_m}^{(\ell)}(y_1^\ell) = \sum_{r_1^\ell \in \{-1,1\}^\ell} p_{\overline{\delta}}(r_1^\ell) \cdot \varphi(y_1^\ell; r_1^\ell \otimes \theta_m). \tag{214}$$

Similarly, let $\tilde{P}_{\theta_m}^{(\ell)}$ be the Gaussian model with Markovian signs with flip probability $1/2$, that is, a Gaussian mixture model (which is, in fact, memoryless),

$$\tilde{P}_{\theta_m}^{(\ell)}(y_1^\ell) = \sum_{r_1^\ell \in \{-1,1\}^\ell} p_{1/2}(r_1^\ell) \cdot \varphi(y_1^\ell; r_1^\ell \otimes \theta_m) \tag{215}$$

$$= \prod_{i=1}^\ell \left[\frac{1}{2}\varphi(y_i; \theta_m) + \frac{1}{2}\varphi(y_i; -\theta_m)\right]. \tag{216}$$

Now, Lemma 13 implies that

$$1 - \frac{1}{n} \le \min_{r_1^\ell \in \{-1,1\}^\ell} \frac{p_{\overline{\delta}}(r_1^\ell)}{p_{1/2}(r_1^\ell)} \le \frac{\overline{P}_{\theta_m}^{(\ell)}(y_1^\ell)}{\tilde{P}_{\theta_m}^{(\ell)}(y_1^\ell)} \le \max_{r_1^\ell \in \{-1,1\}^\ell} \frac{p_{\overline{\delta}}(r_1^\ell)}{p_{1/2}(r_1^\ell)} \le 1 + \frac{2}{n}, \tag{217}$$

and hence

$$\mathrm{D_{KL}}\left(\overline{P}_{\theta_m}^{(\ell)} \;\|\; \overline{P}_{\theta_0}^{(\ell)}\right) \overset{(a)}{\leq} \mathrm{D_{KL}}\left(\tilde{P}_{\theta_m}^{(\ell)} \;\|\; \tilde{P}_{\theta_0}^{(\ell)}\right) + \log\left[\left(1 + \frac{2}{n}\right)\left(\frac{1}{1 - \frac{1}{n}}\right)\right] \tag{218}$$

$$\overset{(b)}{\leq} \mathrm{D_{KL}}\left(\tilde{P}_{\theta_m}^{(\ell)} \;\|\; \tilde{P}_{\theta_0}^{(\ell)}\right) + 2\log\left(1 + \frac{2}{n}\right) \tag{219}$$

$$\leq \mathrm{D_{KL}}\left(\tilde{P}_{\theta_m}^{(\ell)} \;\|\; \tilde{P}_{\theta_0}^{(\ell)}\right) + \frac{4}{n} \tag{220}$$

$$\overset{(c)}{=} \ell \cdot \mathrm{D_{KL}}\left(\tilde{P}_{\theta_m} \;\|\; \tilde{P}_{\theta_0}\right) + \frac{4}{n} \tag{221}$$

$$\overset{(d)}{\leq} \ell \cdot \chi^2\left(\tilde{P}_{\theta_m} \;\|\; \tilde{P}_{\theta_0}\right) + \frac{4}{n} \tag{222}$$

$$\overset{(e)}{\leq} 8\frac{\ell t^2}{\sigma^4}\cdot\|\theta_0 - \theta_1\|^2 + \frac{4}{n} \tag{223}$$

$$\overset{(f)}{\leq} 32\log(n)\cdot\frac{nt^4\epsilon^2}{\delta} + \frac{4}{n} \tag{224}$$

where $(a)$ follows from Lemma 14, $(b)$ follows since $\frac{1}{1-\frac{1}{n}} \leq 1 + \frac{2}{n}$ for $n \geq 2$, $(c)$ follows from the tensorization property of the KL divergence, $(d)$ follows from the fact (cf. [Tsybakov, 2008, Eq. (2.27)]) that $\mathrm{D_{KL}}(P \;\|\; Q) \leq \chi^2(P \;\|\; Q)$ for any pair of probability measures $P$ and $Q$, $(e)$ follows from Lemma 15, and $(f)$ follows from property (iv) of the packing set $\|\theta_0 - \theta_1\|^2 \leq 4t^2\epsilon^2$ and $\sigma^2 = \frac{1}{k} = \frac{\delta}{\log n}$ and $\ell = \frac{n}{k}$.

Recall that from Fano's argument (143), the largest $\epsilon > 0$ so that $\mathrm{D_{KL}}(\overline{P}_{\theta_m}^{(\ell)} \;\|\; \overline{P}_{\theta_0}^{(\ell)}) \leq \frac{d}{16}$ assures the bound $\mathsf{M}(n, d, \delta, t) \geq \frac{\epsilon t}{128}$. Assuming that $\frac{4}{n} \leq \frac{d}{32}$, that is $n \geq \frac{128}{d}$, this will occur if $32\log(n)\cdot\frac{nt^4\epsilon^2}{\delta} \leq \frac{d}{32}$. This can be achieved by the choice $\epsilon = \frac{1}{32\sqrt{\log(n)}}\cdot\min\left\{1, \frac{1}{t^2}\sqrt{\frac{d\delta}{n}}\right\}$. Using this value in the Fano's based bound then completes the proof of the theorem. $\qquad\square$

**Relaxing $d \geq 3$ to $d \geq 2$** In the above proof, we assumed that $d \geq 3$. This condition can be relaxed to $d \geq 2$ by using a more careful construction of the packing set.

**Lemma 16.** *Let $d \geq 2$ be an integer and $\alpha \in \left(0, \frac{\pi}{2}\right]$ be an angle. Then there exists a $\left(2\sin\left(\frac{\alpha}{2}\right)\right)$-packing on $\mathbb{S}^{d-1}$ of size at least $\frac{\cos(\alpha)}{\sin^{d-1}(\alpha)}$.*

*Proof.* We will greedily construct the desired packing set. Let $\Delta = 2\sin\left(\frac{\alpha}{2}\right)$. Start with an arbitrary point on $\mathbb{S}^{d-1}$. In each of the following steps, put into the packing set another arbitrary point that is $\Delta$-far from any existing points in the packing set and their antipodal points. Repeat this process until no more points can be put without violating the distance guarantee. Note that this construction guarantees that for any pair of distinct points $\theta, \theta'$ in the packing set,

$$\mathsf{loss}(\theta, \theta') = \min\{\|\theta - \theta'\|, \|\theta + \theta'\|\} > 2\sin\left(\frac{\alpha}{2}\right) = \Delta. \tag{225}$$

We then lower bound the cardinality of the above packing set. For $\alpha \in [0, \frac{\pi}{2}]$, let $S(\alpha)$ denote the surface area of a spherical cap on $\mathbb{S}^{d-1}$ of angular radius $\alpha$. For $\theta \in \mathbb{S}^{d-1}$ and $\alpha \in [0, \frac{\pi}{2}]$, let $\mathbb{K}(\theta, \alpha) := \{\theta' \in \mathbb{S}^{d-1} : |\langle\theta, \theta'\rangle| \geq \cos(\alpha)\}$ denote the union of two spherical caps centered around $\theta$ and $-\theta$, respectively, of angular radius $\alpha$ each. Note that each new point $\theta$ in the construction induces a *forbidden region* on $\mathbb{S}^{d-1}$ of surface area at most $|\mathbb{K}(\theta, \alpha)| = 2S(\alpha)$ in which following points cannot lie, since any point within $\mathbb{K}(\theta, \alpha)$ has distance at most $2\sin\left(\frac{\alpha}{2}\right) = \Delta$ to either $\theta$ or $-\theta$. Also, the surface area of the forbidden region is *upper* bounded by $2S(\alpha)$ since $\mathbb{K}(\theta, \alpha)$ may overlap with $\mathbb{K}(\theta', \alpha)$ for some previous $\theta'$. Therefore, by the step at which the greedy construction terminates, one must have put at least $\frac{2S(\frac{\pi}{2})}{2S(\alpha)} = \frac{S(\frac{\pi}{2})}{S(\alpha)}$ many points, where the numerator is nothing but the surface area of $\mathbb{S}^{d-1}$. [Blachman and Few, 1963, Eq. (1)] upper bounds the area ratio between

two spherical caps on $\mathbb{S}^{d-1}$ as follows:

$$\frac{S(\alpha)}{S(\beta)} = \frac{\int_0^\alpha \sin^{d-2}(x)\mathrm{d}x}{\int_0^\beta \sin^{d-2}(x)\mathrm{d}x} \leq \sec(\alpha) \cdot \frac{\sin^{d-1}(\alpha)}{\sin^{d-1}(\beta)}. \tag{226}$$

Using this bound, we conclude that the cardinality of the constructed packing set is at least $\frac{\cos(\alpha)}{\sin^{d-1}(\alpha)}$, which finishes the proof. $\qquad\square$

By Lemma 16, a $\Delta$-packing set can be obtained by setting $2\sin\left(\frac{\alpha}{2}\right) = \Delta$. Therefore,

$$\sin\left(\frac{\alpha}{2}\right) = \frac{\Delta}{2}, \quad \cos\left(\frac{\alpha}{2}\right) = \sqrt{1 - \frac{\Delta^2}{4}}, \tag{227}$$

$$\sin(\alpha) = 2\sin\left(\frac{\alpha}{2}\right)\cos\left(\frac{\alpha}{2}\right) = \Delta\sqrt{1 - \frac{\Delta^2}{4}}, \quad \cos(\alpha) = 1 - 2\sin^2\left(\frac{\alpha}{2}\right) = 1 - \frac{\Delta^2}{2}, \tag{228}$$

and the cardinality of the packing set is at least

$$\left(1 - \frac{\Delta^2}{2}\right)\left(\Delta\sqrt{1 - \frac{\Delta^2}{4}}\right)^{-(d-1)}. \tag{229}$$

Setting $\Delta\sqrt{1 - \frac{\Delta^2}{4}} = \frac{1}{8}$, we have $\Delta = \frac{1}{2}\sqrt{8 - 3\sqrt{7}} \geq \frac{1}{8}$ and we get a $\frac{1}{8}$-packing set $\Theta_M = \{\theta_m\}_{m\in[M]}$ of size at least $M \geq \frac{3\sqrt{7}}{8} \cdot 8^{-(d-1)} \geq 0.99 \cdot 8^{-(d-1)}$.

The above construction can be used in place of the one described at the beginning of Appendix B.2. The resulting lower bound on the minimax error rate follows from similar reasoning with suitably adjusted numerical constants. We sketch the rest of the proof below. By Fano's method [Wainwright, 2019, Proposition 15.2] (see also (140)),

$$\mathsf{M}(n, d, \delta, t) \geq \frac{\epsilon t}{16}\left[1 - \frac{I(J; X_1^n) + \log 2}{\log M}\right] \tag{230}$$

$$\geq \frac{\epsilon t}{16}\left[1 - \frac{\max_{m\in[M]} \mathrm{D_{KL}}(P_{\theta_m}^{(n)} \,\|\, P_{\theta_0}^{(n)}) + \log 2}{(d-1)\log 8 + \log 0.99}\right] \tag{231}$$

$$\geq \frac{\epsilon t}{16}\left[\frac{1}{2} - 4 \cdot \frac{\max_{m\in[M]} \mathrm{D_{KL}}(P_{\theta_m}^{(n)} \,\|\, P_{\theta_0}^{(n)})}{d}\right], \tag{232}$$

where the last inequality follows since $d \geq 2$ implies (i) $(d-1)\log 8 + \log 0.99 \geq \frac{d}{4}$, (ii) $\frac{\log 2}{(d-1)\log 8 + \log 0.99} \leq \frac{\log 2}{\log 8 + \log 0.99} \leq \frac{1}{2}$. The proof of the bound $\max_{m\in[M]} \mathrm{D_{KL}}(P_{\theta_m}^{(n)} \,\|\, P_{\theta_0}^{(n)}) \leq \frac{d}{16}$ can be completely reused which implies $\mathsf{M}(n, d, \delta, t) \geq \frac{\epsilon t}{64}$. The same choice of $\epsilon$ then yields the desired lower bound on the minimax error rate in Theorem 2 for any $d \geq 2$.

# C Proofs for Section 3: Estimation of $\delta$ for a given estimate of $\theta_*$

## C.1 Proof of Theorem 4: Analysis of the estimator

*Proof of Theorem 4.* We first note that since the estimator (19) does not exploit the correlation between $X_i$ and $X_j$ for $|i-j| \geq 2$, we may assume that the $n/2$ pairs of random variables $(S_{2i-1}, S_{2i})$ are independent by multiplying each pair of samples $(X_{2i-1}, X_{2i})$ by an i.i.d. random sign. By explicitly using $X_i = S_i\theta_* + Z_i$ in the definition of $\hat{\rho}$, and using the triangle inequality, we obtain that

$$|\hat{\rho} - \rho| \leq \left|\frac{\|\theta_*\|^2}{\|\theta_\sharp\|^2}\frac{2}{n}\sum_{i=1}^{n/2} S_{2i}S_{2i-1} - \rho\right| + \left|\frac{2}{\|\theta_\sharp\|^2 n}\sum_{i=1}^{n/2} S_{2i}\theta_*^\top Z_{2i-1}\right|$$

$$+ \left|\frac{2}{\|\theta_\sharp\|^2 n}\sum_{i=1}^{n/2} S_{2i-1}\theta_*^\top Z_{2i}\right| + \left|\frac{2}{\|\theta_\sharp\|^2 n}\sum_{i=1}^{n/2} Z_{2i}^\top Z_{2i-1}\right|. \tag{233}$$

We begin with the analysis of the first term. We note that

$$\frac{2}{n}\sum_{i=1}^{n/2} S_{2i}S_{2i-1} \stackrel{d}{=} \frac{2}{n}\sum_{i=1}^{n/2} R_i \tag{234}$$

where

$$R_i = \begin{cases} 1, & \text{w.p. } \frac{1+\rho}{2} \\ -1, & \text{w.p. } \frac{1-\rho}{2} \end{cases} \tag{235}$$

and $\{R_i\}_{i\in[n/2]}$ are i.i.d. with $\mathbb{E}[R_i] = 1 - 2\delta = \rho$ and $\mathbb{V}[R_i] = 4\delta(1-\delta) \leq 4\delta$. Now, the first term in (233) is bounded by

$$\left| \frac{\|\theta_*\|^2}{\|\theta_\sharp\|^2} \frac{2}{n}\sum_{i=1}^{n/2} R_i - \rho \right|$$

$$= \left| \left(1 + \frac{\|\theta_*\|^2 - \|\theta_\sharp\|^2}{\|\theta_\sharp\|^2}\right) \frac{2}{n}\sum_{i=1}^{n/2} R_i - \rho \right| \tag{236}$$

$$\leq \left| \frac{2}{n}\sum_{i=1}^{n/2} R_i - \rho \right| + \frac{\left|\|\theta_*\|^2 - \|\theta_\sharp\|^2\right|}{\|\theta_\sharp\|^2}, \tag{237}$$

where the last inequality follows since $\left|\frac{2}{n}\sum_{i=1}^{n/2} R_i\right| \leq 1$. In this last equation, the last term can be considered a bias of the estimator due to the mismatch between $\theta_*$ and $\theta_\sharp$. Now, by Bernstein's inequality for a sum of independent, zero-mean, bounded random variables $R_i - \mathbb{E}[R_i] \in [-2, 2]$ (cf. (323) from [Wainwright, 2019, Propostion 2.14])

$$\mathbb{P}\left[\left|\frac{2}{n}\sum_{i=1}^{n/2} R_i - \mathbb{E}[R_i]\right| \geq t\right] \leq 2\exp\left(-\frac{\frac{n}{2}t^2}{2\left(4\delta + \frac{2t}{3}\right)}\right), \tag{238}$$

and so with probability larger than $1 - \epsilon$

$$\left|\frac{2}{n}\sum_{i=1}^{n/2} R_i - \mathbb{E}[R_i]\right| \leq \frac{4}{3n}\log\left(\frac{2}{\epsilon}\right) + \frac{4}{3n}\sqrt{9n\delta\log\left(\frac{2}{\epsilon}\right) + \log^2\left(\frac{2}{\epsilon}\right)} \tag{239}$$

$$\leq \sqrt{\frac{16\delta\log\left(\frac{2}{\epsilon}\right)}{n}} + \frac{8\log\left(\frac{2}{\epsilon}\right)}{3n} \tag{240}$$

$$\leq 7\log\left(\frac{2}{\epsilon}\right)\sqrt{\frac{\delta}{n}}, \tag{241}$$

since $\delta \geq \frac{1}{n}$.

The second term in (233) (and similarly, the third term in (233)) is

$$\frac{2}{\|\theta_\sharp\|^2 n}\sum_{i=1}^{n/2} S_{2i}\theta_\sharp^\top Z_{2i-1} \sim N\left(0, \frac{2}{\|\theta_\sharp\|^2 n}\right). \tag{242}$$

Thus, by the standard Chernoff bound for Gaussian random variables

$$\mathbb{P}\left[\left|\frac{2}{\|\theta_\sharp\|^2 n}\sum_{i=1}^{n/2} S_{2i}\theta_\sharp^\top Z_{2i-1}\right| > t\right] \leq 2e^{-\frac{\|\theta_\sharp\|^2 n t^2}{4}}, \tag{243}$$

and so with probability larger than $1 - \epsilon$

$$\left|\frac{2}{\|\theta_\sharp\|^2 n}\sum_{i=1}^{n/2} S_{2i}\theta_\sharp^\top Z_{2i-1}\right| \leq \sqrt{\frac{4}{n\|\theta_\sharp\|^2}\log\left(\frac{2}{\epsilon}\right)}. \tag{244}$$

The fourth term in (233) satisfies

$$\frac{2}{\|\theta_\sharp\|^2 n} \sum_{i=1}^{n/2} Z_{2i}^\top Z_{2i-1} \overset{d}{=} \frac{2}{\|\theta_\sharp\|^2 n} \sum_{i=1}^{nd/2} W_i \tilde{W}_i \tag{245}$$

where $\{W_i\}_{i\in[nd/2]}$ and $\{\tilde{W}_i\}_{i\in[nd/2]}$ are i.i.d. and $W_i, \tilde{W}_i \sim N(0,1)$ are independent. It is well known [Vershynin, 2018, Lemma 2.7.7] that the product of two subGaussian random variables (even if they are not independent) is sub-exponential. Here, we have a simpler and exact characterization. Letting

$$W_i \tilde{W}_i = \left(\frac{W_i + \tilde{W}_i}{2}\right)^2 - \left(\frac{W_i - \tilde{W}_i}{2}\right)^2 \overset{d}{=} V_i^2 - \tilde{V}_i^2 \tag{246}$$

and since $V_i = \frac{1}{2}(W_i + \tilde{W}_i) \sim N(0,1)$ and $\tilde{V}_i = \frac{1}{2}(W_i - \tilde{W}_i) \sim N(0,1)$ are uncorrelated, they are independent. Letting $\chi^2_{nd/2}$ and $\tilde{\chi}^2_{nd/2}$ be a pair of independent chi-square random variables with $nd/2$ degrees of freedom, it then holds that

$$\frac{2}{\|\theta_\sharp\|^2 n} \sum_{i=1}^{n/2} Z_{2i}^\top Z_{2i-1} \overset{d}{=} \frac{2}{\|\theta_\sharp\|^2 n} \left(\chi^2_{nd/2} - \tilde{\chi}^2_{nd/2}\right) \tag{247}$$

$$= \frac{2}{\|\theta_\sharp\|^2 n} \left(\chi^2_{nd/2} - \mathbb{E}[\chi^2_{nd/2}] + \mathbb{E}[\tilde{\chi}^2_{nd/2}] - \tilde{\chi}^2_{nd/2}\right). \tag{248}$$

From the chi-square tail bound in (330) and (331) it holds that

$$\mathbb{P}\left[\left|\chi^2_{nd/2} - \mathbb{E}[\chi^2_{nd/2}]\right| \geq 2\sqrt{nd/2t} + 2t\right] \leq 2e^{-t} \tag{249}$$

and so it holds with probability $1 - \epsilon$ that

$$\left|\chi^2_{nd/2} - \mathbb{E}[\chi^2_{nd/2}]\right| \leq \sqrt{2nd\log\left(\frac{2}{\epsilon}\right)} + 2\log\left(\frac{2}{\epsilon}\right) \leq 4\sqrt{nd}\log\left(\frac{2}{\epsilon}\right). \tag{250}$$

Hence, by the union bond

$$\frac{2}{\|\theta_\sharp\|^2 n} \sum_{i=1}^{n/2} Z_i^\top Z_{i-1} \leq \frac{16\log\left(\frac{2}{\epsilon}\right)}{\|\theta_\sharp\|^2} \sqrt{\frac{d}{n}}. \tag{251}$$

with probability $1 - \epsilon$. The claim (20) follows from the analysis of the terms above, the choice of $\epsilon = \frac{2}{n}$ and a union bound. $\qquad\square$

## C.2 Proof of Proposition 6: Impossibility lower bound

**The effect of knowledge of $\theta_*$** To begin, it is apparent that if $\|\theta_*\|^2 = \|\theta_\sharp\|^2$, the first term in (20) vanishes, and the loss is bounded by the remaining terms. Furthermore, if $\|\theta_*\| \leq 1$ then the dominant term in the brackets is $\frac{1}{\|\theta_*\|^2}\sqrt{\frac{d}{n}}$, and evidently, dominant error term suffers from a penalty of $\sqrt{d}$, even though, essentially, $\rho$ is a one-dimensional parameter. If $\theta_*$ is known exactly up to a sign, that is $\theta_\sharp = \pm\theta_*$ (and it is not just their norms which are equal), then the estimation error can be reduce to the case of $d = 1$. This is a simple consequence of the rotational invariance of the distribution of the Gaussian noise, which implies that the projections $(\pm\theta_*^\top X_i)_{i=1}^n$ are sufficient statistics for the estimation of $\rho$. The estimator constructs the projections

$$U_i := \frac{\theta_\sharp^\top X_i}{\|\theta_\sharp\|} = \pm\|\theta_*\| \cdot S_i + \frac{\pm\theta_*^\top Z_i}{\|\theta_*\|^2} \overset{d}{=} \|\theta_*\| \cdot S_i + W_i, \tag{252}$$

where $W_i \sim N(0,1)$. This is effectively a one-dimensional model with parameter given by $\|\theta_*\|$, and so Corollary 5 immediately follows from Theorem 4.

**Tightness of the impossibility lower bound**    Evidently, Corollary 5 and Proposition 6 match in their dependence on the number of samples $\Theta(\frac{1}{\sqrt{n}})$, but it is not clear what the optimal dependence on $\|\theta_*\|$ is. The estimator we propose is based on the moment $\rho = \mathbb{E}[U_i U_{i-1}]$ and can be contrasted with likelihood based methods as follows. Letting $p_\delta(s_0^n)$ denote the probability of the sign sequence $s_0^n$ with flip probability $\delta$, and letting $\varphi(x)$ denote the standard Gaussian density, the likelihood of $u_1^n$ is given by

$$P_{\delta,\theta_*}(u_1^n) = \sum_{s_0^n \in \{\pm 1\}^{n+1}} p_\delta(s_0^n) \cdot \prod_{i \in [n]} \varphi(u_i - s_i\|\theta_*\|). \tag{253}$$

This function is a large degree polynomial in $\delta$ on the order of $n$. Even if one sums only over $s_0^n$ with the typical number of flips, then this degree is $\Theta(\delta n)$, which means this polynomial has a degree which blows up with $n$. Thus, the MLE may indeed be sensitive to empirical errors. The update of the Baum-Welch algorithm (or EM) can also be easily computed and contrasted with our proposed estimator. Letting $P_{\delta,\theta_*}$ denote the probability distribution of the corresponding model with flip probability $\delta$ and mean parameter $\theta_*$, the Baum-Welch estimate $\hat{\delta}^{(j)}$ at iteration $j$ is given by

$$\hat{\delta}^{(j)} = \frac{1}{n} \sum_{i=1}^{n} P_{\hat{\delta}^{(j-1)},\theta_*}(S_i \neq S_{i-1} \mid u_1^n). \tag{254}$$

Evidently, the inner estimate is the probability that $S_i \neq S_{i-1}$ conditioned on the entire sample $u_1^n$, rather than just $u_i, u_{i-1}$ as in our proposed estimator in (19). However, deriving sharp error rate bounds for this estimator seems to be a challenging task.

In terms of minimax lower bounds, the difficulty arises since a reduction to an (almost) memoryless GMM as in lower bound for the estimation of $\theta_*$ in Theorem 2 does not seem fruitful. A standard application of Le-Cam's method requires bounding the total variation between models $P_{\delta_1,\theta_*}$ and $P_{\delta_2,\theta_*}$ for some "close" $\delta_1, \delta_2$ (e.g., $\delta_1 = 0$ and $\delta_2 = \epsilon$). The total variation is then typically bounded by the KL divergence. However, the KL divergence does not tensorize (due to the memory), and using the chain rule requires evaluating the KL divergence for the process $U_1^n$ which is not a Markov process (but rather an HMM). Alternatively, further bounding the $n$-dimensional KL divergence with a chi-square divergence, which is a convenient choice for mixture models, leads to an excessively large bound.

**The proof's main ideas**    The proof is based on Le-Cam's method that is applied to a genie-aided model in which the estimator knows every other sign $S_0, S_2, S_4, \ldots, S_n$. The main technical challenge is then to bound the total variation for this genie-aided model. A complete proof is presented below.

*Proof of Proposition 6.* For a given $\delta \in [0,1]$ let $P_\delta^{(n)}$ denote the probability distribution of $U_1^n$ under the model (252), that is $U_i = t \cdot S_i + W_i$, $i \in [n]$, where $t \equiv \|\theta_*\|$ and $S_i$ is a binary symmetric Markov chain with flip probability $\delta$, and $\mathbb{P}[S_0 = 1] = \frac{1}{2}$. It should be noted that $P_\delta^{(n)}$ is a distribution on $\mathbb{R}^n$ which is *not* a product distribution. To lower bound the estimation error of an estimator $\hat{\delta}(U_1^n)$, we consider a genie-aided estimator which is informed with the values of $S_0, S_2, S_4, \ldots S_n$ (assuming for simplicity that $n$ is even). We then set $\epsilon \in (0, \frac{1}{2})$ and use Le-Cam's two point method [Wainwright, 2019, Section 15.2] with $\delta_0 = \frac{1}{2}$ and $\delta_1 = \frac{1}{2} - \epsilon$. Let us denote by $Q_{U_1^n S_0^n}$ (resp. $P_{U_1^n S_0^n}$) the joint probability distribution of $U_1^n$ and $S_0^n$ under $\delta = \frac{1}{2}$ (resp. $\delta = \frac{1}{2} - \epsilon$), and marginals and conditional versions by standard notation, e.g., $Q_{U_2 U_4 | S_0 S_2}$. The proof of the proposition follows from Le-Cam's two point method, which states that for any estimator $\hat{\delta}(U_1^n, S_0, S_2, S_4, \ldots, S_n)$, and thus also for any less informed estimator $\hat{\delta}(U_1^n)$,

$$\mathbb{E}\left[\left|\hat{\delta} - \delta\right|\right] \geq \frac{\epsilon}{2} \cdot \left(1 - \mathrm{d}_{\mathrm{TV}}\left(P_{U_1 U_2 \cdots U_n S_0 S_2 S_4 \cdots S_n}, Q_{U_1 U_2 \cdots U_n S_0 S_2 S_4 \cdots S_n}\right)\right). \tag{255}$$

We next obtain a bound on the total variation distance in Lemma 17 of $4\sqrt{n}\epsilon$ for $t \leq \frac{1}{\sqrt{2}}$ and choosing $\epsilon = \frac{1}{8\sqrt{n}}$ in (255) completes the proof of the lower bound. $\qquad\square$

**Lemma 17.** *If $t \leq 1/\sqrt{2}$ then*

$$\mathrm{d}_{\mathrm{TV}}\left(P_{U_1 U_2 \cdots U_n, S_0 S_2 S_4 \cdots S_n}, Q_{U_1 U_2 \cdots U_n, S_0 S_2 S_4 \cdots S_n}\right) \leq \sqrt{\frac{5}{2}}n\epsilon + \sqrt{8n}t\epsilon. \tag{256}$$

*Proof.* By Pinsker's inequality (e.g., [Tsybakov, 2008, Lemma 2.5]), it holds for any pair of probability measures $P, Q$ that $d_{\mathrm{TV}}(P, Q) \leq \sqrt{\frac{1}{2} D_{\mathrm{KL}}(Q, P)}$. We next upper bound the KL divergence. To this end, recall the chain rule, that for any joint distributions $P_{XY}$ and $Q_{XY}$ (with conditional distributions $P_{Y|X}, Q_{Y|X}$) the chain rule for the KL divergence states that $D_{\mathrm{KL}}(P_{XY} \parallel Q_{XY}) = D_{\mathrm{KL}}(P_X \parallel Q_X) + D_{\mathrm{KL}}(P_{Y|X} \parallel Q_{Y|X} \mid P_X)$ where $D_{\mathrm{KL}}(P_{Y|X} \parallel Q_{Y|X} \mid P_X) := \int D_{\mathrm{KL}}(P_{Y|X} \parallel Q_{Y|X}) \mathrm{d} P_X$ is the conditional KL divergence [Cover and Thomas, 2006, Theorem 2.5.3]. Thus,

$$D_{\mathrm{KL}}\left(P_{U_1 U_2 \cdots U_n, S_0 S_2 S_4 \cdots S_n} \parallel Q_{U_1 U_2 \cdots U_n, S_0 S_2 S_4 \cdots S_n}\right)$$

$$= D_{\mathrm{KL}}\left(P_{S_0 S_2 S_4 \cdots S_n} \parallel Q_{S_0 S_2 S_4 \cdots S_n}\right) \tag{257}$$

$$+ D_{\mathrm{KL}}\left(P_{U_1 U_2 \cdots U_n | S_0 S_2 S_4 \cdots S_n} \parallel Q_{U_1 U_2 \cdots U_n | S_0 S_2 S_4 \cdots S_n} \mid P_{S_0 S_2 S_4 \cdots S_n}\right). \tag{258}$$

We next bound each of the two KL divergences appearing in (257) and (258). First,

$$D_{\mathrm{KL}}\left(P_{S_0 S_2 S_4 \cdots S_n} \parallel Q_{S_0 S_2 S_4 \cdots S_n}\right) \overset{(a)}{\leq} D_{\mathrm{KL}}\left(P_{S_0^n} \parallel Q_{S_0^n}\right) \tag{259}$$

$$= \sum_{s_0^n \in \{-1,1\}^{n+1}} P_{S_0^n}(s_0^n) \log \frac{P_{S_0^n}(s_0^n)}{Q_{S_0^n}(s_0^n)} \tag{260}$$

$$\overset{(b)}{=} (n+1)\log 2 - \sum_{s_0^n \in \{-1,1\}^{n+1}} P_{S_0^n}(s_0^n) \log \frac{1}{P_{S_0^n}(s_0^n)} \tag{261}$$

$$\overset{(c)}{=} (n+1)\log 2 - H(S_0, S_1, \ldots, S_n) \tag{262}$$

$$\overset{(d)}{=} (n+1)\log 2 - H(S_0) - \sum_{i=1}^{n} H(S_i \mid S_0^{i-1}) \tag{263}$$

$$\overset{(e)}{=} n \cdot \log 2 - \sum_{i=1}^{n} H(S_i \mid S_{i-1}) \tag{264}$$

$$\overset{(f)}{=} n \cdot \left[\log 2 - h_b\left(\frac{1}{2} - \epsilon\right)\right], \tag{265}$$

where $(a)$ follows from the convexity of the KL divergence (recall $S_0^n = S_0, S_1, S_2, \ldots, S_n$), $(b)$ since $Q_{S_0^n}(s_0^n) = 2^{-(n+1)}$ for any $s_0^n \in \{\pm 1\}^{n+1}$, $(c)$ follows by defining the entropy $H$ of $S_0^n$ (under the probability measure $P$), $(d)$ follows from the chain rule of entropy and the definition of conditional entropy [Cover and Thomas, 2006, Theorem 2.2.1], $(e)$ follows from Markovity and $H(S_0) = \log 2$, $(f)$ follows from $H(S_i \mid S_{i-1}) = h_b(\delta_1) = h_b(\frac{1}{2} - \epsilon)$ where $h_b(\delta) := -\delta \log \delta - (1-\delta) \log(1-\delta)$ is the binary entropy function. Now, for $\epsilon \in (0, \frac{1}{2})$ the power series expansion of the binary entropy function results the bound

$$h_b\left(\frac{1}{2} - \epsilon\right) = \log 2 - \sum_{k=1}^{\infty} \frac{(2\epsilon)^{2k}}{2k(2k-1)} \tag{266}$$

$$\geq \log 2 - (2\epsilon)^2 \sum_{k=1}^{\infty} \frac{1}{2k(2k-1)} \tag{267}$$

$$\geq \log 2 - 4\epsilon^2 \sum_{k=1}^{\infty} \frac{1}{(2k-1)^2} \tag{268}$$

$$= \log 2 - \frac{\pi^2}{2}\epsilon^2 \tag{269}$$

$$\geq \log 2 - 5\epsilon^2. \tag{270}$$

Inserting this bound into (265) results that the first term in (257) is upper bounded as

$$D_{\mathrm{KL}}\left(P_{S_0 S_2 S_4 \cdots S_n} \parallel Q_{S_0 S_2 S_4 \cdots S_n}\right) \leq 5n\epsilon^2. \tag{271}$$

We now move on to bound the second term in (258). To this end, we note that the distribution of $U_1^n$ conditioned on $S_0, S_2, S_4, \ldots S_n$ can be decomposed in a simple way. First, under $Q$, the signs $S_0^n$

are i.i.d., and so $U_1^n$ is a vector of independent samples from a Gaussian mixture model. Furthermore, $U_i$ depends on the sign $S_i$ but otherwise is independent of all other $S_1^n \backslash \{S_i\}$. Hence,

$$Q_{U_1 U_2 \cdots U_n | S_0 S_2 S_4 \cdots S_n} = Q_{U_1} \cdot Q_{U_2 | S_2} \cdot Q_{U_3} \cdot Q_{U_4 | S_4} \cdots Q_{U_{n-1}} \cdot Q_{U_n | S_n}, \qquad (272)$$

that is, a model of independent samples, where the odd samples are drawn from a Gaussian mixture model, and the even samples from a Gaussian location model with known sign $S_{2i}$. Second, under $P$, an application of Bayes rule and the Markovity assumption results the decomposition to pairs of samples given by

$$P_{U_1 U_2 \cdots U_n | S_0 S_2 S_4 \cdots S_n}$$
$$= P_{U_1 U_2 | S_0 S_2 S_4 \cdots S_n} \cdot P_{U_3 U_4 | S_0 S_2 S_4 \cdots S_n U_1 U_2} \cdots P_{U_{n-1} U_n | S_0 S_2 S_4 \cdots S_n U_1^{n-2}} \qquad (273)$$
$$= P_{U_1 U_2 | S_0 S_2} \cdot P_{U_3 U_4 | S_2 S_4} \cdots P_{U_{n-1} U_n | S_{n-2} S_n}. \qquad (274)$$

The chain rule for the KL divergence therefore implies that

$$D_{KL}\left(P_{U_1 U_2 \cdots U_n | S_0 S_2 S_4 \cdots S_n} \parallel Q_{U_1 U_2 \cdots U_n | S_0 S_2 S_4 \cdots S_n} \mid P_{S_0 S_2 S_4 \cdots S_n}\right)$$

$$\overset{(a)}{=} D_{KL}\left(P_{U_1 U_2 | S_0 S_2} \parallel Q_{U_1} \cdot Q_{U_2 | S_2} \mid P_{S_0 S_2}\right) + D_{KL}\left(P_{U_3 U_4 | S_2 S_4} \parallel Q_{U_3} \cdot Q_{U_4 | S_4} \mid P_{S_2 S_4}\right)$$
$$+ \cdots + D_{KL}\left(P_{U_{n-1} U_n | S_{n-2} S_n} \parallel Q_{U_{n-1}} \cdot Q_{U_n | S_n} \mid P_{S_{n-2} S_n}\right) \qquad (275)$$

$$\overset{(b)}{=} \frac{n}{2} \cdot D_{KL}\left(P_{U_1 U_2 | S_0 S_2} \parallel Q_{U_1} \cdot Q_{U_2 | S_2} \mid P_{S_0 S_2}\right) \qquad (276)$$

$$\overset{(c)}{=} \frac{n}{2} \cdot D_{KL}\left(P_{U_1 | S_0 S_2} \parallel Q_{U_1} \mid P_{S_0 S_2}\right) + \frac{n}{2} \cdot D_{KL}\left(P_{U_2 | S_0 S_2} \parallel Q_{U_2 | S_2} \mid P_{S_0 S_2}\right) \qquad (277)$$

$$\overset{(d)}{=} \frac{n}{2} \cdot D_{KL}\left(P_{U_1 | S_0 S_2} \parallel Q_{U_1} \mid P_{S_0 S_2}\right) \qquad (278)$$

$$\overset{(e)}{=} \frac{n}{2} \cdot D_{KL}\left(P_{U_1 | S_0 = 1, S_2} \parallel Q_{U_1} \mid P_{S_2 | S_0 = 1}\right) \qquad (279)$$

where $(a)$ follows from (274) and the chain rule for KL divergence, and $(b)$ follows from the stationarity of the Markov chain, $(c)$ follows again from the chain rule, $(d)$ follows since $D_{KL}(P_{U_2 | S_0 S_2} \parallel Q_{U_2 | S_2} \mid P_{S_0 S_2}) = D_{KL}(P_{U_2 | S_2} \parallel Q_{U_2 | S_2} \mid P_{S_2}) = 0$, and $(e)$ follows since by symmetry, we may condition on $S_0 = 1$. Thus, the last KL divergence in (279) should be averaged over the cases $S_2 = -1$ and $S_2 = 1$. For the first case, it holds that $P_{S_1 | S_0 S_2}(\cdot \mid 1, -1)$ is a uniform distribution on $\{\pm 1\}$. Hence, conditioned on $S_0 = 1, S_2 = -1$, under $P$, $U_1$ is a sample from a balanced Gaussian mixture, just as under $Q$. So,

$$D_{KL}\left(P_{U_1 | S_0 = 1, S_2 = -1} \parallel Q_{U_1}\right) = 0. \qquad (280)$$

So, the KL divergence is only comprised of the term in the second case $S_2 = 1$. Continuing to evaluate the KL divergence in (279), we get

$$D_{KL}\left(P_{U_1 | S_0 = 1, S_2} \parallel Q_{U_1} \mid P_{S_2 | S_0 = 1}\right) = P_{S_2 | S_0 = 1}(1) \cdot D_{KL}\left(P_{U_1 | S_0 = 1, S_2 = 1} \parallel Q_{U_1}\right) \qquad (281)$$

$$\overset{(a)}{\leq} D_{KL}\left(P_{U_1 | S_0 = 1, S_2 = 1} \parallel Q_{U_1}\right) \qquad (282)$$

$$\overset{(b)}{\leq} \chi^2\left(P_{U_1 | S_0 = 1, S_2 = 1} \parallel Q_{U_1}\right) \qquad (283)$$

where $(a)$ follows since $P_{S_2 | S_0}(1 \mid 1) = (1 - \delta_1)^2 + \delta_1^2 \leq 1$, and $(b)$ follows from the chi-square divergence bound on the KL divergence (e.g., [Tsybakov, 2008, Eq. (2.27)]). Now, recall that under $Q_{U_1}$, it holds that $U_1 \sim \frac{1}{2} N(t, 1) + \frac{1}{2} N(-t, 1)$, and under $P_{U_1 | S_0 = 1, S_2 = 1}$ it holds that $U_1 \sim (1 - \alpha) \cdot N(t, 1) + \alpha N(-t, 1)$ where

$$\alpha = P_{S_1 | S_0 S_2}(-1 \mid 1, 1) = \frac{\delta_1^2}{(1 - \delta_1)^2 + \delta_1^2}. \qquad (284)$$

Thus, the chi-square divergence in (283) is between a balanced Gaussian mixture (with probability that $S_1 = 1$ being $\frac{1}{2}$) and an unbalanced Gaussian mixture (with probability that $S_1 = 1$ being $1 - \alpha > \frac{1}{2}$). Let $\psi_\beta(u)$ denote the probability density function of $U = t \cdot S + W$ with $\mathbb{P}[S = -1] = 1 - \mathbb{P}[S = 1] = \beta$ and $W \sim N(0, 1)$. Then, if $\varphi(u) \equiv \frac{1}{\sqrt{2\pi}} e^{-u^2/2}$ is the standard Gaussian density function, it holds that

$$\psi_\beta(u) = \beta \cdot \varphi(u + t) + (1 - \beta) \cdot \varphi(u + t) \qquad (285)$$

$$= \beta \cdot \frac{1}{\sqrt{2\pi}} e^{-(u+t)^2/2} + (1-\beta) \cdot \frac{1}{\sqrt{2\pi}} e^{-(u-t)^2/2} \tag{286}$$

$$= e^{-t^2/2} \cdot \varphi(u) \cdot \left[ \beta \cdot e^{-ut} + (1-\beta) \cdot e^{ut} \right]. \tag{287}$$

Specifically, for $\beta = \frac{1}{2}$ it holds that $\psi_{1/2}(u) = e^{-t^2/2} \cdot \varphi(u) \cdot \cosh(ut)$. Therefore, the chi-square divergence from (283) is upper bounded as

$$\chi^2 \left( P_{U_1|S_0=1,S_2=1} \,\|\, Q_{U_1} \right)$$

$$= \int_{-\infty}^{\infty} \frac{\left[ P_{U_1|S_0S_2}(u \mid 1,1) - Q_{U_1}(u) \right]^2}{Q_{U_1}(u)} du \tag{288}$$

$$= e^{-t^2/2} \cdot \int_{-\infty}^{\infty} \varphi(u) \frac{\left[ \left( \alpha - \frac{1}{2} \right) \cdot e^{-ut} + \left( 1 - \alpha - \frac{1}{2} \right) \cdot e^{ut} \right]^2}{\cosh(ut)} du \tag{289}$$

$$\overset{(a)}{\leq} e^{-t^2/2} \cdot \int_{-\infty}^{\infty} \varphi(u) \left[ \left( \alpha - \frac{1}{2} \right) \cdot e^{-ut} + \left( 1 - \alpha - \frac{1}{2} \right) \cdot e^{ut} \right]^2 du \tag{290}$$

$$= e^{-t^2/2} \cdot \int_{-\infty}^{\infty} \varphi(u) \left[ \left( \alpha - \frac{1}{2} \right)^2 \cdot e^{-2ut} + 2 \left( \alpha - \frac{1}{2} \right) \left( 1 - \alpha - \frac{1}{2} \right) + \left( 1 - \alpha - \frac{1}{2} \right)^2 \cdot e^{2ut} \right] du \tag{291}$$

$$= e^{-t^2/2} \cdot \left[ \left( \alpha - \frac{1}{2} \right)^2 e^{2t^2} + 2 \left( \alpha - \frac{1}{2} \right) \left( 1 - \alpha - \frac{1}{2} \right) + \left( 1 - \alpha - \frac{1}{2} \right)^2 e^{2t^2} \right] \tag{292}$$

$$\overset{(b)}{\leq} \left[ \left( \alpha - \frac{1}{2} \right)^2 (1 + 4t^2) + 2 \left( \alpha - \frac{1}{2} \right) \left( 1 - \alpha - \frac{1}{2} \right) + \left( 1 - \alpha - \frac{1}{2} \right)^2 (1 + 4t^2) \right] \tag{293}$$

$$= 4t^2 \cdot \left[ \left( \alpha - \frac{1}{2} \right)^2 + \left( 1 - \alpha - \frac{1}{2} \right)^2 \right] \tag{294}$$

$$= 2t^2 \cdot [1 - 4\alpha(1-\alpha)] \tag{295}$$

$$\overset{(c)}{=} 2t^2 \cdot \left[ 1 - \left( \frac{2 - 8\epsilon^2}{2 + 8\epsilon^2} \right)^2 \right] \tag{296}$$

$$= 2t^2 \cdot \left[ 1 - \frac{2 - 8\epsilon^2}{2 + 8\epsilon^2} \right] \left[ 1 + \frac{2 - 8\epsilon^2}{2 + 8\epsilon^2} \right] \tag{297}$$

$$= 2t^2 \cdot \left[ \frac{64\epsilon^2}{(2 + 8\epsilon^2)^2} \right] \tag{298}$$

$$\leq 32t^2\epsilon^2, \tag{299}$$

where $(a)$ follows since $\cosh(x) \geq 1$ for all $x \in \mathbb{R}$, $(b)$ follows using the assumption $t \leq \frac{1}{\sqrt{2}}$, and so $e^{2t^2} \leq 1 + 4t^2$, and $(c)$ is by setting

$$4\alpha(1-\alpha) = 4 \frac{\delta_1^2 (1-\delta_1)^2}{[(1-\delta_1)^2 + \delta_1^2]^2} = 4 \frac{\left( \frac{1}{2} - \epsilon \right)^2 \left( \frac{1}{2} + \epsilon \right)^2}{\left[ \left( \frac{1}{2} + \epsilon \right)^2 + \left( \frac{1}{2} - \epsilon \right)^2 \right]^2} = \left( \frac{2 - 8\epsilon^2}{2 + 8\epsilon^2} \right)^2. \tag{300}$$

Using the bound (299) in (283), We thus conclude that

$$\mathrm{D_{KL}} \left( P_{U_1|S_0=1,S_2} \,\|\, Q_{U_1} \mid P_{S_2|S_0=1} \right) \leq 32t^2\epsilon^2 \tag{301}$$

and then in (279) that the second KL term of (258) is upper bounded as

$$\mathrm{D_{KL}} \left( P_{U_1U_2\cdots U_n|S_0S_2S_4\cdots S_n} \,\|\, Q_{U_1U_2\cdots U_n|S_0S_2S_4\cdots S_n} \mid P_{S_0S_2S_4\cdots S_n} \right) \leq 16nt^2\epsilon^2. \tag{302}$$

Combining this bound with the bound on the first KL term of (258) given in (271) we obtain

$$\mathrm{D_{KL}} \left( P_{U_1U_2\cdots U_n, S_0S_2S_4\cdots S_n} \,\|\, Q_{U_1U_2\cdots U_n, S_0S_2S_4\cdots S_n} \right) \leq 5n\epsilon^2 + 16nt^2\epsilon^2. \tag{303}$$

The aforementioned Pinsker bound on the total variation distance, and the relation $\sqrt{a + b} \leq \sqrt{a} + \sqrt{b}$ then completes the proof. $\qquad\square$

# D Proofs for Section 3: Analysis of Algorithm 1

*Proof of Theorem 7.* Before presenting the analysis of Algorithm 1, we first specify the choices of the constants $\lambda_\theta \geq 1$ and $\lambda_\delta \geq 1$. Recall that from Theorem 1 for a Markov model $(\theta_*, \delta)$ in low dimension, $d \leq \delta n$, there exists a numerical constant $\lambda_\theta > 0$,[3] so that the estimator $\hat{\theta} \equiv \hat{\theta}_{\text{cov}}(X_1^n; k = \frac{1}{8\delta})$, which assumes a perfect knowledge of $\delta$, achieves with probability $1 - O(\frac{1}{n})$,

$$\text{loss}(\hat{\theta}, \theta_*) \leq \lambda_\theta \cdot \log(n) \cdot \begin{cases} \|\theta_*\|, & \|\theta_*\| \leq \left(\frac{\delta d}{n}\right)^{1/4} \\ \frac{1}{\|\theta_*\|}\sqrt{\frac{\delta d}{n}}, & \left(\frac{\delta d}{n}\right)^{1/4} \leq \|\theta_*\| \leq \sqrt{\delta} \\ \sqrt{\frac{d}{n}}, & \|\theta_*\| \geq \sqrt{\delta} \end{cases} \tag{304}$$

if $d \leq \delta n$, and

$$\text{loss}(\hat{\theta}, \theta_*) \leq \lambda_\theta \cdot \log(n) \cdot \begin{cases} \|\theta_*\|, & \|\theta_*\| \leq \sqrt{\frac{d}{n}} \\ \sqrt{\frac{d}{n}}, & \|\theta_*\| \geq \sqrt{\frac{d}{n}} \end{cases} \tag{305}$$

if $\delta n \leq d \leq n$. We assume that $\lambda_\theta \geq 1$, and otherwise replace $\lambda_\theta$ by 1.

In addition, it holds from Theorem 4 that for the estimator $\hat{\delta}$ for $\delta$, which is based on a mismatched mean $\theta_\sharp$, i.e.,

$$\hat{\delta} \equiv \hat{\delta}_{\text{corr}}(X_1^n; \theta_\sharp) := \frac{1}{2}\left(1 - \hat{\rho}_{\text{corr}}(X_1^n; \theta_\sharp)\right), \tag{306}$$

there exists a numerical constant $\lambda_\delta > 0$, for which with probability larger than $1 - O(\frac{1}{n})$,

$$\text{loss}(\hat{\delta}, \delta) \leq \lambda_\delta \left[\frac{\left|\|\theta_*\|^2 - \|\theta_\sharp\|^2\right|}{\|\theta_\sharp\|^2} + \frac{\log(n)}{\|\theta_\sharp\|^2}\sqrt{\frac{d}{n}}\right] \tag{307}$$

assuming that $\|\theta_\sharp\| \leq 2$ in order to simplify the bound to the regime of interest. We assume here too that $\lambda_\delta \geq 1$, and otherwise replace $\lambda_\delta$ by 1.

For the analysis we assume that all three steps of the algorithm are successful estimation events. By the union bound, this occurs with probability $1 - O(\frac{1}{n})$.

Analysis of Step A:

First, suppose that

$$\|\theta_*\| \leq \lambda_\theta \log(n) \cdot \left(\frac{d}{n}\right)^{1/4}. \tag{308}$$

Then, it holds by (304) and (305) that

$$\|\hat{\theta}^{(A)}\| \leq \|\theta_*\| + \lambda_\theta \log(n) \cdot \left(\frac{\delta d}{n}\right)^{1/4} \leq 2\lambda_\theta \log(n)\left(\frac{d}{n}\right)^{1/4} \tag{309}$$

and thus the algorithm will stop and output $\hat{\theta} = 0$, for which it holds that $\text{loss}(\theta_*, \hat{\theta}^{(A)}) = \|\theta_*\|$. This agrees with (26) and (27).

Second, suppose that $\|\theta_*\| \geq 1$. Now, it holds by (304) and (305)

$$\|\hat{\theta}^{(A)}\| \geq \|\theta_*\| - \lambda_\theta \log(n)\sqrt{\frac{d}{n}} \geq \frac{1}{2}, \tag{310}$$

by the assumption $d \leq \frac{1}{4\lambda_\theta^2 \log^2(n)} \cdot n$. Thus the algorithm will stop and output $\hat{\theta} = \hat{\theta}^{(A)}$, for which it also holds that

$$\text{loss}(\theta_*, \hat{\theta}^{(A)}) \leq \lambda_\theta \cdot \log(n) \cdot \sqrt{\frac{d}{n}} \tag{311}$$

---

[3]The constants can be deduced from the proof, though they were not optimized. We reiterate that both $\lambda_\theta$ and $\lambda_\delta$ below are universal numerical constants. In particular, they do *not* depend on $\theta_*, \delta$ or any other problem parameters. The subscripts are merely to emphasize that they are obtained in the estimation procedure for $\theta_*$ and $\delta$, respectively.

(irrespective of the value of $\delta$). This also agrees with (26) and (27).

Thus, the algorithm will continue to Step B only if

$$\lambda_\theta \log(n) \cdot \left(\frac{d}{n}\right)^{1/4} \leq \|\theta_*\| \leq 1 \tag{312}$$

which we henceforth assume. As a preparation for Step B, in which $\hat{\theta}^{(A)}$ will play the rule of $\theta_\sharp$ (the mismatched mean estimator of Section 3) in an estimation of $\delta$, we bound the absolute value $\left| \|\hat{\theta}^{(A)}\| - \|\theta_*\| \right|$ and also show that $\|\hat{\theta}^{(A)}\| \geq \frac{1}{2}\|\theta_*\|$. To this end, we may repeat the arguments of the proof of Theorem 1, and specifically, arguments similar to (126) to show that

$$\left| \|\hat{\theta}^{(A)}\| - \|\theta_*\| \right| \lesssim \log(n) \cdot \psi\left(n, d, \delta = \frac{1}{2}, k = 1\right) \tag{313}$$

$$= \log(n) \cdot \left[ 2\sqrt{\frac{\delta}{n}} \cdot \|\theta_*\|^2 + 2\sqrt{\frac{d}{n}} \cdot \|\theta_*\| + 13\sqrt{\frac{d}{n}} + 10\frac{d}{n} \right]. \tag{314}$$

Note that the probability of this event is $1 - O(\frac{1}{n})$, and is included in the successful estimation event mentioned in the beginning of the proof. Under the assumption $\|\theta_*\| \leq 1$ it then holds that

$$\left| \|\hat{\theta}^{(A)}\| - \|\theta_*\| \right| \leq \lambda_\theta \log(n) \cdot \sqrt{\frac{d}{n}}. \tag{315}$$

Note that we take $\lambda_\theta > 0$ to be large enough so that this holds. The assumed case $\|\theta_*\| \geq \lambda_\theta \log(n) \cdot \left(\frac{d}{n}\right)^{1/4}$, and the assumption of the theorem $d \leq \frac{n}{16}$ then imply that

$$\frac{1}{2}\|\theta_*\| \leq \|\hat{\theta}^{(A)}\| \leq 2\|\theta_*\| \leq 2. \tag{316}$$

Analysis of Step B:

Setting $\theta_\sharp = \hat{\theta}^{(A)}$ and utilizing the two properties just derived in (315) and (316), (307) implies that

$$\mathsf{loss}(\hat{\delta}^{(B)}, \delta) \leq \lambda_\delta \left[ \frac{\left| \|\theta_*\|^2 - \|\theta_\sharp\|^2 \right|}{\|\theta_\sharp\|^2} + \frac{\log(n)}{\|\theta_\sharp\|^2} \sqrt{\frac{d}{n}} \right] \tag{317}$$

$$\leq 4\lambda_\delta \lambda_\theta \frac{\log(n)}{\|\theta_*\|^2} \sqrt{\frac{d}{n}}. \tag{318}$$

There are two cases. First suppose that $\|\theta_*\| \geq \sqrt{8\lambda_\delta \lambda_\theta \log(n)}\left(\frac{d}{\delta^2 n}\right)^{1/4}$. In this case,

$$\mathsf{loss}(\hat{\delta}^{(B)}, \delta) \leq \frac{\delta}{2} \tag{319}$$

and so

$$\frac{1}{2}\delta \leq \hat{\delta}^{(B)} \leq 2\delta. \tag{320}$$

In addition, from (316) it holds that

$$64\lambda_\delta \lambda_\theta \frac{\log(n)}{\|\hat{\theta}^{(A)}\|^2} \sqrt{\frac{d}{n}} \geq 16\lambda_\delta \lambda_\theta \frac{\log(n)}{\|\theta_*\|^2} \sqrt{\frac{d}{n}} \geq 2\delta \geq \hat{\delta}^{(B)} \tag{321}$$

and so the algorithm will continue to Step C. Now, suppose that $\|\theta_*\| \leq \sqrt{8\lambda_\delta \lambda_\theta \log(n)}\left(\frac{d}{\delta^2 n}\right)^{1/4}$. If the algorithm stops then $\hat{\theta} = \hat{\theta}^{(A)}$, and the estimation rates of the Gaussian mixture model are achieved, which agrees with (26) and (27). If the algorithm does not stop and proceeds to Step C then it holds that

$$\hat{\delta}^{(B)} \geq 64\lambda_\delta \lambda_\theta \frac{\log(n)}{\|\hat{\theta}^{(A)}\|^2} \sqrt{\frac{d}{n}} \geq 16\lambda_\delta \lambda_\theta \frac{\log(n)}{\|\theta_*\|^2} \sqrt{\frac{d}{n}} \geq \delta, \tag{322}$$

by (316) and the assumption $\|\theta_*\| \leq \sqrt{8\lambda_\delta \lambda_\theta \log(n)}\left(\frac{d}{\delta^2 n}\right)^{1/4}$.

To conclude, there are cases in which the algorithm proceeds to Step C. If $\|\theta_*\| \geq \sqrt{8\lambda_\delta \lambda_\theta \log(n)}(\frac{d}{\delta^2 n})^{1/4}$ then it proceeds Step C and (320) holds. Otherwise, the algorithm might proceed to Step C, yet now only $\hat{\delta}^{(B)} \geq \frac{1}{2}\delta$ is assured.

Analysis of Step C:

If the algorithm has proceeded to Step C, then it is guaranteed that $\hat{\delta}^{(B)} \geq \frac{1}{2}\delta$ (in any event). Recall that, ideally, had $\delta$ was known, the choice of the block length $k$ for the estimator $\hat{\theta}_{\text{cov}}(X_{2n+1}^{3n}; k)$ is $k_* := \frac{1}{8\delta}$ (Section 2). It can be readily verified that the analysis of Section 2 is valid when using any *smaller* blocklength $k$, and that the error rate improve as $k$ increases from $k = 1$ to $k = k_*$. In accordance to the analysis of Step B, there are two possible cases. If $\|\theta_*\| \geq \sqrt{8\lambda_\delta \lambda_\theta \log(n)}(\frac{d}{\delta^2 n})^{1/4}$ then it holds that $\frac{1}{2}\delta \leq \hat{\delta}^{(B)} \leq 2\delta$. Using $k = \frac{1}{16\hat{\delta}^{(B)}}$ then assures that $k \leq k_*$. In addition, since the error bound scales linearly with $\psi(n, d, \delta, k)$ (see (92)), the error can increase by a factor at most 2. Thus, $\hat{\theta}^{(C)}$ achieves the error rates in (304) and (305), with a factor of 2. There are again two cases to consider. If $d \geq \frac{1}{64\lambda_\delta^2 \lambda_\theta^2 \log^2(n)}\delta^4 n$, then the interval $(\sqrt{8\lambda_\delta \lambda_\theta \log(n)}(\frac{d}{\delta^2 n})^{1/4}, \sqrt{\delta})$ is empty (its right end point is smaller than its left end point), and plugging $k$ in the error rates of (304) and (305), the resulting error rates are as in the Gaussian mixture model. If $d \leq \frac{1}{64\lambda_\delta^2 \lambda_\theta^2 \log^2(n)}\delta^4 n$ then the aforementioned interval is non-empty, and the resulting error rates agree with (26) and (27). Otherwise, if $\|\theta_*\| \leq \sqrt{8\lambda_\delta \lambda_\theta \log(n)}(\frac{d}{\delta^2 n})^{1/4}$, and the estimator $\hat{\theta}_{\text{cov}}(X_{2n+1}^{3n}; k)$ operates with $k \leq k_*$, but $k$ can be as low as 1. Thus, error rates of the Gaussian mixture are again achieved, and this agrees with (26) and (27). $\qquad\square$

# E  Useful results

**Bernstein's inequality**  Let $X_1, \cdots, X_\ell$ be independent random variables and $|X_i| \leq b$ almost surely for every $i \in [\ell]$. Then [Wainwright, 2019, Proposition 2.14][4] states

$$\mathbb{P}\left[\left|\frac{1}{\ell}\sum_{i=1}^{\ell}(X_i - \mathbb{E}[X_i])\right| \geq \delta\right] \leq 2\exp\left(-\frac{\ell\delta^2/2}{\frac{1}{\ell}\sum_{i=1}^{\ell}\mathbb{V}[X_i] + b\delta/3}\right). \tag{323}$$

**Norm of subGaussian random vectors**  A random vector $X \in \mathbb{R}^d$ is said to be $\sigma^2$-subGaussian if $\mathbb{E}[X] = 0$ and

$$\mathbb{E}\left[\exp(tv^\top X)\right] \leq \exp\left(\frac{\sigma^2 t^2}{2}\right) \tag{324}$$

for every $t \in \mathbb{R}$ and every $v \in \mathbb{S}^{d-1}$. Let $X \in \mathbb{R}^d$ be a $\sigma^2$-subGaussian random vector. Then from [Rigollet and Hütter, 2019, Theorem 1.19], we have for any $\delta > 0$,

$$\mathbb{P}\left[\max_{v \in \mathbb{B}^d} v^\top X \leq 4\sigma\sqrt{d} + 2\sigma\sqrt{2\log\left(\frac{1}{\delta}\right)}\right] = \mathbb{P}\left[\|X\| \leq 4\sigma\sqrt{d} + 2\sigma\sqrt{2\log\left(\frac{1}{\delta}\right)}\right] \geq 1 - \delta. \tag{325}$$

**Gaussian covariance estimation**  Let $W \in \mathbb{R}^{\ell \times d}$ be a matrix with i.i.d. $N(0, 1)$ entries. Then [Wainwright, 2019, Example 6.2] implies that for any $\delta > 0$,

$$\mathbb{P}\left[\left\|\frac{1}{\ell}W^\top W - I_d\right\|_{\text{op}} \leq 2\left(\sqrt{\frac{d}{n}} + \delta\right) + \left(\sqrt{\frac{d}{n}} + \delta\right)^2\right] \geq 1 - 2e^{-n\delta^2/2}. \tag{326}$$

**Davis-Kahan's perturbation bound**  Let $\Sigma, \hat{\Sigma}$ be symmetric matrices with the same dimensions. Let $\lambda_i(\Sigma)$ and $v_i(\Sigma)$ denote the $i$th largest eigenvalue and the associated eigenvector (of unit norm) of $\Sigma$. Fix $i$ and assume that $\lambda_i(\Sigma)$ is well-separated from the rest of the spectrum of $\Sigma$:

$$\min_{j \neq i}|\lambda_i(\Sigma) - \lambda_j(\Sigma)| = \delta > 0. \tag{327}$$

---

[4]In the original form in [Wainwright, 2019, Proposition 2.14], on the right hand side of the inequality, $\mathbb{V}[X_i]$ is replaced with $\mathbb{E}[X_i^2]$ which leads to a seemingly worse concentration bound. However, applying this form of Bernstein's inequality to $Y_i = X_i - \mathbb{E}[X_i]$ allows us to conclude (323).

Then from [Vershynin, 2018, Theorem 4.5.5], we have

$$\mathsf{loss}(v_i(\Sigma), v_i(\hat{\Sigma})) \leq 4 \frac{\|\Sigma - \hat{\Sigma}\|_{\mathrm{op}}}{\delta}, \tag{328}$$

where $\mathsf{loss}(\cdot, \cdot)$ was defined in (3).

**Packing number** Let $M(\delta; \mathbb{B}^d, \|\cdot\|)$ and $N(\delta; \mathbb{B}^d, \|\cdot\|)$ denote the $\delta$-packing number and $\delta$-covering number of $\mathbb{B}^d$ w.r.t. $\|\cdot\|$, respectively. From [Wainwright, 2019, Lemma 5.7 and Example 5.8], we have

$$M(\delta; \mathbb{B}^d, \|\cdot\|) \geq N(\delta; \mathbb{B}^d, \|\cdot\|) \geq \left(\frac{1}{\delta}\right)^d. \tag{329}$$

**Chi-square tail bounds** From the chi-square tail bound in [Boucheron et al., 2013, Remark 2.11]

$$\mathbb{P}\left[\chi_\ell^2 - \ell \geq 2\sqrt{\ell t} + 2t\right] \leq e^{-t} \tag{330}$$

and

$$\mathbb{P}\left[\chi_\ell^2 - \ell \leq -2\sqrt{\ell t}\right] \leq e^{-t}. \tag{331}$$

# F  Numerical validation

In this section, we provide numerical validation of the performance of the estimators $\hat{\theta}_{\mathrm{cov}}(X_1^n; k)$ (cf. (14)), $\hat{\delta}_{\mathrm{corr}}(X_1^n; \theta_\sharp) = \frac{1}{2}(1 - \hat{\rho}_{\mathrm{corr}}(X_1^n; \theta_\sharp))$ (cf. (19)) and Algorithm 1 proposed and analyzed in Theorem 1, Theorem 4 and Corollary 5, and Theorem 7, respectively. Numerical results for these estimators/algorithm are plotted in Figures 3, 4 and 5, respectively.

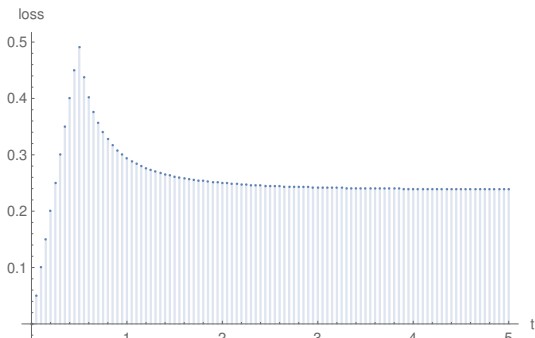

Figure 3: Numerical validation of the performance of the $\theta_*$-estimator $\hat{\theta}_{\mathrm{cov}}(X_1^n; k)$ (cf. (14)) used in the proof of Theorem 1 which assumes a known $\delta$. We take the minimum loss (cf. (3)) achieved by $\hat{\theta}_{\mathrm{cov}}\left(X_1^n; k = \frac{1}{8\delta}\right)$ and the trivial estimator $\hat{\theta}_0(X_1^n) = 0$. We plot the loss as a function of $t = \|\theta_*\|$ for $t \in [0, 5]$ with step size 0.05. We take $n = 5000, d = 250, \delta = 0.05$. Therefore, $\delta > \frac{1}{n}$ and $d < \delta n$.

# G  Open directions

We discuss some open directions pertaining to Remark 3.

1. Suppose $Z_i \sim N(0, \Sigma)$ i.i.d. for some general covariance $\Sigma \succ 0$. If $\Sigma$ is *unknown*, in contrast to the observation above, then the problem becomes significantly more delicate and challenging. A well-known and intuitive example (see, e.g., [Ferguson, 1982]) shows that the maximum likelihood estimator (MLE) does not exist even for estimating the mean $\mu$ and variance $\sigma^2$ of a Gaussian mixture with two components $N(0, 1)$ and $N(\mu, \sigma^2)$ where $\mu \in \mathbb{R}, \sigma \in \mathbb{R}_+$. In fact, fitting both mean and scale parameters was studied in [Dwivedi

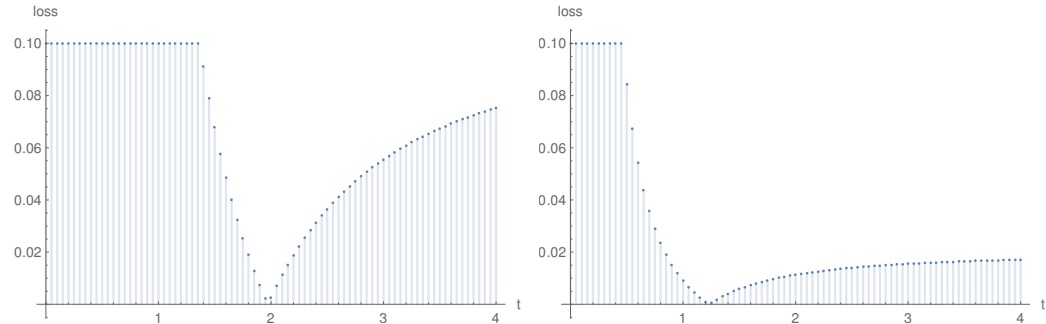

Figure 4: Numerical validation of the performance of the $\delta$-estimator $\hat{\delta}_{\mathrm{corr}}(X_1^n; \theta_\sharp) = \frac{1}{2}(1 - \hat{\rho}(X_1^n; \theta_\sharp))$ where $\hat{\rho}_{\mathrm{corr}}(X_1^n; \theta_\sharp)$ is given by (19). This estimator is used in Theorem 4 (which assumes $\theta_\sharp$ is a mismatched estimate of $\theta_*$), Corollary 5 (which assume $\theta_\sharp = \pm\theta_*$) and also Theorem 7 (which concerns estimating $\theta_*$ without the knowledge of $\delta$). We take the loss $|\hat{\delta} - \delta|$ achieved by $\hat{\delta}_{\mathrm{corr}}$ and the trivial estimators $\hat{\delta}_0(X_1^n) = 0, \hat{\delta}_1(X_1^n) = 1, \hat{\delta}_{1/2}(X_1^n) = \frac{1}{2}$. We plot the loss as a function of $t = \|\theta_*\|$ for $t \in [0, 1]$ with step size 0.05. We take $n = 500, d = 250, \delta = 0.1$. In the left panel, we assume the estimator has access to $\theta_\sharp$ with $\|\theta_\sharp\| = 1.2 \cdot \|\theta_*\|$. In the right panel, we assume $\theta_\sharp = \theta_*$ and therefore the model (2) is equivalent to the model (252).

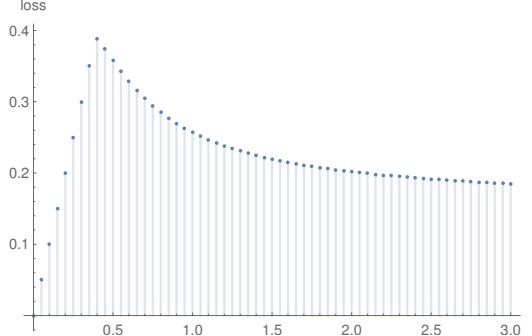

Figure 5: Numerical validation of the performance of Algorithm 1 in Theorem 7. We take the minimum loss (cf. (3)) achieved by Algorithm 1 and the trivial estimator $\hat{\theta}_0(X_1^{3n}) = 0$. We plot the loss as a function of $t = \|\theta_*\|$ for $t \in [0, 4]$ with step size 0.05. We take $n = 100, d = 5, \delta = 0.1$.

et al., 2019, Ren et al., 2022] in the context of the EM algorithm, in a rather restricted setting: The distribution of the samples is standard Gaussian $N(0, I_d)$, yet the estimator is allowed to (over)fit a two-component Gaussian mixture, with symmetric means $\pm\theta$ and a covariance matrix $\sigma^2 \cdot I_d$ with any $\sigma > 0$. Even in this restricted setting, the result is rather delicate and there are differences, e.g., between one- and multi-dimensional models. We finally remark that even method-of-moments based estimators (as the one we use in our paper) the analysis is also typically made for isotropic noise, e.g., [Hsu and Kakade, 2013, Wu and Yang, 2020]. Addressing these issues in the context of Markovian model or models with more sophisticated dependence structures is left as an important yet challenging future task.

2. Another direction beyond Gaussian noise is to look at noise with a heavy-tailed distribution. There has been some recent progress on this topic in high-dimensional statistics [Hopkins, 2020, Hopkins et al., 2020, Cherapanamjeri et al., 2020]. The estimation error rate is expected to depend on the decay rate of the tail. Additional ideas and techniques will most likely be needed in order to handle heavy tails. We leave it for future research.