# OpenReview forum: "Mean Estimation in High-Dimensional Binary Markov Gaussian Mixture Models"
_NeurIPS.cc/2022/Conference — NeurIPS 2022 Accept_

### Official Review · Reviewer_fjjw · 2022-07-11

**Rating:** 6
**Confidence:** 2
**Soundness:** 3 good
**Presentation:** 3 good
**Contribution:** 3 good

**Summary:**

The paper considers high-dimensional mean estimation problem over hidden Markov models (HMMs). It considers whether and to what extent the memory between samples can benefit the estimation problem. Specifically, given a Markov chain S^n with fixed flip probability \delta and correlation \rho between adjacent samples, the goal is to infer the parameter \theta* of the model.

It studies three different memory cases: (i) \delta = 0, (ii) \delta = 1/2, and (iii) \delta\in(0,1/2), where (i) corresponds to infinite memory, (ii) to no memory, and (iii) to simple versions of HMMs. When \delta is known, the authors show that a principal component computation based estimator achieves asymptotically optimal rate. When \delta is unknown, the authors propose an algorithm that first obtains a gross estimate of \theta by assuming the worst memory case, i.e. (iii); then estimates \delta using the estimate of \theta; at last, it refines the estimate of \theta using the new \delta. It shows that the information of knowing \delta plays a role in the performance of the estimator.


**Questions:**

In Algorithm 1, step B relies on the estimate obtained from Step A. I wonder how good the estimate of \theta should be to facilitate the following steps? Do more samples in Step A essentially help? How strong is the assumption of a Gaussian distribution, and can it be replaced with weaker assumptions?


**Limitations:**

The authors may address the limitation of the Gaussian assumption.


**Strengths And Weaknesses:**

The authors were able to demonstrate how memory between the samples affect the rate of estimation. The paper shows sharp bounds on error rates in both low and high-dimensional scenarios. The paper is generally well written. However, some parts of the paper are less clear to the reader.

---

> ### Author Response · Authors · 2022-08-02
> **Response to reviewer fjjw's comments and questions**
>
> 1. "However, some parts of the paper are less clear to the reader."
> We would be grateful if the reviewer could point out the parts that were less clear. We have already revised a few unclear parts and will be glad to further improve the presentation.
> 2. "I wonder how good the estimate of $\\theta$ should be to facilitate the following steps? Do more samples in Step A essentially help?"
> The condition for entering Step B is given by $(d/n)^{1/4}\\lesssim\\|\\hat{\\theta}^{(A)}\\|\\lesssim1$. Otherwise, the algorithm terminates at Step A. We have proved guarantees on the estimation error rate for each of the steps.  The final rate is obtained by properly analyzing all cases and combining the error rates incurred in different steps. Adding more samples to Step A will not change the obtained minimax rate becuase it already uses $n$ samples, and even if all $3n$ samples are used just for Step A, the improvement in the error of Step A will be by a constant factor, which is immaterial for minimax rate analysis.
> 3. "How strong is the assumption of a Gaussian distribution, and can it be replaced with weaker assumptions?"
> The Gaussianity assumption is mild, and can be relaxed in multiple ways:
>     * The Gaussian noise can be relaxed to subGaussian noise: Our minimax upper bound can be shown to hold for subGaussian $Z_i$, since all the concentration bounds for Gaussian random variables we use admit subGaussian analogues. The impossibility result on the minimax rate trivially holds since Gaussian noise is a special case of subGaussian noise.
>     * Isotropic noise can be relaxed to anisotropic noise: Suppose $Z_i\\sim N(0_d,\\Sigma)$ i.i.d. for some $\\Sigma$. Our results can be trivially extended to the case a $\\Sigma\\succ0$ is _known_. The estimator multiplies the samples by $\\Sigma^{-1/2}$ and reduces the problem to the isotropic setting. Then it applies the estimator isotropic estimator we propose, and obtain its final estimator by multiplying this estimate by $\\Sigma^{1/2}$. The loss in this case, however, is gauged by the Mahalanobis distance (parameterized by $\\Sigma$). The problem becomes significantly more challenging if $\\Sigma$ is _unknown_. A well-known example shows this in the context of maximum likelihood estimation (MLE) (see, e.g., [Ferguson 1982](https://www.jstor.org/stable/2287314)): Consider the problem estimating the mean $\\mu$ and variance $\\sigma^2$ of a Gaussian mixture with two components $N(0,1)$ and $N(\\mu,\\sigma^2)$ where $\\sigma>0$. For any given sample $X_1,\\ldots,X_n$ from this distribution, the likelihood function can be arbitrarily large by taking e.g. $\\mu=X_1$ and $\\sigma$ arbitrarily small, and so the MLE does not exist. This is why typical papers on problems similar to ours are limited to mean and weight parameters. For example, there are many recent papers on the expectation maximization (EM) algorithm for estimation of Gaussian mixtures parameters. Almost all of them are restricted to isotropic noise (or known covariance matrix) -- we refer the reviewer to the second paragraph in the revised paper for a partial list. As an exception, fitting both mean and scale parameters was studied in [Dwivedi et al 2019](https://arxiv.org/abs/1902.00194) in the context of the EM algorithm, and in a very restricted setting: The distribution of the samples is standard Gaussian $N(0,I_d)$, yet the estimator is allowed to (over)fit a two-component Gaussian mixture, with symmetric means $\\pm\\theta$ and a covariance matrix $\\sigma^2I_d$ with any $\\sigma>0$. Even in this restricted setting, the result is rather delicate, and there are differences, e.g., between one- and multi-dimensional models. This demonstrates the challenges associated with fitting scale parameters. We finally remark that even method-of-moments based estimators (as the one we use in our paper) are typically analyzed under isotropic noise, e.g. [Anandkumar et al 2014](https://arxiv.org/abs/1210.7559) and [Wu and Yang 2020](https://arxiv.org/abs/1807.07237). Since the trade-off in the minimax rate already involves four parameters $(n,d,\\delta,t)$, we assumed known variance (normalized to $1$) to highlight the role of the memory ($\\delta$).
>     * The Gaussian distribution can be relaxed to a heavy-tailed distribution: There has been some recent progress on this topic (see Appendix G in the revised paper for references). In such model, the estimation error rate will also depend on the decay rate of the tail.  We leave it for future research, because: (1) Our minimax rate already depends on four parameters $(n,d,\\delta,t)$, and adding another one will obscure the main message of the paper. (2)  Additional ideas are needed to handle heavy tails, and this will make the analysis cumbersome and unapproachable.
>
> In the revised manuscript, we added Remark 3 on p. 6 which discusses possible extensions, and also added a new section to the appendix (Appendix G) that discusses more challenging ones (as open problems).

---

### Official Review · Reviewer_9nBR · 2022-07-11

**Rating:** 6
**Confidence:** 4
**Soundness:** 3 good
**Presentation:** 3 good
**Contribution:** 3 good

**Summary:**

This work focuses on obtaining minimax rates in high-dimensional Binary Markov Gaussian Models.  A random sign is multiplied to the sample and this sign is generated by Markovian coin flips w.p $\delta$. The true distribution is a high dimensional Gaussian distribution with identity covariance. Thus this model interpolates between a standard mean estimation problem for Gaussians when $\delta=0$ and when $\delta=1/2$, this is a GMM with two components. The authors provide an estimator that needs to account for the Markovian samples to obtain near optimal minimax error rates (upto log factors) when $\delta$ is known which degenerates to the known minimax rates when $\delta = 0$ or $\delta=1/2$.  Additionally, when $\delta$ is unknown they provide an upper bound of the minimax rates.

**Questions:**

See weaknesses above.

**Limitations:**

Yes, the limitations are somewhat adequately addressed. Please see weaknesses for some clarifications.

**Strengths And Weaknesses:**

Strengths:

This is one of the few recent works that study estimation under dependent samples, with Markovian dependency specifically. The model is interesting as it interpolates between a standard Gaussian location model and Gaussian Mixture model simultaneously. They obtain near optimal minimax rates from the estimators. The estimate relies on carefully splitting the data into batches of appropriate size, so that the local estimate in each batch forms an i.i.d sequence of estimates.

Weaknesses:

1) The main argument relies on the mixing time of the Markov chain that generates the signs and each batch length is decided based on that. It would be good to see why this $\Theta(1/\delta)$ for the sake of completion.  [Nagaraj et al. 2020], apply a variant of SGD for the least squares regression problem with Markovian data and that relies on the mixing time of the Markov chain. Their argument relies on wasting certain amount of data and beyond which there is an approximate. i.i.d behavior in the samples and the errors due to correlation are taken care of.  I am unable to see this argument here and it feels less convincing as to why the data is fully i.i.d after splitting it into batches according to the Markov chain mixing time and the errors due to some residual correlation doesnt seem to appear. Also it is surprising that there is no data wastage. Even if you assume ergodicity, you might have to discard some amount of samples initially. I hope this is clarified better.

2) The results seem to apply only for isotropic Gaussians.

Update:
After the rebuttal, I think the authors have clarified my questions convincingly and I am inclined to increase my score.
References :

Nagaraj, Dheeraj, et al. "Least squares regression with markovian data: Fundamental limits and algorithms." Advances in neural information processing systems 33 (2020): 16666-16676.

---

> ### Author Response · Authors · 2022-08-02
> **Response to reviewer 9nBR's comments and questions**
>
> 1. "It would be good to see why this $ \Theta(1/\delta) $ for the sake of completion."
> The block size is chosen to be $ k=\Theta(1/\delta) $ because this is the largest value such that the ``average gain'' $ \overline{S}_i = \frac{1}{k}\sum_j S_j $ of each block is close to 1 in expectation (cf. Lemma 8 and Eqn. (63)) and with high probability (cf. Eqn. (67)).
>   Having $ \overline{S}_i $ close to 1 makes sure that the block-wise averaging reduces the noise variance but not the signal strength. We have explained the trade-off in choosing the block size on the first paragraph of page 6. We hope that it is clear, but we would be happy to further clarify if needed.
> 2. "Also it is surprising that there is no data wastage. Even if you assume ergodicity, you might have to discard some amount of samples initially."
> We thank the reviewer for drawing the connection to the work of [Nagaraj et al. 2020](https://arxiv.org/abs/2006.08916) which studies the effect of Markovity on linear regression, and uses similar ideas. On a technical level, we assume that the Markov chain is at its stationary distribution already from its first sample ($S_0$ is uniform Rademacher), and so no samples are wasted at initialization. On a higher level, in our problem, the _average_ of all the samples in the block is used to reduce the variance of the Gaussian noise by a multiplicative factor of the size of each block (which is chosen to be $\Theta(1/\delta)$). Since this achieves the minimax rate, it also shows that the "one sample per block" strategy used in [Nagaraj et al. 2020] for the linear regression setting is suboptimal in our problem.  On an intuitive level, the blocks in which there is a sign change, and so the "average gain" $ \overline{S}_i = \frac{1}{k}\sum_j S_j $ is not close to $\pm1$ can be considered as ``bad blocks'', which deteriorate the estimation error. However, due to the optimized choice of block-length as $k=\Theta(1/\delta)$, the affect of bad blocks on the estimation error is provably negligible. We thank the reviewer for pointing out this seemingly counterintuitive result compared to the linear regression problem, and agree it is surprising that no data is wasted. We believe that this is an important message, and presenting it to the community will impact and benefit researchers working in this area.
> 3. "I am unable to see this argument here and it feels less convincing as to why the data is fully i.i.d after splitting it into batches according to the Markov chain mixing time and the errors due to some residual correlation doesn't seem to appear."
> Our estimator partitions the samples into blocks, and then each block is randomized with an i.i.d. Rademacher (the  random sign is common to each of the $k$ blocks but independent among them). This randomization effectively ``restarts'' the Markov chain at the beginning of each block, thus decorrelates it from all previous blocks (due to the Markov property). In other words, instead of using a single instance of a length $n$ Markov chain, the estimator uses $k$ independent instances (blocks) of the Markov chain, each of  length $n/k$. This transformation works for any chosen block length, but it is optimal in terms of estimation error to choose $k$ proportional to the mixing time. Without this randomization, there will indeed be residual correlation between the blocks. However, since our estimator is minimax optimal, eliminating this correlation is essentially optimal.
> 4. "The results seem to apply only for isotropic Gaussians."
> The results were presented for isotropic Gaussians for the sake of clarity, but this assumption can be relaxed in multiple ways:
>   First, the Gaussian noise can be easily relaxed to subGaussian noise, and the isotropic noise can be relaxed to anisotropic noise, if the covariance matrix $\Sigma\succ 0$ is _known_ (by pre-multiplying the samples by $ \Sigma^{-1/2} $ and then post-multipling the isotropic estimator by
> $ \Sigma^{1/2} $). In the revised manuscript, we added Remark 3 on p. 6 that explains this in detail. It is also perceivable that the results can be generalized to unknown covariance matrix, as well as to heavy-tailed distributions. This is, however, more challenging and will require additional ideas, will make the analysis unapproachable, and  ultimately obscure the main message of the paper (memory between samples). We have included a new appendix (Appendix G) that discusses these more challenging open problems. In addition, our answer to Reviewer fjjw has contains more details on this matter (omitted here due to space limitation).

---

> > ### Comment · Reviewer_9nBR · 2022-08-07
> > **Further Clarification**
> >
> > I thank the authors for their detailed responses and their prompt revisions. It would be great if the authors could comment on the generality of their techniques and the model. The Binary Markov model seem useful to model sample dependency in symmetric two component mixture models mainly, is there evidence that these Binary Markov models are previously studied/ useful in practice? It would be great to understand the motivation of studying dependence through a Binary Markov model and for instance what model could be potentially used to study a k-component mixture model?
> >
> >
> > "We thank the reviewer for pointing out this seemingly counterintuitive result compared to the linear regression problem, and agree it is surprising that no data is wasted. We believe that this is an important message, and presenting it to the community will impact and benefit researchers working in this area."
> >
> > I agree that this message is useful and in the context of the above question I posed, could authors clarify under what conditions you will not have this wastage? Is it necessary to use the Binary flip model for the above message?

---

> > > ### Author Response · Authors · 2022-08-08
> > > **Further clarifications**
> > >
> > > 1. "It would be great if the authors could comment on the generality of their techniques and the model. The Binary Markov model seem useful to model sample dependency in symmetric two component mixture models mainly, is there evidence that these Binary Markov models are previously studied/ useful in practice?"
> > > * **Generality:** In a general model, the state $S_{i}$ takes $k\\ge2$ possible values, and the component mean at time $i$ is determined from $k$ possible vectors $\\{\\theta_{*,j}\\}_{j\\in[k]}$ by setting $j=S_i$. In a Markov model, the state transitions form a (hidden) Markov chain. We elaborate below.
> > > * **Previous studies:** The *two-component* symmetric case is addressed in almost all the papers on the Expectation Maximization (EM) algorithm for memoryless models (see our revised Introduction). In the context of HMMs, [Yang et al., 2015] studied the EM (Baum-Welch) algorithm. They derive results for general models ($k\\geq2$), but eventually focus on the binary case which we focus on too. As mentioned in the paper (cf. Sec. 1.4), even in this binary case, the error rate obtained in [Yang et al., 2015] by analyzing the Baum-Welch algorithm is suboptimal in $(n,d,\\delta,\\|\\theta_{*}\\|)$, and this was one of our motivations for this work.
> > > * **Utility in practice:** The binary model is widely applicable since in many cases there are two types of components (note that general component means $\\theta_{ * ,1},\\theta_{ * ,2}$ can be accurately reduced to symmetric means $\\pm\\theta_*$ by empirical centering). For example, consider a list of medical vector samples, each collected from either a male or a female, but this is not labeled. In some cases, it is reasonable to assume that there are long sequences of consecutive male samples followed by long female sequences, and so the Markov model is a good fit. The same applies to a communication signal, for which phase inversions sparsely occur in an unknown manner, but otherwise remain fixed over time. Naturally, the $k>2$ case is also very practical, and we find this generalization to be an important one.
> > > 2. "It would be great to understand the motivation of studying dependence through a Binary Markov model and for instance what model could be potentially used to study a k-component mixture model?"
> > > A natural generalization to $k>2$ is a chain that remains at its current state with probability $1-\\delta$, and changes to one of the other $k-1$ possible states with equal probability $\\frac{\\delta}{k-1}$. Therefore, for blocks of size $\\Theta(1/\\delta)$ symbols, the model roughly acts as a Gaussian location model with a fixed mean (of course, there also will be "bad" blocks). We used this idea to reduce the binary Markov chain to a two-component symmetric Gaussian mixture  model. Similarly, the Markov chain described above can be reduced to a $k$-component Gaussian mixture model, for which efficient estimators exist (e.g., ones based on tensor decomposition), and theoretical guarantees are available. However, in the non-binary case, the dependence of the estimation error on the separation between the components is delicate. For example, one can suggest the minimum pairwise distance. However, [Romanov et al., 2022] recently criticized that, and proposed a different property to quantify "separation". Therefore, we have focused on the binary case, which leads to clear and complete results, and the estimation error is clearly characterized by the signal strength $\\|\\theta_{*}\\|$ (besides $n,d,\\delta$). That being said, the $k>2$ is not an insurmountable barrier, and we believe that our model will be generalized in multiple ways, both by us and by other researchers. We also believe that with some innovation, general Markov kernels and general structure of dependencies (e.g., graphical models) can also be solved.
> > > 3. "I agree that this message is useful and in the context of the above question I posed, could authors clarify under what conditions you will not have this wastage? Is it necessary to use the Binary flip model for the above message?"
> > > The message of "no wastage" is general. For instance, assume that $\\{S_{i}\\}_{i=1}^{n}$ form an Ising model. A natural idea that follows our current paper is to partition the underlying graph of the Ising model in such a way that samples in the same partition are highly dependent, and so the model is accurately approximated by a Gaussian location model in each of the subsets. Then, averaging samples in each subset of variables will reduce the effective noise variance by a multiplicative factor of the size of the partition. Taking one sample per partition ("wasting samples") is then suboptimal compared to averaging samples per subset. The Ising model is rather general, and thus we hope that this demonstrates the fundamental difference between memory in the mean estimation problem and the linear regression problem.
> > >
> > > In light of the above, we would kindly ask you to re-evaluate our
> > > paper.

---

> > > > ### Comment · Reviewer_9nBR · 2022-08-08
> > > > **Thank You Authors**
> > > >
> > > > In light of your response, I am inclined to raise my score to 6 from 5. I would ask the authors in their contribution section to highlight the usefulness of their techniques compared to previous work wrt Ising models and the work on Markovian Linear Regression, in light of the above discussions.

---

> > > > > ### Author Response · Authors · 2022-08-08
> > > > > **Revision in accordance with the above discussion and the reviewer's suggestion**
> > > > >
> > > > > We have added a short paragraph (due to space constraint) to the end of the Contribution section to reflect the reviewer's suggestions.
> > > > > In particular, the usefulness of our techniques in more general models and the connections/differences with prior related work are highlighted.
> > > > > We thank the reviewer for helping us situate our results in a broader context.

---

### Official Review · Reviewer_KdtU · 2022-07-12

**Rating:** 6
**Confidence:** 2
**Soundness:** 3 good
**Presentation:** 3 good
**Contribution:** 3 good

**Summary:**

The authors study an interpolation between Gaussian location models and Gaussian mixture models, in which samples are Gaussian noise-corrupted versions of a sequence in $\\{-1, 1\\}^n$ with each sequence element being the flipping of the previous one with probability $\delta \in [0, \frac12]$. They characterise the minimax risk under Euclidean distance error to the near of $\theta_*$ and $-\theta_*$. They thus generalize known results for the GLM and GMM, extending these results to estimators and risk parameterized by $\delta$, with the GLM and GMM at the extremes. Their ultimate result is to provide a three-step estimator to estimate $\theta_*$ when $\delta$ is not known, and to provide asymptotic results for its loss. This involves estimators for $\theta_*$ under known $\delta$, and for $\delta$ under approximately known $\theta_*$, for which they provide both achievability and lower bounds on worst-case risk within a logarithmic factor of each other.

**Questions:**

1. What motivated or inspired the problem? I'm fairly open to different sorts of motivations; there need not necessarily be concrete practical applications. But it would be good to have an understanding of where this fits into statistical learning, and/or any broader open questions that this contributes towards, and/or any actual or imagined applications. The fact that it is an interpolation between two known extremes doesn't by itself make it interesting: what sorts of questions fall in between the two extremes?

2. I was a bit confused about the argument for why $\overline{X}_i$ is i.i.d. Equation (8) says that $\overline{X}_i$ is equal to $\overline{S}_i \theta^* + \overline{Z}_i$ only in distribution. I think this is correct, because $R_i$ is independently drawn, so would break sure equality but maintain equality in distribution. But then the argument for $\overline{X}_i$ being i.i.d. reasons via $\overline{S}_i$ and $\overline{Z}_i$. I don't this carries over an equality in distribution, unless $\overset{d}{=}$ was intended to indicate equality of the joint distribution of the entire sequence, as opposed to of each element. But also, I don't think it needs to, because I think $\overline{X}_i = R_i \cdot \left[ \overline{S}_i \theta^* + \overline{Z}_i \right]$, and since $R_i$ is also i.i.d., that makes $\overline{X}_i$ i.i.d., right? Please feel free to correct me; any clarification would be appreciated.

Typos/grammar:
- 95: "allows to achieve" should be "allows us to achieve" ("to allow" takes either object + infinitive, or a noun phrase, but not just an infinitive)
- 162: I think Bachmann is spelled with two n's
- 204: "blcok"
- 205: "elemnatary"
- 206: "crucuial"

**Limitations:**

I think the authors were reasonably upfront about questions that were left open. The work was entirely theoretical so I don't expect any societal impact, though I'd still encourage the authors to explain what motivates the problem.

**Strengths And Weaknesses:**

I thought the technical elements of the paper were sound. The problem and its relationship to prior results were clear, there was no ambiguity in notation, and all concepts were explained with suitable precision. I haven't had a chance to walk through the proofs in the appendices step by step, but I read through the proof of Theorem 1 and the approach, based on concentration bounds on a decomposition of the loss, seems sound to me.

Where I think the paper could do with improvement is in the contextualization it provides for the problem and results. Essentially none is given—the authors set up the problem straight away, and the only context given is in Section 1.2, where they state prior results for the GLM and GMM. Why is this problem interesting or significant? Does it have potential applications, or is it inspired indirectly by any potential applications or open questions, or does it get us a step closer to understanding some larger problem? I basically think the lack of contextualization or motivation substantially holds back this paper, and it would be a reasonably strong paper if material was added to address its significance.

-----------------------------

_Edit, 7 August 2022, after author revision:_
- _Increased presentation score to 3 (was 2)_
- _Increased contribution score to 3 (was 2)_
- _Increased overall rating to 6 (was 4)_

---

> ### Author Response · Authors · 2022-08-02
> **Response to reviewer KdtU's comments and questions**
>
> 1. "Contextualization and motivation."
> We agree that our presentation did not reflect well the context and motivation of the problem, and we have made a substantial effort to revise it. In particular, the opening paragraphs of the introduction are now exclusively devoted to this matter. We refer the reviewer to that paragraphs, and here further explain:
>     * The broad question: Statistical dependencies in the data are ubiquitous in practice, and understanding how this memory can be used to improve statistical inference was extensively studied for classical, fixed-dimensional models, but much less for modern, high-dimensional models. Our research goal is to understand how to optimally exploit memory to improve high-dimensional statistical inference. The problem studied in this paper is an instance of this general theme. Since even this basic model was not understood before implies that there is much room for research in this direction.
>     * Motivation: We next describe two problems which were our primary motivation and inspiration. The first one is practice-oriented, and the second one is an on-going theoretical research thread.
>         * Improving estimation using social network data: Consider a population of $n$ individuals, each belong to either one of a few types. Each individual is characterized by a set of features, and the goal is to estimate the means of each type. Without further information, this can be cast as a standard mixture estimation problem. However, in modern applications, one typically has information on the social connections between the $n$ individuals. This can be used to significantly reduce the estimation error, since close friends are typically of the same type. We have focused on the simplest network structure possible – a homogeneous Markov chain, which is important and practical on its own. We expect that our results can be generalized to more complex structures. Since network data is widely available, such an analysis will have a broad practical impact.
>         * The seminal paper of [Balakrishnan et al 2017](https://arxiv.org/abs/1408.2156) has provided the first finite-sample guarantees on the expectation-maximization (EM) algorithm applied to memoryless models with latent variables. This has spurred large research in this direction (see references in the second paragraph of the paper). However, when the results of [Balakrishnan et al 2017] are applied to a two-component Gaussian mixture, they are not minimax optimal, and such optimality was later proved in [Wu and Zhou 2019](https://arxiv.org/abs/1908.10935). Analogously to [Balakrishnan et al 2017], [Yang et al 2015](https://arxiv.org/abs/1512.08269) studied the EM algorithm (Baum-Welch) applied to the Markov setting we address too. Again, [Yang et al 2015] provided the first theoretical guarantees on Markov models, but not minimax tight bounds. The goal of our paper is to determine the minimax rates in this Markov model - a challenging task. Future research on the EM and other algorithms should use our precise minimax results as a benchmark.
>     * Impact: Our message is that Markov memory in the samples should be exploited to reduce the estimation error compared to an i.i.d. model. Nonetheless, it also shows that for high dimensions (specifically $d\\gtrsim\\delta n$), this improvement is negligible (as the minimax rate is as for the GMM $\\delta=1/2$). One may conclude, e.g., that practitioners should focus their attempts on using memory to improve error in low-dimensional models, before such an attempt is made for high-dimensional ones.
>
>     To conclude, our setting stem from a broad and under-explored research theme (“high dimensional statistical inference with samples memory”); has a practical motivation (“improving performance using social networks side information”); is tightly related to an active research thread (“EM and other computationally efficient algorithms for mixture models)”; and has important messages to convey (“what are the regimes in which memory is useful”). We believe that this serves a strong motivation and contextualization for our paper.
> 2. "I was a bit confused about the argument for why $\\overline X_i$ is i.i.d."
> The confusion is in place, and due to our sloppy use of the re-randomization variables $\\{R_i\\}_{i=1}^n$.  In the revised manuscript, we have corrected this, and properly re-defined $\\overline{S}_i$ and $\\overline{Z}_i$ as the average of $S_i$ and $Z_i$, respectively, over a block _multiplied by an independent Rademacher variable $R_i$_. This guarantees that $\\{\\overline{S}_i\\}$ and $\\{\\overline{Z}_i\\}$ are all i.i.d. In fact, due to the re-randomization step, one can w.l.o.g. assume that the averages of $S_i$ and $Z_i$ in different blocks are independent (without residual correlation across blocks),
> and therefore we occasionally omit $R_i$ in the rest of the paper.
>
> 3. All the typos and grammatical mistakes have been fixed. Thank you.

---

> > ### Comment · Reviewer_KdtU · 2022-08-07
> > **Acknowledgement of revision; rating revised**
> >
> > I wish to thank the authors greatly for their work revising the paper and in particular adding substantial information on the context and motivation. This material is very helpful, and accordingly I am happy to revise my rating to 6 (weak accept) (previously 4). I have also changed the presentation and contribution scores to 3 (both previously 2).
> >
> > Just a couple of minor notes:
> >
> > - On the revised re-randomization explanation—would I be correct in believing that given the revised definitions of $\bar{S}\_i$ and $\bar{Z}\_i$, the equality on line 196 (previously eq. (8) in the original manuscript) $R\_i \cdot \frac{1}{k} \sum\_{j \in \mathcal{I}\_i} X\_j \overset{d}=\bar{S}\_i \theta_* + \bar{Z}\_i$ now holds with sure equality, not just equality in distribution?
> >
> > - In the revised introduction, some of the citation formatting is slightly off: I think the author names are meant to be outside the brackets in some places, specifically on line 28 (citing Györfi et al. [2002]), line 37 (citing Balakrishnan et al. [2017] etc.), line 41 (Balakrishnan et al. again), 43 (Wu and Zhou [2019]) and 44 (Balakrishnan et al.).

---

> > > ### Author Response · Authors · 2022-08-07
> > > **Further clarification and revision**
> > >
> > > We are grateful for the reviewer's appreciation of the revised version of the manuscript.
> > > 1. The reviewer is indeed correct regarding the definition of $\\overline{X}_i$. The second equality in the equation on line 196 is in fact an exact equality, not only equality in distribution, given the revised definition of $\\overline{S}_i$ and $\\overline{Z}_i$. We have revised the equation on line 196 in the updated manuscript.
> > > 2. We thank the reviewer for pointing out the misformatted citations. They have been corrected in the updated manuscript.

---

### Official Review · Reviewer_nbjp · 2022-07-25

**Rating:** 6
**Confidence:** 1
**Soundness:** 3 good
**Presentation:** 3 good
**Contribution:** 3 good

**Summary:**

This paper is about the problem of estimating the mean in high-dimensional binary Markov Gaussian mixture models. If $\delta$ denotes the associated flip probability of the stationary homogeneous Markov chain (and is known), the authors explore the regime of $\delta$ between the previously studied values of 0 and ½. Moreover, in the other direction, they show that given the sampled (to be estimated) vector $\theta$, one can compute some rough estimate on the parameter $\delta$. These two results nicely combine into a method for estimating \theta even if \delta is not known.


**Questions:**

Would it make sense to choose the $S_i$ values from some other distribution?

Page 2: Lines 55–56 are not justified, and there is no citation.


Page 5: Could you please elaborate on the second equality of Equation (13)?

Page 6:

Line 204: “blcok” should be “block.”

Line 205: “elemantary” should be “elementary.”

Line 206: “crucuial” should be “crucial.”

Footnote 2: I do not understand the last sentence.

Page 7: Please explain why Equation (19) yields a natural estimator.

Page 8: Please further explain lines 280–283.

Page 9: Can you please elaborate on the claims of “The impact of lack of knowledge of $\delta$?

**Limitations:**

I did not see an explicit section addressing this. However, this may not be relevant to this paper.

**Strengths And Weaknesses:**

Originality: The originality of this paper mostly stems from its ambitious goal, namely, to understand the problem of mean estimation in high-dimensional binary Markov Gaussian mixture models for the case where the flip probability $\delta$ can be any value in the interval [0,½]. Prior to this work, only the cases $\delta = 0,½$ had been studied.

Quality: The quality of the paper is good. It is well written and the results seem strong.

Clarity: The presentation of the paper and writing are generally clear. Section 1.3 (Contributions) could be made a bit more reader-friendly.

Significance: The work is significant: see Originality above. Another non-trivial reason that this paper is significant is about the techniques employed therein; in particular the use of a principal component of a suitable covariance matrix that is used in the aforementioned smooth interpolation between $\delta$ = 0 and $\delta$ = ½. I found that quite interesting!

---

> ### Author Response · Authors · 2022-08-02
> **Response to reviewer nbjp's comments and questions**
>
> 1. "Section 1.3 (Contributions) could be made a bit more reader-friendly."
> We have thoroughly revised this section.
>
> 2. "Would it make sense to choose the $ S_i $ values from some other distribution?"
> Yes, though we focused on the binary uniform case as it allows to obtain a clear view of the trade-off between $(n,d,t,\\delta)$. Indeed, in the two-component case, the components can be centered and then the norm of the mean vector $t$ is a measure of "separation". If $ S_i $ takes values in a larger finite set, then the model has multiple components, and the resulting minimax rates are much more delicate. First, the ``separation'' of the multiple components need to be parameterized. Minimum pairwise distance is a natural choice, but there are recent papers [Romanov et al., 2022](https://arxiv.org/abs/2202.07707) which doubt that and propose global parameters. Second, it is known [Jin et al., 2016](https://arxiv.org/abs/1609.00978) that even with just three components, the likelihood can contain local maxima, suggesting that the multi-component problem is significantly more challenging. Another possible generalization is non-uniform distribution. Here too it is known [Weinberger and Bresler, 2022](https://arxiv.org/abs/2103.15653) that local maxima exist even for memoryless models.  We presume that any result on multiple or non-uniform components will hinge on our result (and novel ideas), and we hope that presenting the paper at the conference would spur such research.
> 3. "Page 2: Lines 55-56 are not justified, and there is no citation."
> The analysis of the estimator $\\hat\\theta=0$ is trivial, and the analysis of the averaging estimator leads to the parametric error rate of $\\Theta(\\sqrt{d/n})$ known for the Gaussian location model. We have added a citation to a full rigorous proof.
>
> 4. "Page 5: Could you please elaborate on the second equality of Equation (13)?"
>   The definition of $ \\overline{X} $ in Eqn. (8) implies
>   $$ \\mathbb{E}[\\overline{X}\\, \\overline{X}^\\top]
>   = \\mathbb{E}[(\\overline{S}\\theta_*+\\overline{Z})(\\overline{S}\\theta_*+\\overline{Z})^\\top] \\\\
>   = \\mathbb{E}[\\overline{S}^2] \\theta_*\\theta_*^\\top + \\mathbb{E}[\\overline{S}\\theta_*\\overline{Z}^\\top] + \\mathbb{E}[\\overline{S}\\,\\overline{Z}\\theta_*^\\top] + \\mathbb{E}[\\overline{Z}\\,\\overline{Z}^\\top] \\\\
>   = \\xi_k \\theta_*\\theta_*^\\top + \\frac{1}{k} I_d . $$
>   The last equality follows since $ \\overline{Z}\\sim  N(0, I_d/k) $ and is independent of $\\overline{S}$ (therefore the cross terms vanish). This was omitted due to space limitation.
>
> 5. "Footnote 2: I do not understand the last sentence."
>   The PCA-based estimator analyzed in Theorem 1 yields a rate higher than the minimax rate of Theorem 2 whenever the signal strength is too low (see also Eqn. (7)).
>   However, since the estimator is assumed to know $t$ (a common formulation in high-dim statistics), it can just output zero, and obtain the promised rate in the low signal strength regime.
>   Thus, for any signal strength, the minimax rate is achieved by the minimum rate of the PCA-based estimator and the zero estimator.
>   We have revised the footnote in the paper accordingly.
>
> 6. "Page 7: Please explain why Equation (19) yields a natural estimator."
>   This is a natural estimator since it replaces the population mean
>   \\begin{align}
>   \\mathbb{E}[X_i^\\top X_{i+1}] &= \\mathbb{E}\\left[(S_i \\theta_* + Z_i)^\\top(S_{i+1} \\theta_* + Z_{i+1})\\right]
>   = \\rho \\|\\theta_*\\|^2 , \\notag
>   \\end{align}
>   with an empirical mean computed over pairs of adjacent samples. We have revised the sentences leading to Eqn. (19) (now Eqn. (11)) to better explain this.
>
> 7. "Page 8: Please further explain lines 280-283."
> Per the minimax rates of GLM and HMM (Eqn. (5) and (6), respectively, see also Fig. 1), we see that if $ \\|\\theta_*\\|\\gtrsim1 $, then the best rate possible is the parametric rate $ O(\\sqrt{d/n}) $ of GLM, regardless of the value of $\\delta$.  In that case, it is information-theoretically impossible to exploit the Markov structure to reduce the error rate. This rate, however, can already be achieved by the PCA estimator in Step A, and so Algorithm 1 terminates at this step.
>
> 8. "Page 9: Can you please elaborate on the claims of "The impact of lack of knowledge of $\\delta$"?"
> We have revised this paragraph in the paper. In Sec. 1.3 we state our bound on the minimax rate (Eqn. (7)) and explain how memory of $\\delta$ improves estimation error compared to $\\delta=1/2$ from three aspects. This bound, however, is for an estimator which knows $\\delta$. In the paragraph under question we revisit these three improvements in case $\\delta$ is unknown and Algorithm 1 is used. The claims made in the paragraph can be deduced from Theorem 7, in the same way they have been deduced from Theorem 2.
>
> 9. All the typos and grammatical mistakes have been fixed.

---

### Author Response · Authors · 2022-08-02
**A summary of our response to the reviewers and the revision of the paper**

We have answered all the comments made by the reviewers, and also
notably revised the paper. We believe that the paper has significantly
improved from the main aspects raised by the reviewers. Specifically:
* Re reviewer KdtU (Rating 4): The reviewer said that the paper is solid
and clear, and her/his main claim is that the paper lacks contextualization
and motivation. We fully agree that the presentation in the original
submission was lacking from this aspect, and we have significantly revised
the paper. Specifically, the motivation and context of the problem are now highlighted in the opening two paragraphs
of the paper. These paragraphs present the broad research goal that we address, and two important motivations -- one that is practice-oriented,
which stems from the availability of side information in modern-day estimation
problems, and the second is theory-oriented, which stems from a recent
on-going effort of understanding the minimax rates of high dimensional
mixture models. We then conclude this part with a sentence describing
the possible broader impact of our obtained results. We believe that
this explanation serve as a significant motivation and contextualization
for the problem studied in this paper. We have also explained this with
more detail in our response to the reviewer (within the space limitation).

* Re reviewer 9nBR (Rating 5): The reviewer was surprised that our estimation
techniques and analysis are capable of establishing minimax rates.
We have made our best attempt to clarify the doubts of the reviewer,
and explain why our techniques are both correct and natural. We believe
that having our techniques surprise an expert in this problem domain
is nothing but a strong motivation for the paper to be presented to the research community. We have also
fully addressed all the other questions of the reviewer, both as detailed
answers to the reviewer, as well as in a revision of the paper (e.g.,
Remark 3 on pp. 6-7 and Appendix G that address the Gaussian assumption).

* Re reviewers nbjp and fjjw (Rating 6): Both the reviewers appreciated
the ambitious goal of the paper, its quality, its writing and its
results. We believe that our revision based on their comments further improved the paper.

---

### Meta-Review · Area_Chair_4ccq · 2022-08-24

**Recommendation:** Accept
**Confidence:** Certain

**Metareview:**

The paper addresses the problem of high-dimensional statistical inference from dependent samples. This is a recently emerging area, and the authors establish nearly tight minimax error rate bounds for a basic statistical model (gaussian hidden markov model).

The reviewers appreciated the technical strength of the paper, but there were some questions about the framing and context of the problem. The authors clarified these issues suitably in their rebuttal, clearing the way for acceptance.

**Award:**

No

---

### Decision · Program_Chairs · 2022-09-14

Accept